## RESOURCE
# Single-cell RNA sequencing reveals evolution of immune landscape during glioblastoma progression

Alan T. Yeo[1,2,7], Shruti Rawal[1,7], Bethany Delcuze[1,2,7], Anthos Christofides[1], Agata Atayde[1], Laura Strauss[1], Leonora Balaj[3], Vaughn A. Rogers[1], Erik J. Uhlmann[4], Hemant Varma[5], Bob S. Carter[3], Vassiliki A. Boussiotis[1,6 ✉] and Al Charest[1,6 ✉]

Glioblastoma (GBM) is an incurable primary malignant brain cancer hallmarked with a substantial protumorigenic immune component. Knowledge of the GBM immune microenvironment during tumor evolution and standard of care treatments is limited. Using single-cell transcriptomics and flow cytometry, we unveiled large-scale comprehensive longitudinal changes in immune cell composition throughout tumor progression in an epidermal growth factor receptor-driven genetic mouse GBM model. We identified subsets of proinflammatory microglia in developing GBMs and anti-inflammatory macrophages and protumorigenic myeloid-derived suppressors cells in end-stage tumors, an evolution that parallels breakdown of the blood–brain barrier and extensive growth of epidermal growth factor receptor[+] GBM cells. A similar relationship was found between microglia and macrophages in patient biopsies of low-grade glioma and GBM. Temozolomide decreased the accumulation of myeloid-derived suppressor cells, whereas concomitant temozolomide irradiation increased intratumoral GranzymeB[+] CD8[+]T cells but also increased CD4[+] regulatory T cells. These results provide a comprehensive and unbiased immune cellular landscape and its evolutionary changes during GBM progression.

GBM—an incurable primary malignant brain cancer—has heightened genomic and cellular heterogeneity, a 14-month median survival and absence of an effective treatment. IDH1 wild-type (WT) GBMs are classified into transcriptomic subtypes (classical, proneural and mesenchymal), with defining genetic mutations[1]. Classical GBMs are characterized by epidermal growth factor receptor (EGFR) amplification and mutations often associated with losses of *CDKN2A* and *PTEN* tumor suppressor genes[1]. Building on this clinical information, we genetically engineered mice to faithfully mimic these genomic events and modeled classical GBMs[2–5], to study GBM development in an immune competent context. GBMs are heavily (>30% of cellular mass) infiltrated by immune cells[6]. Cataloging this diversity using single-cell RNA-sequencing (scRNA-seq) is an ongoing process[7], and a full representation of the immune composition during GBM initiation, progression and standard of care (SOC) (ionizing radiation (IR) and temozolomide (TMZ)) treatments remains absent.

Interpatient and intratumoral heterogeneity poses a formidable challenge to the successful development of targeted and immunotherapeutic approaches for the treatment of GBM. An in-depth understanding of the interactions between GBM cells and their immune microenvironment is therefore critical and such knowledge is currently lacking. Here, we performed droplet-based scRNA-seq to study the immune and the nonimmune composition of mouse GBMs at single-cell resolution, and performed longitudinal analyses of populations during initiation and progression. We investigated the effects of SOC on the immune composition of GBM in

mice by flow cytometry. We observed drastic changes in the innate immunity population during progression from proinflammatory microglia early during GBM development to high infiltration of immunosuppressive macrophages and neutrophils in end-stage GBMs. We validated the clinical relevance of these distinct immunological profiles by analyzing specimens collected from patients with low-grade glioma and GBM. In both patient samples and our mouse GBM model, dynamics of tumor cells growth coincided with infiltration of immunosuppressive cells. In addition, we identified populations of myeloid- and lymphoid-derived cells that are present only in GBM. Together, our results establish a road map of events that leads to the establishment of the immunosuppressive characteristics of GBMs and offer an in-depth, unbiased resource for studies designed toward therapeutic interventions for GBM.

## Results

**scRNA-seq identifies tumor, stromal and immune cells in GBM.** To determine cellular interactions during GBM progression, we leveraged our de novo GBM mouse model[2–4] wherein tumors are initiated in situ from conditional (Cre/Lox) overexpression of human *EGFR* combined with loss of *Cdkn2a* and *Pten*. A conditional luciferase reporter gene[8] was used for monitoring GBM growth by bioluminescent imaging (BLI). Stereotactic intracranial injections of a Cre lentivirus initiated GBMs and progression was monitored by BLI, and corresponded to tumor volume (Fig. 1a,b). Along with tumor-free normal brains as controls, GBMs were harvested for scRNA-seq at two stages of growth (Early <10[8] BLI and Late >10[8] BLI)

[1]Department of Medicine, Beth Israel Deaconess Medical Center, Harvard Medical School, Boston, MA, USA. [2]Sackler School of Graduate Studies, Tufts University School of Medicine, Boston, MA, USA. [3]Department of Neurosurgery, Massachusetts General Hospital, Harvard Medical School, Boston, MA, USA. [4]Department of Neurology, Beth Israel Deaconess Medical Center, Harvard Medical School, Boston, MA, USA. [5]Department of Pathology, Beth Israel Deaconess Medical Center, Harvard Medical School, Boston, MA, USA. [6]Cancer Research Institute, Beth Israel Deaconess Medical Center, Boston, MA, USA. [7]These authors contributed equally: Alan T. Yeo, Shruti Rawal, Bethany Delcuze. ✉e-mail: vboussio@bidmc.harvard.edu; acharest@bidmc.harvard.edu

971

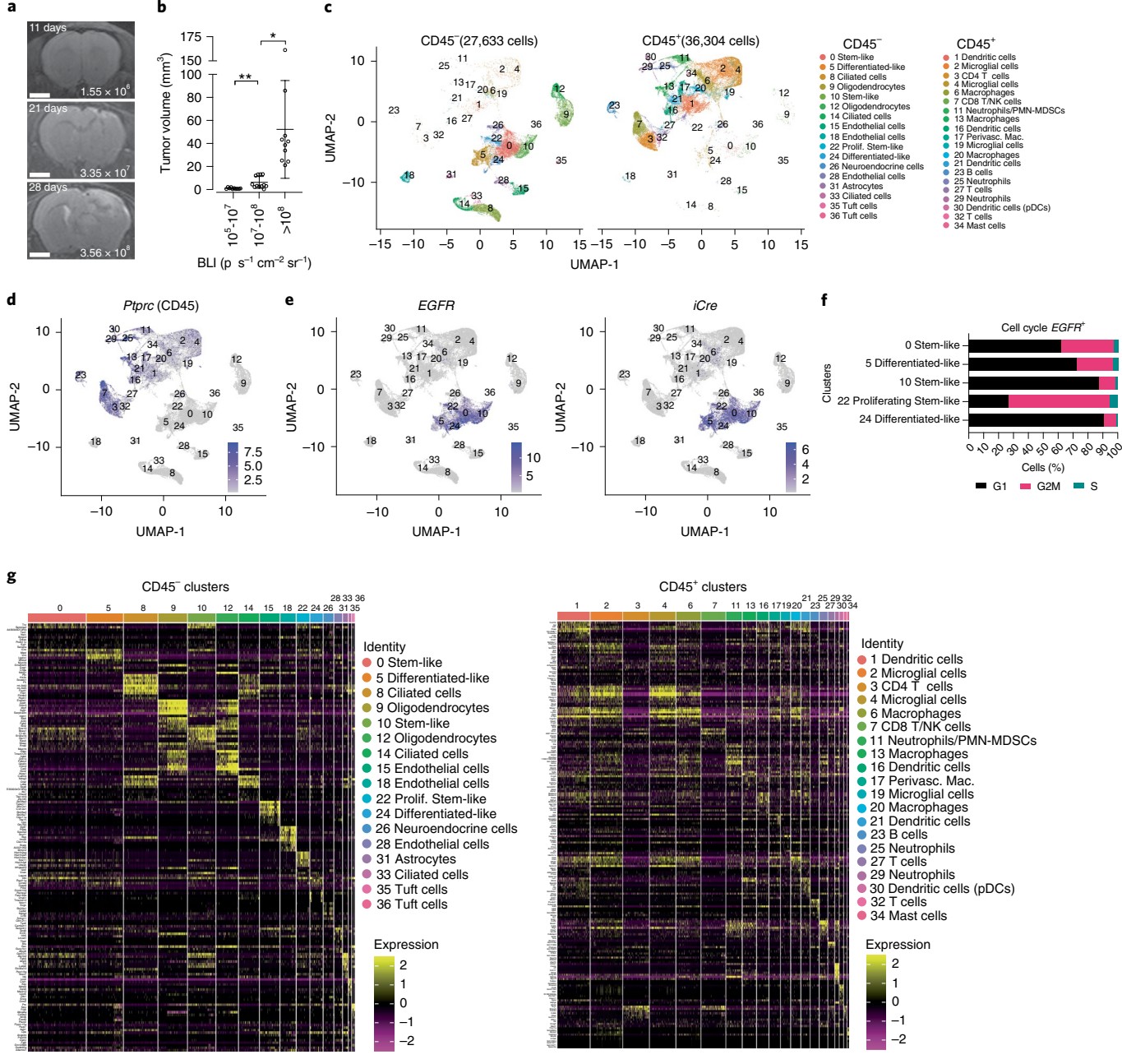

**Fig. 1 | scRNA-seq identifies CD45⁻ and CD45⁺ cell populations in GBM. a**, Representative MRIs of GBM development over time with associated BLI outputs. Scale bar, 2 mm. **b**, MRI–BLI, data are presented as mean values ± s.d. of biologically independent replicates, *$P = 0.013$, **$P = 0.0045$, unpaired $t$ test, two-tailed, $n = 10$, 12 and 10 for BLI $10^5$–$10^7$, $10^7$–$10^8$ and >$10^8$, respectively. **c**, UMAP of cell types clustered by single-cell transcriptional analysis of brain CD45⁻ ($n = 27,633$) and CD45⁺ ($n = 36,304$) cells isolated from normal and GBM mice. **d**,**e**, Expression of *Ptprc* (CD45) (**d**), human *EGFR* gene (*EGFR*) and *iCre* recombinase (**e**) transcripts overlaid on the CD45⁻ and CD45⁺ UMAP space. **f**, Percentage of cells expressing G1, G2M and S phase markers from the indicated *EGFR*⁺ GBM cell clusters. **g**, Hierarchical clustering of CD45⁻ and CD45⁺ cells grouped by top expression of genes.

(Extended Data Fig. 1a and Supplementary Table 1). CD45⁻ tumor/nonimmune cells and CD45⁺ immune cells were sorted by flow cytometry and subjected to scRNA-seq (Extended Data Fig. 1b). Uniform manifold approximation and projection (UMAP) dimension reduction was performed on 27,633 CD45⁻ cells and 36,304 CD45⁺ cells, resulting in 17 and 20 clusters from the CD45⁻ and CD45⁺ samples, respectively (Fig. 1c and Supplementary Table 2). Expression of *Ptprc* (Cd45) is exclusively restricted to the CD45⁺ sorted samples (Fig. 1d). *EGFR* and *iCre* clusters were identified as tumor cells (Fig. 1e) and related more to human scRNA-seq categories defined by Johnson-Verhaak[9] than those defined by

Neftel-Suvà[10] (Extended Data Fig. 1c). We thus adopted the Johnson-Verhaak nomenclature to label *EGFR*⁺ tumor clusters (Fig. 1c,f) with cluster 22 proliferating stem-like cells displaying the most proliferating cells (Fig. 1f).

Top differentially expressed genes from each pooled population were identified (Fig. 1g). Cluster cell types were identified based on the expression of known marker genes (Extended Data Fig. 1d). Four clusters (1, 16, 21 and 30) of dendritic cells (DCs) were detected (Fig. 1c,g and Extended Data Fig. 1d). We identified cluster 16 as conventional DCs subset cDC1 (*Flt3*⁺*Irf8*⁺*Xcr1*⁺) and cluster 30 as plasmacytoid DCs (*Siglech*⁺, *Ccr9*⁺ and *Pacsin1*⁺)

(Extended Data Fig. 1d) whereas the other two clusters express core dendritic cell markers (*Cd74* and *H2-Aa* (MHCII[hi])). We detected four clusters of macrophages (6, 13, 17 and 20). We identified cluster 17 macrophages as brain-resident perivascular macrophages (*Cd163*[+] *Mrc1*[+])[11–13] and cluster 13 as border-associated macrophages (BAM) that included signature genes *Lyz2*, *Ms4a7*, *Ms4a6c* and *Tgfbi*[14] (Extended Data Fig. 1d). Four clusters of T cells/natural killer (NK) cells were observed. Cluster 3 are CD4 T cells (*Cd4*[+]), cluster 7 is composed of CD8 T cells (*Cd8*[+]) and NK cells (*Gzma*[+]) and clusters 3 and 32 contain regulatory T cells (Tregs, *Cd4*[+] *Il2ra*[+]) (Extended Data Fig. 1d). Three clusters of microglia (*P2ry12*[+] and *Tmem119*[+]) (2, 4 and 19) were identified and three clusters of neutrophils (11, 25 and 29) based on expression of *S100a8*, *S100a9* and *Ly6g* were observed (Extended Data Fig. 1d). One cluster each of B cells (cluster 23, *Cd79a*[+], *Igkc*+) and mast cells (cluster 34, *Cpa3*[+]) were identified.

**Regionalization of transcriptionally distinct *EGFR*[+] cells.** To define the underpinnings of the distribution of *EGFR*[+] tumor cells into five transcriptionally distinct clusters, we determined the number and area of EGFR[+] cells and showed a progression from multiple EGFR[+] single-cell to small independent clusters to a single observable mass over time (Fig. 2a and Extended Data Fig. 2a). This oligoclonality in early- and mid-lesions is not unequivocal evidence that late-stage tumors are composed of independently arising clones. Lineage tracing using the multi-fluorophore Cre-reporter allele R26R-*Confetti* strain[15] performed on GBMs harvested 10 and >30 days postinjection demonstrated that all four fluorophores were present at comparable levels and distributed evenly with similarly variegated patterns of expression, suggesting multiple clones are in residence in early and end-stage tumors (Fig. 2b,c). This oligoclonal nature, however, is discordant to the scRNA-seq data that identified three transcriptionally distinct populations of tumor cells. In fact, deciphering the transcriptional trajectories of single tumor cells using Monocle3 pseudotime tracing[16] suggest that cells in cluster 10 gives rise to cells composing clusters 0, 22 and 24, whereas cluster 5 cells represent an independent origin (Fig. 2d).

To better understand the molecular basis driving the separation of stem-like (clusters 0 and 10) from differentiated-like cells (clusters 5 and 24), we identified differentially expressed genes (DEGs) (Fig. 2e). Gene ontology (GO) pathway analysis pointed to significantly upregulated pathways pertaining to responses to INFα/β/γ, cell migration and angiogenesis in cluster 5 and oligodendrocyte differentiation, myelination and cell adhesion in cluster 0 (Extended Data Fig. 2b). The high proliferative index of cells in cluster 22 (Fig. 1g) is represented by spindle organization, mitotic cytokinesis, chromosome segregation and cell division (Extended Data Fig. 2c). Surprisingly, the mode of EGFR signaling of clusters 0/10 and 5/24 appears to be distinct. Cells from clusters 0/10 have increased expression of *Tgfα* while those from clusters 5/24 have increased expression of *Hbegf* (Fig. 2e,f)—two EGFR ligands that exert differential signaling in cancer models[17–19]. Other EGFR ligands were detected in neuroendocrine cells (*Egf* and *Btc*), microglia (*Btc*) and CD4[+] T cells (*Areg*) (Extended Data Fig. 2d).

The genes most differentially expressed in cells from clusters 0/10 and 5 are *Pdgfra* and *Lgals1* (Gal1), respectively (Fig. 2e,g and Extended Data Fig. 2e). To better define signaling differences downstream of EGFR in these two populations, CD45[−]EGFR[+] PDGFRA[+] and CD45[−]EGFR[+] GAL1[+] cells were flow sorted from end-stage GBMs and their transcriptome resolved by bulk RNAseq (Fig. 2h and Extended Data Fig. 2f–h). GO analysis of DEGs revealed that PDGFRA[−] cells are enriched in genes important for neutrophil chemotaxis, inflammatory and immune response (Extended Data Fig. 3a,b) similar to GAL1[+] cells (Extended Data Fig. 3c,d). Analyzing DEGs between PDGFRA[+] and GAL1[+] cells showed that the latter are enriched in transcriptomes of neutrophil chemotaxis (*Ccl12*, *Csf3r*, *Cxcl3*, *Cxcl2*, *Ccl9*, *Ccl8*, *Ccl6*, *Ccl5*, *Ccl4*, *Cxcr2*, *Ccl3* and *Ccl24*), whereas upregulated genes in PDGFRA[+] population involve oligodendrocyte differentiation (*Nlgn3*, *Slc8a3*, *Ntrk2*, *Nkx6-2*, *Ptprz1*, *Olig1*, *Sox8*, *Olig2*, *Ascl1* and *Sox6*) (Fig. 2i and Extended Data Fig. 3e). Reactome analysis of DEGs between EGFR[+]PDGFRA[+] and EGFR[+] GAL1[+] cells showed upregulation of mechanisms of translation, Il-1 signaling regulation, and immunoregulation of interactions between lymphoid and nonlymphoid cells in the EGFR[+]GAL1[+] samples and neuronal system, synaptic functions and protein interactions at synapses in EGFR[+]PDGFRA[+] cells (Fig. 2j).

These differences in signaling from EGFR[+] GBM cells led us to test whether regional attributes define cells from clusters 0, 10, 5, 24 and 22. We leveraged The IVY Genome Atlas Project (IVY GAP)[20] and superimposed the most highly expressed gene in each cluster onto the expression patterns of patient samples. Genes from clusters 5 and 24 shared expression patterns with GBM cells dissected from pseudopalisading regions and the perinecrotic zone (Extended Data Fig. 3f). Genes from cluster 22 shared patterns with cellular tumor, hyperplastic blood vessels and microvascular proliferation, whereas top genes of cluster 10 were highly upregulated in cellular tumor bulk, leading edge and infiltrating tumor regions (Extended Data Fig. 3f). Similarly, alignment of highly expressed genes from EGFR[+] PDGFRA[+] and EGFR[+]GAL1[+] revealed that EGFR[+]PDGFRA[+] cells display profiles similar to those from cells of cellular tumor, leading edge and infiltrating tumor, whereas EGFR[+]GAL1[+] cells express genes similar to perinecrotic zones, pseudopalisading regions and hyperplastic blood vessels (Extended Data Fig. 3g). Together, these findings demonstrate a persistent oligoclonal nature of the model and suggest that the distinct gene expression profiles observed in the EGFR[+] cancer cells are potentially influenced by their ligand usage and localization in the tumor, ultimately shaping considerable heterogeneity in signaling pathway activation patterns.

**GBMs harbor a proliferative population of microglia.** In the uninjured brain, microglia have limited self-renewal capacities[21]. The presence of three clusters of microglia in GBM may represent a functional categorization. In fact, microglia cluster 19 cells express higher levels of G2M and S phase cell cycle genes than clusters 2 and 4 (Fig. 3a). Additionally, cluster 19 S phase cells increased considerably in early and late GBM mice compared with normal brain controls (Extended Data Fig. 4a). Validation using systemic

---

**Fig. 2 | GBMs are composed of transcriptionally distinct populations of EGFR[+] cancer cells. a**, Area (in mm²) of EGFR[+] cell clusters at 7, 14 and 21 days post Cre virus injection. Data are presented as mean ± s.e.m. of biologically independent tumors, *P < 0.0001, unpaired *t* test, two-tailed, *n* = 9 (three or four sections per tumor), 16 and 11 for 7, 14 and 21 days, respectively. **b**, Confetti sections of a > 30 days late-stage GBM. Scale bars, left panel 100 μm, right panel 500 μm. NB, normal brain. **c**, Fluorescent cells (A and B) or clusters (C and D) from biologically independent mice bearing early-stage (10 days) and late-stage (>30 days) post Cre virus injection. Data are presented as mean ± s.e.m. of *n* = 4–6 fields of view per section and *n* = 4–5 sections per biologically independent tumors were quantified. **d**, Monocle3 trajectory inference on *EGFR*[+] clusters. **e**, DEGs between *EGFR*[+] clusters 0 and 5. NS, not significant. **f**, Expression of EGFR ligands *Hbegf* and *Tgfa* overlaid on UMAPs. **g**, Expression of *Lgals1* (GAL1) and *Pdgfra* overlaid on UMAPs. **h,i**, Heat map (**h**) and volcano plot (**i**) of DEGs from bulk RNAseq of hsEGFR[+]PDGFRA[+] and hsEGFR[+]GAL1[+] flow-sorted cells. **j**, Reactome analysis of upregulated genes in hsEGFR[+]PDGFRA[+] and hsEGFR[+]GAL1[+] cell populations.

in vivo DNA labeling with 5-ethynyl-2′-deoxyuridine (EdU) and flow cytometry showed a fivefold increase in EdU⁺ microglia from GBM mice compared with control brain (10.07 ± 2.21% versus 2.22 ± 1.92%) (Fig. 3b), demonstrating that increased proliferation of microglia is occurring specifically in GBM tissues. In fact, flow cytometry of tumor bed and contralateral brain over time for total

and Ki67⁺ microglia cells demonstrated increases only in tumoral microglia during GBM progression (Fig. 3c,d).

To better define this cycling microglia population, we used flow cytometry to isolate (Extended Data Fig. 4b–d) EdU⁺ and EdU⁻ microglia, contralateral GBM microglia, normal brain microglia and GBM macrophages of late-stage GBMs from age- and

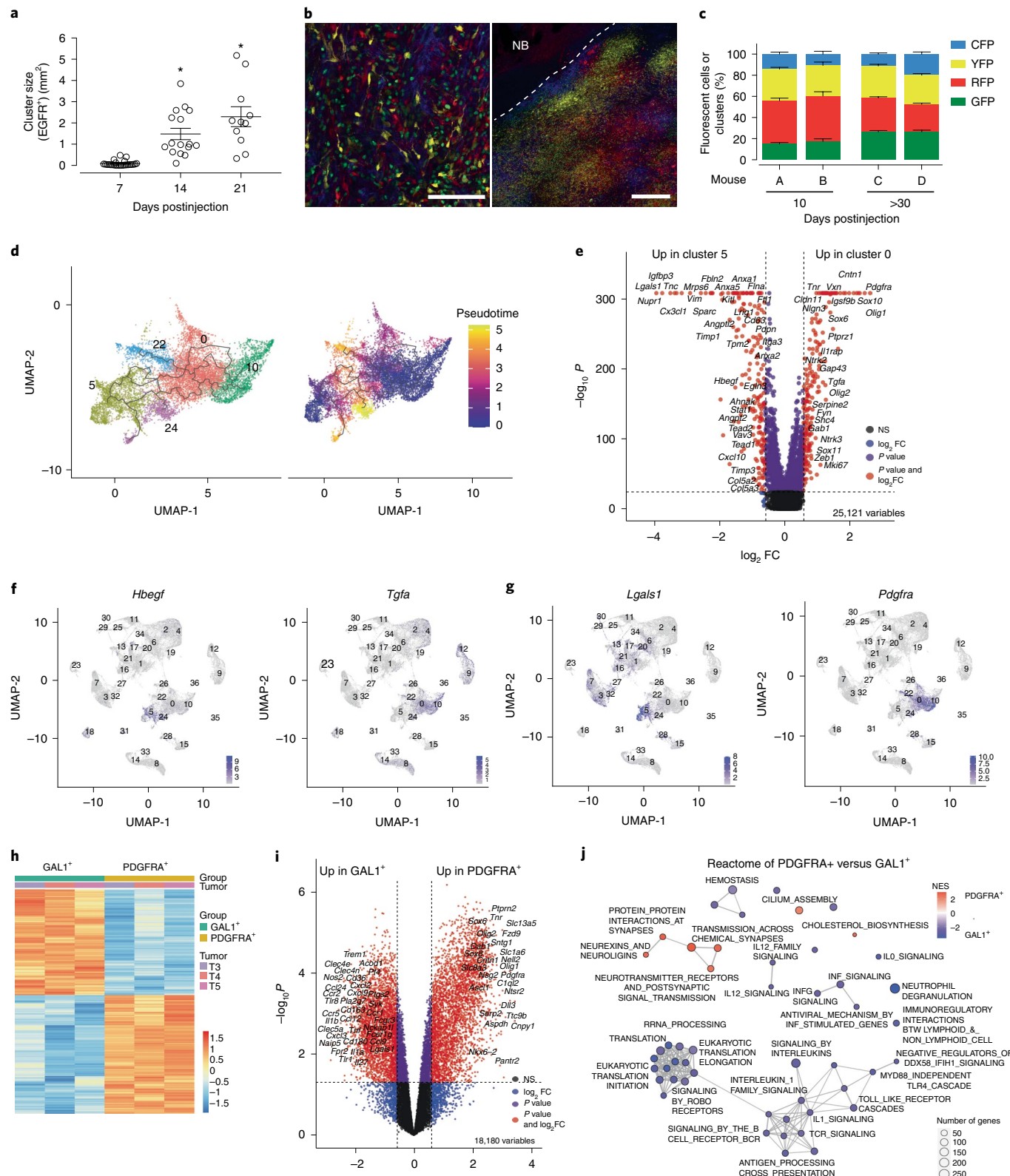

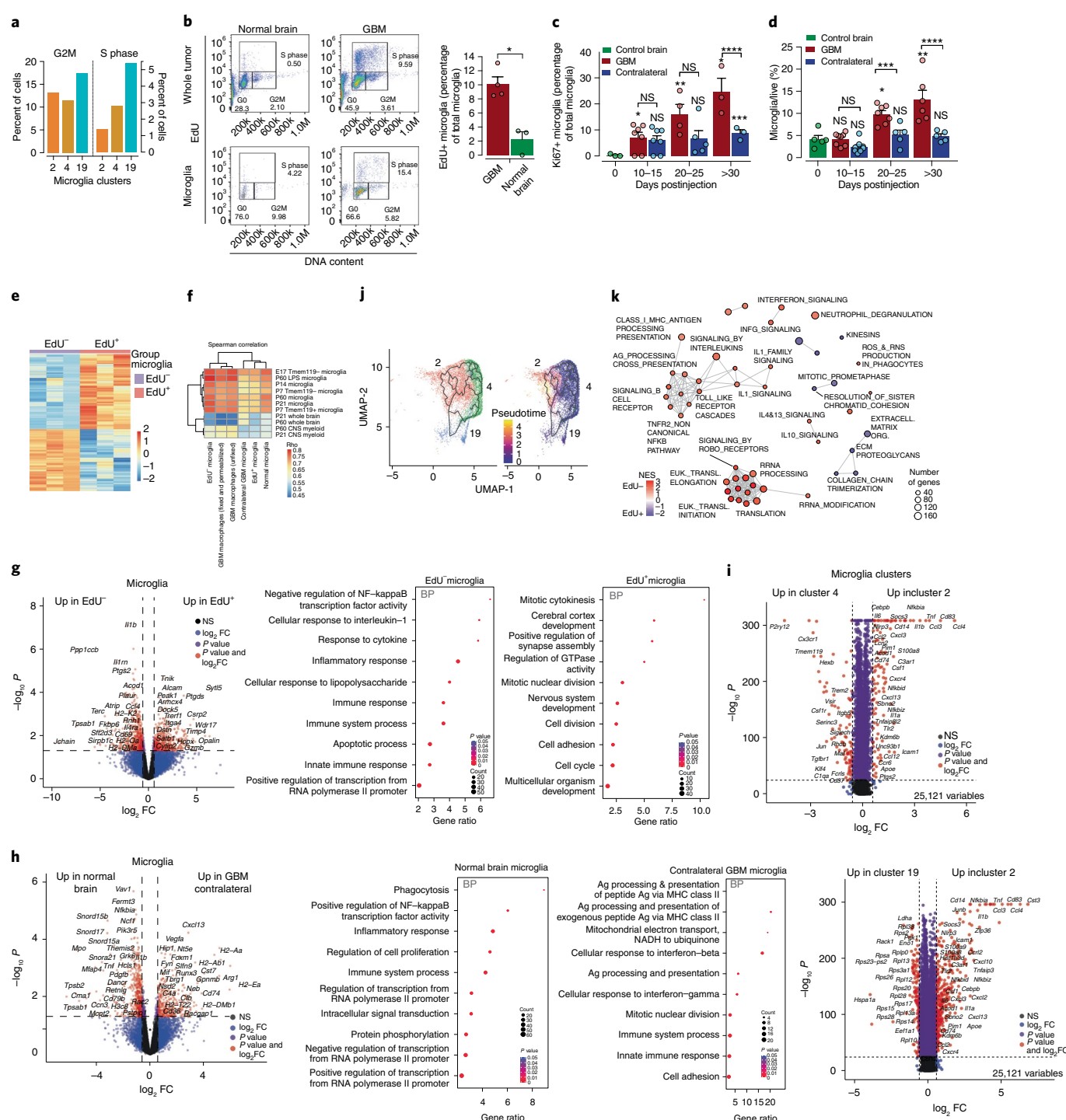

**Fig. 3 | Proliferative microglia are present in EGFR GBMs. a**, Expression of cell cycle genes in microglia clusters 2, 4 and 19 populations. **b**, EdU flow plots and quantitation of microglia. Data are mean ± s.d. of biologically independent replicates GBM *n* = 4, normal brain *n* = 3. *\*P* = 0.028, unpaired *t* test, two-tailed. **c**, Proliferative index of GBM microglia over time. Ki67 positive microglia as percentage of total microglia. Data are mean ± s.d. of biologically independent replicates, control brain day 0 *n* = 3, GBM and contralateral *n* = 7, 4 and 3 for days 10–15, 20–25 and >30, respectively. Statistical analysis, without brackets relative to control brain microglia (day 0). *\*P* = 0.0171; *\*\*P* = 0.0104; *\*\*\*P* = 0.0477; *\*\*\*\*P* = 0.0039; *\*\*\*\*\*P* = 0.0449; NS, not significant, unpaired *t* test two-tailed. **d**, Increase in microglia content during GBM progression plotted as percentage of live cells. Data are mean ± s.d. of biologically independent replicates. Control brain day 0 *n* = 5, GBM *n* = 8, 7 and 5 and contralateral *n* = 8, 5 and 5 for days 10–15, 20–25 and >30, respectively. Statistical analysis, without brackets relative to control brain microglia (day 0). *\*P* = 0.0008; *\*\*P* = 0.0055; *\*\*\*P* < 0.0054; *\*\*\*\*P* = 0.0072; NS, not significant, unpaired *t* test two-tailed. **e**, Heat map of DEGs from bulk RNAseq of EdU+ and EdU− flow-sorted microglia. **f**, Spearman correlation of transcriptomes of flow-sorted microglia (EdU+, EdU−), macrophage and contralateral microglia to indicated bulk RNAseq datasets. **g**, DEGs and GO analysis between EdU+ and EdU− flow-sorted microglia populations. **h**, DEGs and GO analysis between flow-sorted normal brain microglia and GBM-bearing contralateral microglia. **i**, DEGs between microglia clusters 2 and 4 and between clusters 2 and 19. **j**, Monocle3 trajectory inference on microglia clusters. **k**, Reactome analysis of EdU+ and EdU− microglia populations.

gender-matched mice, and performed bulk RNAseq (Fig. 3e,f and Extended Data Fig. 4e,f). As expected, EdU+ GBM microglia contained upregulated genes associated with mitotic cytokinesis, cell division and cell cycle (Fig. 3g). Microglia sensome[22] gene set enrichment analysis (GSEA) validated the identity of microglia (Extended Data Fig. 4g). EdU− microglia express genes associated with inflammatory response, innate immune response, cytokine response, cellular response to interleukin-1 (IL-1) and negative regulation of nuclear factor kappa B (NF-kB) transcription factor activity (Fig. 3g). Regressing out cell cycle gene signatures from the EdU+ microglia dataset did not reveal additional GOs (Extended Data Fig. 4h). Spearman correlation to embryonic (E17), postnatal and adult Tmem119 positive and negative microglia and LPS-stimulated adult microglia show that the EdU+ microglia population is less related to embryonic, postnatal (Tmem119+ and Tmem119−) and adult (LPS-stimulated and control) microglia than EdU− microglia (Fig. 3f), suggesting that cycling microglia in GBM adopt a transcriptome that is different from noncycling (EdU−) microglia, the latter seemingly adopting a proinflammatory polarization.

Comparing the transcriptomes of contralateral GBM microglia with those of normal brain microglia showed that microglia located far from the GBM tumor bed are activated significantly, upregulating genes involved in major histocompatibility complex II (MHCII) antigen processing and presentation, cellular responses to interferon (IFN)-β,γ and innate immune response, whereas the top gene signatures of normal brain microglia related to phagocytosis, and positive regulation of NF-KB transcription (Fig. 3h and Extended Data Fig. 4i). This suggests that the presence of a GBM stimulates distant microglia that are otherwise not in direct physical contact with the tumors. Applying pseudotime tracing to microglia clusters determined that cluster 4 gives rise to cluster 2 (Fig. 3j). Interestingly, the levels of expression of canonical microglia markers P2ry12 and Tmem119 are higher in cells of cluster 4 than cluster 2 (Fig. 3i), reinforcing the notion that cluster 2 is derived from cluster 4. Reactome analysis of EdU+ and EdU− transcriptomes confirms the mitotic nature of EdU+ sorted cells and the antigen processing and presentation, and IL-1 signaling of EdU− cells (Fig. 3k). Together, these findings suggest that a subpopulation of microglia in the GBM tumor microenvironment (TME) respond to the tumor by proliferating and committing less to polarization programs.

**GBM microglia show a proinflammatory transcriptome.** To understand the molecular features driving the formation of microglial clusters 2, 4 and 19, we identified DEGs (Fig. 3i). Cells from cluster 2 had a significant upregulation of genes associated with proinflammatory processes, including Tnf, Il1b, Il1a and Cxcl10 (Fig. 3i). GO analysis of cluster 2 shows enrichment for pathways associated with inflammatory response, LPS-mediated signaling pathway and response to LPS (Tnf, Cxcl2, Cxcl10, Il6, Ccl2, Nlrp3, Cd14, Cebpb, Acod1, Cxcl3, Cxcl13, Cxcl10, Il1b) chemokine-mediated signaling and chemotaxis (Ccl12, Cxcr4, Ccl4, Ccrl2, S100a9, S100a8, Cxcl10, Ccl12, Ccl4, Ccl3, Ccl2, Cxcl13, Cxcl3, Cxcl2) (Extended Data Fig. 4j). High Tnf levels are observed only in cluster 2 microglia with enrichment of other 'M1'-like proinflammatory markers (Il1a, Il1b, Ccl3) (Fig. 3i). On the other hand, microglia cluster 4 expresses genes associated with positive regulation of phagocytosis (Fcgr3, Fcer1g, Il1b, Sirpa, Mertk, Tnf), proinflammatory hallmark activation markers and retains expression of genes involved in chemotaxis suggesting that cells in cluster 4 represent a more actively phagocytic population of microglia than those in cluster 2. Microglia cluster 19 is enriched in genes that are involved in ribosomal small unit assembly and biogenesis (Fig. 3i and Extended Data Fig. 4j), the significance of which is unknown. Together, these data suggest that one subset of microglia respond to the tumor by supporting a proinflammatory program, whereas another subset of microglia lacks this function.

**Most GBM cytokines originate from intratumoral immune cells.** Recruitment of immune cells to tumors has been investigated largely from a cancer cell centric perspective. However, a growing number of studies report on the contribution of TME cells to this process[23]. To address this, we analyzed a panel of 32 cytokines by quantitative PCR with reverse transcription (RTqPCR) from CD45− and CD45+ flow-sorted cells from mouse GBMs (Fig. 4a). We found that expression of most (22/32) cytokines was enriched in CD45+ cells (Fig. 4a) and validated by scRNA-seq results (Fig. 4b). Microglia cluster 2 expresses highest levels of Tnf (Fig. 4b–d and Extended Data Fig. 5a), which was validated independently by flow cytometry (Fig. 4e). Similarly, most cytokines receptors are expressed in CD45+ cell populations (Fig. 4c). Notably, Cxcl12 levels are higher in CD45− endothelial cells (Fig. 4a,b,e) whereas its receptor Cxcr4 is expressed on T cells, NK cells, B cells, neutrophils (including polymorphonucler myeloid-derived suppressor cells (PMN-MDSCs)), macrophages and DCs (Fig. 4b,e) perhaps reflecting an unbeknown paracrine circuitry between endothelial and several immune cells. Together, these results reinforce the developing notion that most intratumoral cytokines and their receptors are expressed more from CD45+ cell populations and much less so from cancer cells.

**Expression of immune checkpoints in CD45+ cells of GBMs.** The common belief that cancer cells are the sole source of checkpoint ligands and responsible for suppression of T cell immune responses is slowly changing as recent reports from us and others demonstrated the importance of myeloid-derived checkpoint molecules[24–27]. Reinforcing this notion are our findings that most checkpoint transcripts are expressed in CD45+ cells, mostly in T and myeloid cells (Fig. 4f). Notably, transcripts for PD-1 (Pdcd1) were observed only in T cell clusters (Fig. 4g and Extended Data Fig. 5b) whereas PD-L1 (Cd274) transcripts were more abundant in neutrophils, DCs and macrophages and PD-L2 (Pdcd1lg2) in DCs and not in EGFR+ cancer cells (Fig. 4f,g and Extended Data Fig. 5b). Transcript levels for VISTA (Vsir) were identified and validated in microglia populations (Fig. 4f and Extended Data Fig. 5c,d), and Ceacam1 and Galactin9 (Lgals9)—both TIM-3 ligands[28,29]—were found elevated in CD45+ cell populations (Fig. 4f and Extended Data Fig. 5e,f). Together, these results indicate that checkpoint receptors are present on innate immune cells in addition to T cells, and their ligands are expressed mostly in noncancer cells.

**Longitudinal evolution of GBM TME during tumor progression.** To uncover cellular and molecular changes to the GBM TME over time, normal brain and early and late stages of GBM development (Extended Data Fig. 6a and Supplementary Table 1) were analyzed. As expected, EGFR+ cells were absent in normal brain and increased dramatically in early and late GBMs (Fig. 5a,b and Supplementary Table 2). Flow cytometry validated the cell cluster compositions of early and late samples, in which early GBMs were composed mostly of microglia and late GBMs saw increased infiltration of macrophages (Fig. 5c,d). Flow cytometry of independent cohorts of GBM mice further validated these observations (Fig. 5e,f). Notably, an initial quasi-stagnant accumulation of EGFR+ cells for the first 25 days was followed by an explosive growth expansion culminating in a moribund late stage preceded by an accumulation of CD45+ cells (Fig. 5e). The observed increases in macrophages, PMN-MDSCs, lymphocytes, monocytes and NK cells over time were also not detected in contralateral brain tissues, indicating that these unique cellular dynamics are localized to the TME (Fig. 5f).

Other than EGFR+ cancer cells, there are noticeable cell clusters that are unique to GBM (Fig. 5a). For instance, oligodendrocyte (9), ciliated cell (33), macrophage (6,20), and neutrophil/PMN-MDSCs (11) are only observed in early and late GBMs (Fig. 5a,

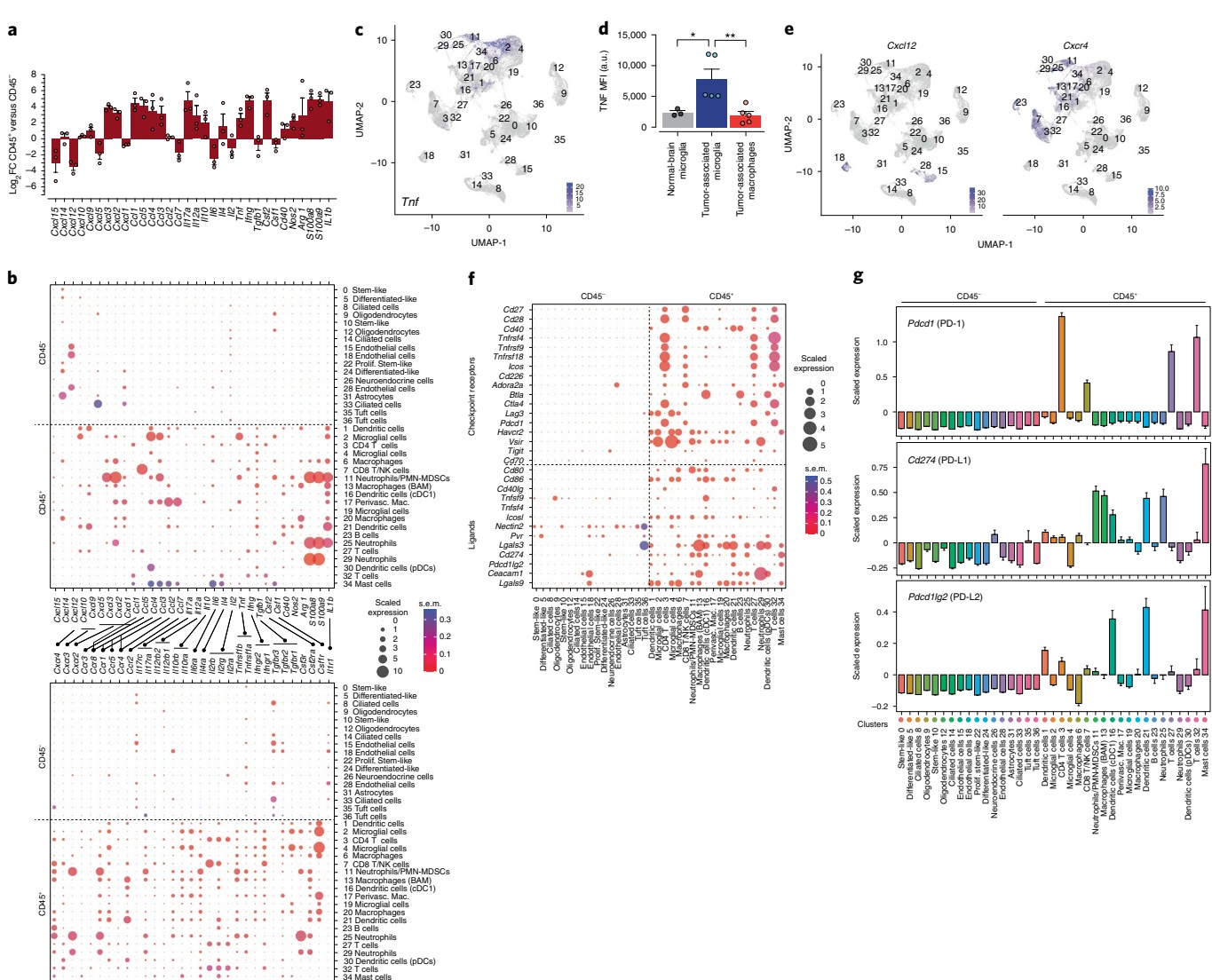

**Fig. 4 | Cytokines, checkpoint receptors and their ligands are expressed mostly in the CD45⁺ compartment. a**, RT-PCR log$_2$FC for the indicated cytokines from flow-sorted CD45⁺ and CD45⁻ cells from GBM. Data are mean ± s.d. of biologically independent tumors $n = 3$. **b**, Scaled expression of the indicated cytokines and their receptors from the CD45⁻ and CD45⁺ scRNA-seq datasets. s.e.m. indicated. The lines between the two panels represent receptor ligand pairs. **c**, Expression of *Tnf* overlaid on UMAP. **d**, TNF mean fluorescence intensity (MFI) expression by flow cytometry of GBM tumors. Data are mean ± s.d. of biologically independent tumors, $n = 3$, 5 and 5 of normal brain microglia, tumor-associated microglia and macrophages, respectively *$P = 0.0472$, **$P = 0.0097$. **e**, Expression of *Cxcl12* and *Cxcr4* transcripts overlaid on UMAPs. **f**, Scaled expression for the indicated checkpoint receptors and ligands from the CD45⁻ and CD45⁺ scRNA-seq datasets. **g**, Scaled expression of *Cd274* (PD-L1), *Pdcd1lg2* (PD-L2) and *Pdcd1* (PD-1) transcripts. Data are mean ± s.e.m.

Supplementary Table 2). Oligodendrocyte cluster 9 only appears in GBM samples, suggesting GBM-stimulated production and/or differentiation of brain-resident oligodendrocytes. Monocle3 pseudotime tracing analysis revealed that cluster 9 cells derive from cluster 12 (Fig. 5g). Furthermore, we found that cells from cluster 12 are enriched in genes involved in oligodendrocyte myelination (*Mag, Gjc3, Gal3st1, Ugt8a, Pllp, Lpar1, Mal, Qk, Tspan2, Mbp, Jam3, Nkx6-2, Omg, Cntn2, Sox10, Aspa*) and differentiation and development (*Nkx6-2, Cnp, Olig1, Sox8, Tspan2, Tppp, Sox10, Gstp1, Hdac11, Lpar1*)) (Fig. 5h and Extended Data Fig. 6b). Cells from cluster 9 preferentially expressed genes known to function in aerobic respiration (*Mt-Nd4, Mt-Co1, Mt-Co3* and *Mt-Nd1*), response to hyperoxia (*Mt-Atp6, Cdkn1a, Gm10925*), regulation of ATP biosynthesis (*Tmsb4x, Mt-Co2, Bcl2l1*) cellular response to insulin stimulus (*Errfi1Irs2, Srsf5,Sgk1*) and regulation of cell cycle (*Cdkn1a, Gadd45b, Srsf5, Sgk1*) (Fig. 5h and Extended Data Fig. 6b),

suggesting a change in energy requirement for these newly derived oligodendrocytes.

**Infiltration of macrophages, and PMN-MDSCs parallels blood–brain barrier leakage.** The significant increases in macrophages (6,20), PMN-MDSCs (11) and EGFR⁺ cells late during GBM progression may reflect a disruption of the blood–brain barrier (BBB). Intravenous injection of Evans Blue (EB) and sodium fluorescein (NaF) to assess BBB integrity[30] at 10, 20, 25 and 30+ (moribund) days after tumor initiation led to extravasation and rapid accumulation after 25 days (Fig. 5i). Orthogonal validation by contrast-enhanced molecular resonance (MR) imaging over time normalized to contralateral central nervous system, showed increased enhancement between 26 and 32 days post-tumor initiation, corresponding to extensive BBB leakage (Fig. 5j). Together, these results demonstrate that the integrity of the BBB remains intact until later stages

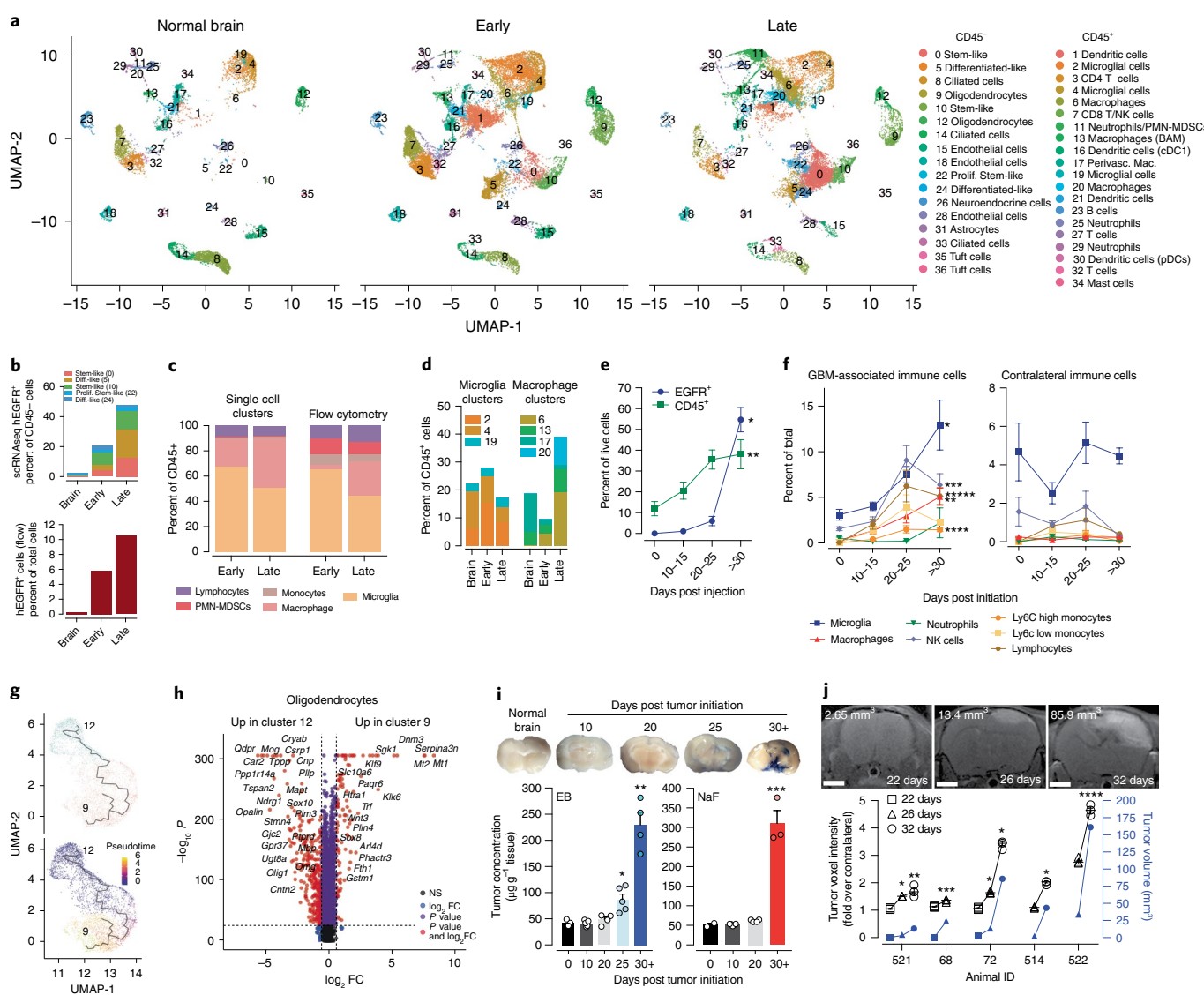

**Fig. 5 | Dynamics of immune cell infiltration over time. a**, UMAPs of normal brain, Early and Late stage GBMs. **b,c**, Relative number of EGFR⁺ (**b**) and the indicated cell types (**c**) from scRNA-seq data and flow cytometry of the GBM samples used for scRNA-seq. **d**, Cluster composition of microglia and macrophages from scRNA-seq. **e**, Flow cytometry of EGFR⁺ and CD45⁺ cells from GBMs. Data are mean ± s.e.m. of biologically independent replicates. Control brain day 0 $n = 3$, GBM $n = 6$, 6 and 3 for days 10–15, 20–25 and >30, respectively. At >30 days relative to day 0, *$P = 0.0001$, **$P = 0.0285$, unpaired $t$ test two-tailed. **f**, Longitudinal flow cytometry of the indicated GBM cell types. Data are mean ± s.e.m. of biologically independent replicates. Left panel, control brain day 0 $n = 3$, GBM $n = 8$, 8 and 9 for days 10–15, 20–25 and >30, respectively, *$P = 0.0122$, **$P = 0.0236$, ***$P = 0.0488$, ****$P = 0.0006$, *****$P = 0.0079$, unpaired $t$ test two-tailed. Right panel, day 0 $n = 3$, GBM $n = 7$, 5 and 3 for days 10–15, 20–25 and >30, respectively. At >30 days relative to day 0, all $P > 0.05$ and considered not significant, unpaired $t$ test two-tailed. **g,h**, Pseudotime trajectory (seed cluster 12) (**g**) and DEGs (**h**) of oligodendrocyte clusters 9 and 12. **i,j**, Assessment of GBM BBB integrity. Representative photomicrographs of GBM brains post EB and NaF administration and quantification (**i**). Data are mean ± s.e.m. of biologically independent replicates. EB, $n = 3$, 5, 4, 5 and 4, for days 0, 10, 20, 25 and 30+, respectively and NaF, $n = 2$, 5, 4, and 3 for days 0, 10, 20, and 30+, respectively. *$P = 0.0209$, **$P = 0.0012$, ***$P = 0.0089$ when compared with day 0, unpaired $t$ test two-tailed. Representative gadolinium-enhanced T1-weighted MRIs and quantification of a developing GBM imaged at 22, 26 and 32 days post-tumor initiation (**j**). Tumor volumes (in mm³) were measured using T2 weighted images. Scale bar, 2 mm; data represent mean ± s.d. of technical replicates. *$P < 0.0001$, **$P = 0.0071$, ***$P = 0.0029$, ****$P = 0.0002$ when compared with 22-day voxel intensities, unpaired $t$ test two-tailed.

of tumor development—a time that coincides with the influx of immunosuppressive cells into GBM.

**Changes in innate immunity during GBM progression.** Leveraging the longitudinal attributes of our model, we observed that microglia cluster 2 gained expression of proinflammatory markers (*Cxcl2, Cxcl3, Cxcl10, Il1b, Tnf, Ccl3*), neutrophil chemotaxia (*Cxcl10, Il1b, Ccl4, Ccl3, Cxcl3, Cxcl2, S100a9*) and positive response to phagocytosis (*Il1b, Gata2, Tnf*) in early GBM when compared with normal brain (Fig. 6a and Extended Data Fig. 7a), suggesting that early GBMs activate an inflammatory response in microglia (2). Further progression to late GBM saw increases in expression of *Cxcl13* and immediate-early genes (IEGs) (*Fos, Jun, Junb, Egr1, Zpf36, Nfkbia, Dusp1*) (Fig. 6a). Early GBM microglia (4) demonstrated gain in expression of complement genes (*C1qb, C1qa, C1qc*) and lipoprotein catabolic process (*Apoe, Ctsd*) (Fig. 6a) and loss of gene expression involved in mitochondrial electron transport, ATP-synthesis-coupled electron transport and aerobic respiration

(*mt-Nd1/2/4/5/6, mt-Co1/2*) during early to late GBM progression (Fig. 6a), suggestive of changes in energy use in microglia (4) during GBM progression. Early GBMs saw resident BAM (13,17) gained expression of genes in inflammatory response (Extended Data Fig. 7b), reinforcing the notion that, during early GBM growth, innate immunity cells adopt a proinflammatory status. Upon progression, BAM (13) gained expression of genes involved in neutrophil chemotaxis, positive regulation of angiogenesis and inflammatory response (Extended Data Fig. 7b).

**Recruitment of immunosuppressive cells during progression.** To better define the molecular mechanisms by which immune cells are recruited to the GBM TME, we determined DEGs between late and early GBMs. In macrophage (6, 20), gains were observed in expression of genes involved in negative regulation of inflammatory response, neutrophil chemotaxis and aggregation (6) and angiogenesis and metabolic process (20) (Fig. 6b and Extended Data Fig. 7c). In DC (1), gains in genes modulating antigen processing and presentation via MHCII and leukocyte chemotaxis were observed (Fig. 6b and Extended Data Fig. 7d). In PMN-MDSCs (11), upregulation of genes in NFAT signaling and cellular response to hypoxia and downregulation of genes that characterize inflammatory responses and regulation of T cell proliferation were observed (Fig. 6b and Extended Data Fig. 7e). We observed decreases in transcripts of the cytokine:receptor *Ccl2* and *Ccr2* in macrophage (17) and (13), respectively (Fig. 6c), suggesting a cytokine relationship between perivascular macrophages and other BAMs in normal brain. PMN-MDSCs (11) expressed more *Cxcl2* and less *Cxcl3* in early than late GBM, and expression of their receptor *Cxcr2* is highest in neutrophil (25) (Fig. 6c), perhaps reflecting a switch in *Cxcr2* ligand use during GBM progression. The high expression of *Il1b* in PMN-MDSCs (11) and in neutrophil (25) (Fig. 6c) and its receptor on endothelial cells (18,28) (Fig. 4c), also suggest a functional interaction between granulocytes and endothelial cells in GBM.

GSEA of microglia and macrophage clusters for markers of classical 'M1-like' proinflammatory polarization and 'M2-like' protumorigenic polarization[31] reveal that microglia (2) had similar normalized enrichment scores (NES) in normal brain and early and late GBM samples whereas microglia (4) preferentially displayed protumorigenic NES in early and late GBMs (Fig. 6d). Infiltrating macrophages (6,20) had higher NES for 'M2-like' protumorigenic polarization markers in early and late GBMs compared with clusters (13,17) (Fig. 6d). Taken together, this suggests that microglia and BAM adopt a proinflammatory immune microenvironment early in tumor growth, which is lost during GBM progression.

**Bone marrow changes early during gliomagenesis.** Bone marrow (BM) myeloid progenitors generate MDSCs during tumor-driven emergency myelopoiesis[24,32]. Using flow cytometry (Extended Data Fig. 8a,b), we show that GBM significantly increased the hematopoietic progenitors LSK and LK, with the granulocyte/monocyte progenitors (GMPs) in the LK population showing the biggest increase (Fig. 7a). GBM mice had splenic myeloid cells increases—evidence for GBM-induced emergency myelopoiesis. In these cells, PMN-MDSC, M-MDSC and DC were increased, whereas macrophages were diminished compared with control mice (Fig. 7b). GBM mice had a significant increase in systemic T cells, characterized by elevated expression of CTLA-4 (Fig. 7a,b). These results

indicate that GBM induces robust emergency hematopoiesis with significant systemic immunosuppressive implications.

**TMZ depletes BM-derived GMP.** To glean insights into the effects of SOC on the GBM immune microenvironment, we analyzed immune cells from brain, BM and spleen of TMZ-treated mice by flow cytometry. A dose of 25 mg kg$^{-1}$ daily of TMZ in mice is equivalent to the IR/TMZ SOC dosage (75 mg m$^{-2}$) given to GBM patients, whereas a dose of 66.67 mg kg$^{-1}$ daily in mice is equivalent to 150–200 mg m$^{-2}$ postradiation adjuvant therapy[33]. After a 2-week treatment of TMZ, tissues were harvested at 24, 72 or 168 h (Extended Data Fig. 8c) and analyzed by flow cytometry (Extended Data Fig. 8d,e). Microglia numbers were unchanged; however, BAMs showed a significant decrease (control 2.75 ± 0.38 versus treated 1.35 ± 0.25) in the 66.67 mg kg$^{-1}$ and a trend in the 25 mg kg$^{-1}$ treated cohorts 24 h after cessation of TMZ (Extended Data Fig. 8e), effects that were not observed at later times, suggesting a temporary sensitivity of BAMs to TMZ. Dramatic depletion of GMPs in low and high doses of TMZ were observed in BM, which was more striking at 24 h, and did not recover fully until 1-week post cessation of treatments (Extended Data Fig. 8e). In contrast to GMP, the levels of common myeloid progenitors (CMP) remained unchanged, consistent with the hypothesis that more rapidly proliferating/differentiating myeloid progenitors are more sensitive to TMZ. Splenic myeloid cells were unaffected by TMZ (Extended Data Fig. 7e). Note that splenic myelotoxicities are likely to occur at later time points based on the kinetics of nadir development in patients treated with TMZ[34]. Together, these results demonstrate that treatment of healthy, nontumor mice with TMZ has little effect on population levels of microglia, BAMs and CMPs, whereas GMPs are sensitive to prolonged TMZ treatment, requiring several days for the myelotoxicity to dissipate.

**TMZ effects on PMN-MDSCs, macrophages and microglia.** Treatment of GBM mice with 66.67 mg kg$^{-1}$ of TMZ (Extended Data Fig. 8f) extended survival by 14 days compared with control (median survivals (days): control 9, TMZ-25 9.5 and TMZ-66.7 23) (Fig. 7c) and decreased BLI output (Extended Data Fig. 8g), whereas low dose (25 mg kg$^{-1}$) TMZ had no effect on survival and GBM growth (Fig. 7c and Extended Data Fig. 8g), analogous to different glioma models[35]. Flow cytometry analysis revealed that high dose (66.7 mg kg$^{-1}$) TMZ significantly increased microglia in GBM mice but not control animals and resulted in a decrease in macrophages, whereas decreases in PMN-MDSC populations were observed with both TMZ concentrations (Fig. 7d). No statistically significant differences were observed in the relative numbers of CD45$^+$ cells and in splenic MDSCs upon TMZ treatments (Fig. 7d and Extended Data Fig. 8h), suggesting that decreases in tumoral PMN-MDSCs upon TMZ treatment may stem from recruitment deficiencies to the TME. Our data also imply that high dose TMZ may promote a proinflammatory phenotype by increasing microglia and decreasing suppressive populations of PMN-MDSCs and macrophages in addition to direct anti-cancer cell effects. This TMZ-mediated switch in the immune microenvironment was not sufficient, however, to induce tumor clearance, suggesting a lack of adaptive response and development of resistance mechanisms.

**IR/TMZ therapy promotes effector CD8 T cells and survival.** To determine the effect of SOC therapy on the immune microenvironment of GBM, mice were treated for 2 weeks, analyzed by

**Fig. 6 | Loss of macrophage and microglia proinflammatory polarization over time. a**, Volcano plots of DEGs of microglia clusters 2 and 4 between Early versus normal and Late versus Early GBM tumors. **b**, Volcano plots of DEGs of DC cluster 1, Neutrophils/PMN-MDSCs cluster 11 and macrophage clusters 6 and 20 in Late versus Early GBM tumors. **c**, Cytokine ligand:receptor pairs analysis. Expression levels of the indicated ligands and receptors for the indicated cell type clusters in normal, Early and Late GBM samples are plotted. **d**, NES from GSEA of the indicated microglia and macrophage clusters from normal, Early and Late GBM samples analyzed for proinflammatory 'M1-like' markers and anti-inflammatory, protumorigenic alternatively polarized 'M2-like' markers.

flow cytometry and monitored to establish survival (Extended Data Fig. 8i). TMZ treatments of 25 mg kg⁻¹ daily failed to extend survival; however, fractionated daily radiotherapy (10 × 2 Gy), either alone or in combination with TMZ, significantly extended survival (median survival (days); control 9, TMZ-25 9.5, IR 43 and IR/TMZ 45.5) (Fig. 7e). IR alone and IR/TMZ increased the number of

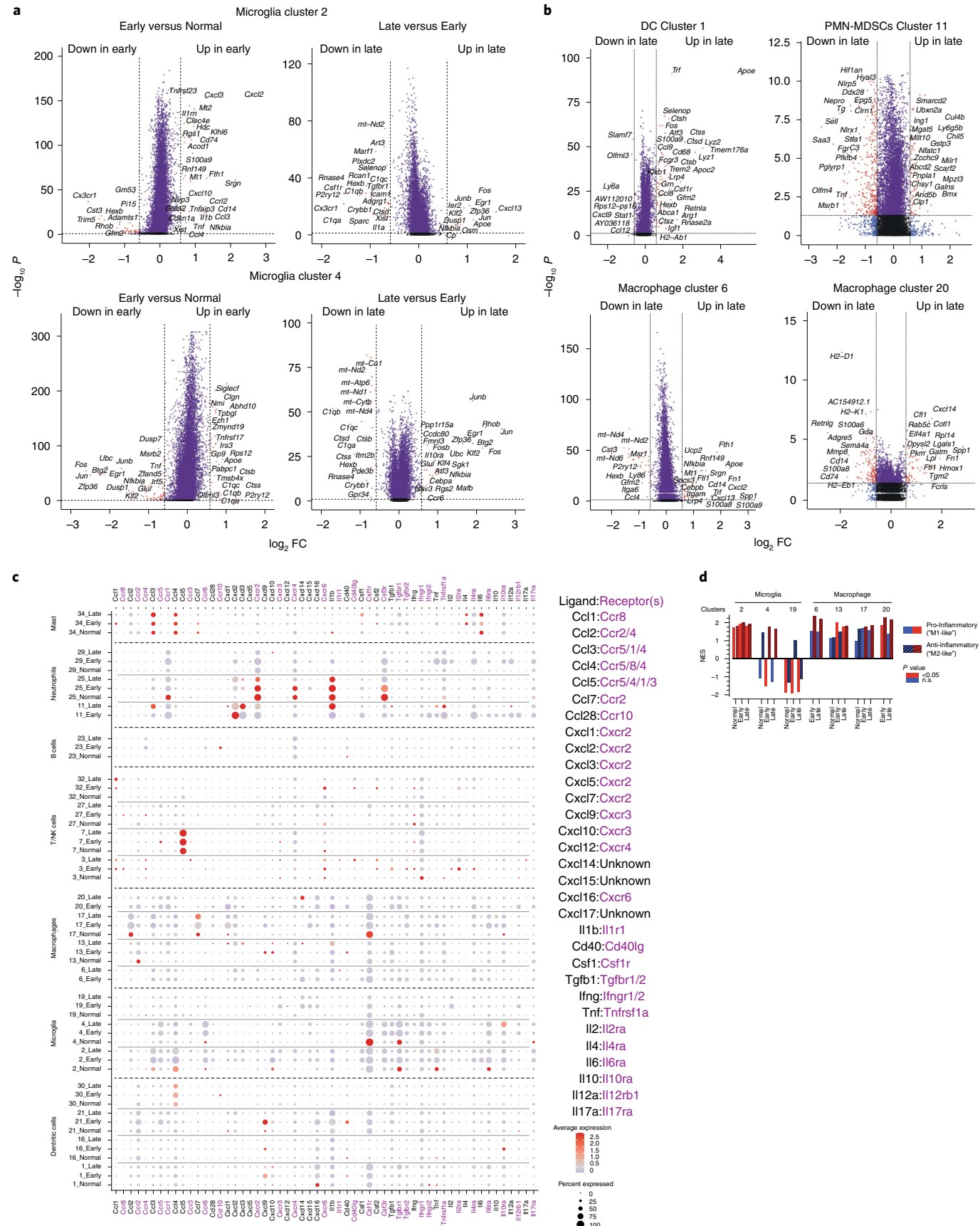

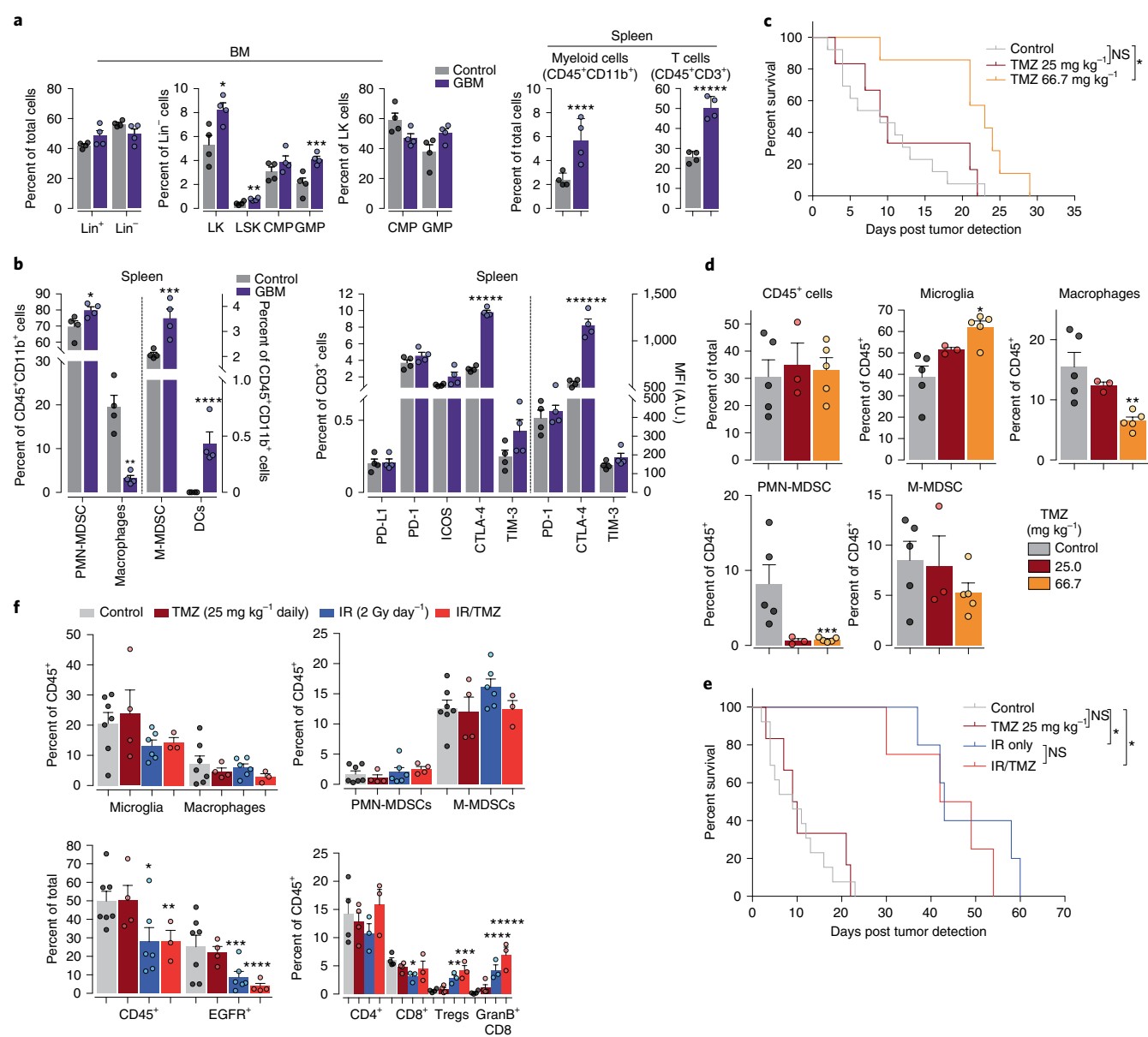

**Fig. 7 | TMZ and radiation change GBM immune microenvironment and prolong survival. a**, Flow cytometry of BM and spleens of control non-GBM-bearing mice and early-stage (BLI $10^7$ p s$^{-1}$ cm$^{-2}$ sr$^{-1}$) GBM mice for the indicated cell type and progenitors. Data are represented as mean ± s.e.m. of biologically independent replicates. Comparisons between control ($n=4$) and GBM ($n=4$) mice. *$P=0.0266$, **$P=0.015$, ***$P=0.0081$, ****$P=0.0139$ and *****$P=0.0002$, unpaired $t$ test, two-tailed. **b**, Flow cytometry of spleen cells as in **a** for the indicated cell markers and checkpoint molecules. Data are represented as mean ± s.e.m. of biologically independent replicates. Comparisons between control ($n=4$) and GBM-bearing ($n=4$) mice, *$P=0.0491$, **$P=0.0094$, ***$P=0.0087$, ****$P=0.0053$, *****$P=0.0001$ and ******$P<0.0001$, unpaired $t$ test, two-tailed. **c**, Kaplan–Meier survival analysis of GBM mice treated with control vehicle or 25 or 66.7 mg kg$^{-1}$ TMZ daily. TMZ 66.7 g kg$^{-1}$ versus control *$P<0.0001$, NS, not significant, log-rank (Mantel–Cox). Data are represented from biologically independent replicates, $n=13$, 6 and 7 for control, 25 and 66.7 mg kg$^{-1}$ TMZ, respectively. **d**, Flow cytometry for the indicated cell types from GBM of control and TMZ-treated (25 mg kg$^{-1}$ or 66.7 mg kg$^{-1}$ daily) mice. Comparisons of control and TMZ-treated. Data are represented as mean ± s.e.m. of biologically independent replicates, $n=5$, 3, and 5 for control, 25.0 and 66.7 mg kg$^{-1}$ TMZ respectively, *$P=0.0051$, **$P=0.0074$, ***$P=0.0194$, unpaired $t$ test, two-tailed. **e**, Kaplan–Meier survival analysis of GBM mice treated with control vehicle, 25 mg kg$^{-1}$ TMZ, IR and IR/TMZ as indicated. Data represent biologically independent replicates, $n=13$, 6, 4 and 5 for control, 25 mg kg$^{-1}$ TMZ, IR and IR/TMZ, respectively. *$P<0.0001$ log-rank (Mantel–Cox). **f**, Flow cytometry of GBMs for the indicated cell type from control and TMZ (25 mg kg$^{-1}$), IR (2 Gy/day) and IR/TMZ-treated GBM mice. Comparisons of control and treatment. Data represent mean ± s.e.m. of biologically independent tumors. Microglia/macrophages, PMN-MDSCs/M-MDSCs and CD45$^+$/EGFR$^+$ panels $n=7$, 4, 6, 3 for control, TMZ, IR and IR/TMZ, respectively. *$P=0.0397$, **$P=0.0456$, ***$P=0.0487$, ****$P=0.034$, unpaired $t$ test, two-tailed. CD4$^+$/CD8$^+$/Tregs/GranB$^+$CD8$^+$ panel $n=4$, 4, 3, 3, for control, TMZ, IR and IR/TMZ, respectively of the *$P=0.0125$, **$P=0.0065$, ***$P=0.0049$, ****$P=0.0054$, *****$P=0.0026$. Unpaired $t$ test, two-tailed.

GranzymeB$^+$ CD8$^+$ T cells (Fig. 7f), reduced the number of EGFR$^+$ cancer cells and CD45$^+$ cells (Fig. 7f) and decreased BLI output (Extended Data Fig. 8j). These results reflect the notion that IR

induces cancer cell death while accumulation of GranzymeB$^+$ CD8$^+$ T cells are likely engaged by radiation-induced neo-epitopes, perhaps being responsible for the prolongation in survival. However,

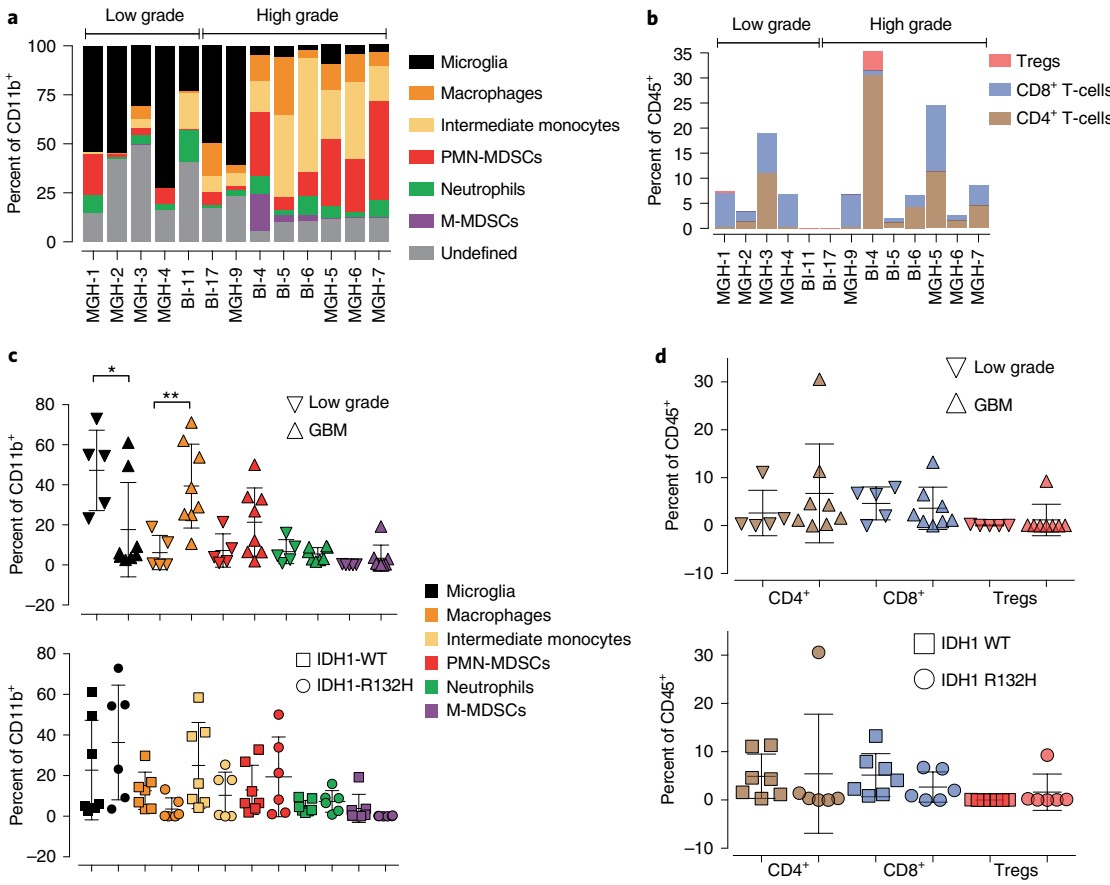

**Fig. 8 | Patient immune heterogeneity in low-grade gliomas and GBMs. a**, Flow cytometry from low-grade and GBM patient tumor samples analyzed for the indicated cell types. **b–d**, Flow cytometry of tumors in **a** for Tregs, CD4+ and CD8+ lymphocytes (**b**), the indicated cell types as percent of CD11b+ population in low-grade and GBM tumors and IDH1 R132H mutant and IDH1 wild-type gliomas (**c**); note that in low-grade and GBM panel, monocytes and macrophages were combined and for Tregs, CD4+ and CD8+ lymphocytes in low-grade, GBM, IDH1 R132H mutant and IDH1 wild-type gliomas (**d**). Data are presented as biologically independent tumours. $n = 5$ and 8 low-grade and GBM, respectively. *$P = 0.04$, **$P = 0.007$ unpaired $t$ test, two-tailed.

we noted that these beneficial effects were accompanied by an increase in Tregs (Fig. 7f), potentially providing an explanation for the transient therapeutic benefit provided by SOC therapy.

**Patient immune heterogeneity of low and high-grade gliomas.**
GBM patients are diagnosed with symptomatic late-stage disease due to the asymptomatic nature of early-stage GBMs, which considerably hampers longitudinal studies of evolution of the immunological landscape of GBM. To overcome this limitation, and to correlate immune composition with disease aggressiveness, we analyzed freshly isolated low-grade glioma ($n = 5$) and GBM ($n = 8$) patient (seven males and six females) tumor samples by flow cytometry (Extended Data Fig. 9a and Supplementary Table 3 for patient demographics). Relative amounts of CD45+ and CD11b+ cells were similar in low-grade and GBM tumors and the median ages at diagnosis were $44 \pm 4.9$ years old versus $54.3 \pm 4.1$ years old for low-grade and GBM, respectively (Extended Data Fig. 9b,c). Median survival of GBM patients was 73.8 weeks and undefined for the low-grade patients (Extended Data Fig. 9d). Flow cytometry revealed that GBM immune profiles are distinct from low-grade gliomas. Notably, higher numbers of microglia are observed in low-grade gliomas than in GBMs, the latter having higher numbers of PMN-MDSCs, intermediate monocytes and macrophages (Fig. 8a–d). Neither gender, anatomical location nor IDH1 mutant status correlated with these differences (Fig. 8c,d and Extended Data Fig. 9e-j).

**Discussion**
GBM is an aggressive and universally fatal primary brain cancer. Clinical progress remains restricted because we lack comprehensive knowledge of the evolution of GBM tumor and immune cells during treatments. Such information is unattainable using patient GBM samples due to the inherent difficulties in performing longitudinal surgical samplings. To overcome these roadblocks, we performed scRNA-seq on a genetically engineered EGFR-driven mouse GBM model and flow cytometric analyses of mouse and patient GBM samples to determine their immunological cellular landscape at various stages of disease evolution and during SOC therapies.

The most striking observation of our studies was the changes in immune cell composition as GBM developed. We uncovered an immune microenvironment progression from 'M1'-like pro-inflammatory microglia early during GBM tumorigenesis and in low-grade glioma tumors from patients, towards an 'M2'-like pro-tumorigenic macrophage-centric infiltration in the advanced stages of GBM in the mouse model and in patient samples. These transitions parallel a breakdown of the BBB and an explosive growth of EGFR+ cancer cells.

MDSCs are immature myeloid cells with immunosuppressive and tumor-promoting properties that accumulate in tumors[36]. MDSCs are associated with poor outcomes and correlate negatively with response to chemotherapy, immunotherapy and cancer vaccines[37–40]. In our studies, we observed that PMN-MDSCs expressed high levels of *Csfr3*, *Ccr1*, *Cxcr2* and *Cxcr4*, whereas

*Cxcl12*, the ligand for *Cxcr4*, was expressed preferentially in CD45⁻ GBM endothelial cells, unraveling a previously unidentified mechanism for recruitment of immunosuppressive MDSC into the GBM microenvironment. Thus, GBM progression might not simply reflect the development of cancer cell resistance and escape from chemoradiotherapy, but might be the direct consequence of immunosuppressive mechanisms induced by an accumulation of MDSCs.

Another key finding from our work is the identification of the immune cell types expressing many cytokines and their receptors in GBM. In our model, the bulk of cytokines and their receptors are expressed in CD45+ cells with few from EGFR+ cancer cells. We also uncovered a potential circuitry between *Cxcl12* positive endothelial cells and *Cxcr4* positive PMN-MDSCs, perhaps facilitating MDSC recruitment and infiltration. Although *Cxcl12* and *Cxcr4* expression has been reported previously in GBM[41–45], our results of *Cxcr4* positive intratumoral PMN-MDSCs reveal a potential axis for therapeutic intervention, in particular in the context of checkpoint inhibition[46–48].

Finally, an important advancement is the discovery of a highly proliferating population of GBM-associated microglia, which may play a decisive role in activating emergency myelopoiesis in GBM patients and recruit BM-derived immunosuppressive myeloid cells to the GBM TME. More in-depth molecular analyses of this population are required. Collectively, our results offer a unique view of the dynamics of immune and nonimmune cells during GBM initiation and progression and point to new areas for therapeutics development.

## Online content

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

## Methods

**Study design.** These were prospective studies in genetically engineered mouse models and GBM patient samples designed to study the longitudinal changes in the cellular composition and transcriptomes of the immune microenvironment of mouse GBM. For the single cell RNA-seq study, normal brain controls ($n=2$ samples, each pooled from two mice), Early GBM samples ($n=4$ samples each pooled from two independent mice or tumors) and Late GBM samples ($n=3$, single tumor per mouse) (see Supplementary Table 1). Patient samples were obtained postsurgery and completely deidentified. For flow cytometry analyses, cohorts of mice were initiated and GBM growth parameters monitored using BLI.

**Genetically engineered mouse models.** All mouse procedures were carried out in accordance with Beth Israel Deaconess Medical Center recommendations for care and use of animals and were maintained and handled under protocols approved by Institutional Animal Care and Use Committee (IACUC). EGFR conditional transgenic mice (*Mus musculus musculus*) were generated as described[2–4] and crossed with Cdkn2A[−/−49], PTEN flox/flox[50] and conditional luciferase reporter transgenic mice[8]. Animals are kept on a mixed genetic background composed of C57Bl/6J, 129S4, FVBN/J, SJL and Balb/cJ. Brainbow 2.1 mice[15] were obtained from Jackson Laboratory and crossed to EGFR conditional transgenic mice. A 1:1 ratio of 12- to 16-week-old males and females were used in all experiments. Mice were housed using a ventilated cages system on a 12-h light/12-h dark cycle at an ambient temperature of around 25 °C with 40–60% humidity.

**Human samples.** All human GBM samples were collected according to procedures approved by institutional review board protocol nos. 2017P001581 (MGH) and 2017P000635 (BIDMC). All patients signed written consent forms and were not compensated. Demographics of study patients can be found in Supplementary Table 3. Freshly excised patient GBM samples were processed as indicated below for mouse tissues to isolate single cell suspensions and processed for flow cytometry as described below. Information on the antibodies used for flow cytometry can be found in Supplementary Tables 4 and 5. Antibody combinations against the following target antigens were used to define these cell types: PMN-MDSC, neutrophil, granulocyte, CD45$^+$; CD11b$^+$; Ly6G$^+$; Ly6C$^+$; M-MDSC, monocyte, CD45$^+$; CD11b$^+$; Ly6G$^-$; Ly6C$^+$; tumor-associated macrophages, CD45$^+$; CD11b$^+$; Ly6G$^-$; Ly6C$^-$; CD4 T-cell, CD45$^+$; CD3$^+$; CD4$^+$; CD8 T-cell, CD45$^+$; CD3$^+$; CD8$^+$; Treg, CD45$^+$; CD3$^+$; CD4$^+$; Foxp3$^+$.

**Virus production and protocol.** The pTyf TGFa-IRES-iCre and pTyf -iCre lentivirus plasmids were produced at high titer for intracranial injection by transient transfections in HEK293T (ATCC CRL-3216) cells as follows: HEK293T cells were seeded in $4 \times 15\,cm^2$ culture plates at a density of 40–50% confluency and maintained in DMEM supplemented with 10% FBS and antibiotics. For each 15-cm$^2$ culture plate, lentivirus was produced by cotransfection of 20 μg lentiviral transducing vector, 15 μg of ΔR8.9 packaging vector and 10 μg of VSV-G envelope vector using TransIT-LT1 transfection reagent (Mirus) according to the manufacturer's instructions. After 72 h post-transfection, the virus-containing medium was collected and pooled. Conditioned medium was concentrated using 40 μm Amicon filters (Millipore). The viral stock was further concentrated by ultracentrifugation at 100,000*g* for 1.5 h. The concentrated virus was resuspended in 0.1 ml PBS, aliquoted and stored at −80 °C. To determine the functional titer, a dilution series of pTyf TGFa-IRES-iCre high titer viral preparations were used to infect a Tdt-tomato expressing cell line using a final concentration of 8 μg ml$^{-1}$ of polybrene. At 48 h postinfection, cells were harvested, washed with PBS and analyzed via flow cytometry. A dilution series of commercially available adenoviral-Cre was used as a positive control for titration.

**Stereotactic injections.** We performed stereotactic injections on adult animals (6–12 weeks of age). Mice of the indicated genotype were anesthetized with an intraperitoneal injection of Ketamine/Xylazine (ketamine 100–125 mg kg$^{-1}$, xylazine 10–12.5 mg kg$^{-1}$) and mounted on a stereotaxic frame with nonpuncturing ear bars. The incision site was shaved and sterilized with betadine and ethanol surgical scrubs and a single incision was made from the anterior pole of the skull to the posterior ridge. A 1-mm burr hole was drilled at the stereotactically defined location of the striatum (1 mm rostral to the bregma, 2 mm lateral to the midline and at 2.5 mm depth to the pia surface) and a 1 μl Hamilton syringe mounted onto a Stoelting QSI (Stoelting) was used to inject the indicated Cre virus at a rate of 0.1 μl min$^{-1}$. Following retraction of the syringe or pipette, the burr hole was filled with sterile bone wax, the skin was drawn up and sutured and the animal placed in a cage with a padded bottom atop a surgical heat pad until ambulatory.

**Bioluminescent imaging.** Mice were imaged by BLI starting at 18 days postinjection for tumor detection. Isoflurane-anesthetized mice were injected intraperitoneally with 150 mg kg$^{-1}$ D-luciferin and imaged after 10 min using a Xenogen BLI set up. A region of interest (ROI) was drawn around the head and quantitated on total luminescence (photons per second per cm$^2$ per seridan (p cm$^{-2}$ s$^{-1}$ sr$^{-1}$)). Mice were monitored biweekly until tumor development, or weekly after tumor development.

**IR/TMZ treatments.** Mice were imaged until luminescence reached $10^7$ p cm$^{-2}$ s$^{-1}$ sr$^{-1}$ and enrolled in treatment groups. TMZ was resuspended at 2.5 mg ml$^{-1}$ or 66.67 mg ml$^{-1}$ in OraPlus oral suspension medium and mice were dosed via oral gavage at 10 μl g$^{-1}$ for a final dosage of 25 mg kg$^{-1}$ or 66.67 mg kg$^{-1}$, respectively. For survival studies mice were dosed daily until moribund, for timepoint studies mice were dosed for 1 week and harvested at 72 h following the final treatment.

For irradiation, mice were anesthetized with ketamine/xylazine (ketamine 50–100 mg kg$^{-1}$, xylazine 5–6 mg kg$^{-1}$) to immobilize them in the apparatus. Mice were placed in the X-RAD x-ray irradiator with only their head in the field of irradiation. Mice were protected from stray radiation to the body with a sheet of lead formed loosely around them. Following radiation treatment, mice were placed on a heated pad and observed until ambulatory. Treatment regime for radiation is 2 Gy for 5 days, 2 days off, followed by 2 Gy for a further 5 days. Mice for moribund harvest are monitored until moribund, timepoint mice are harvested 72 h following the final treatment.

**Brainbow immunofluorescence.** GBM tumor-bearing mice were perfused with PBS to flush out circulating blood cells, and then perfused with freshly made 4% paraformaldehyde. Brains were harvested and postfixed in 4% paraformaldehyde at 4 °C overnight. Brains were then transferred to 20% sucrose at 4 °C overnight for cryoprotection, followed by 30% sucrose. Sections (50 μm) cut using a microtome were mounted and rehydrated in PBST (PBS 1% Tween20) for 10 min before slides were mounted using ProLong Gold Anti-fade mountant with 4,6-diamidino-2-phenylindole and cured overnight. Slides were imaged on Zeiss 880 upright confocal microscope and analyzed using Fiji software. The fluorescence intensity threshold was set for each image to identify signal above background fluorescence. A pixel count of masks for each color was quantitated using Image J and normalized to total fluorescent pixels per image. A minimum of four fields of view per section was quantitated and a minimum of four sections per tumor were analyzed.

**Single-cell tumor harvesting.** Tumors were dissociated as described above, and a small aliquot was taken for flow cytometry analysis and stained for innate and adaptive immune cell markers. The remaining sample was stained with CD45 and Zombie Yellow fixable viability dye. Cells were washed with RNase-free PBS with 0.1% BSA and immediately sorted on a fluorescence-activated cell sorting (FACS) Aria Cell Sorter for viable CD45$^+$ and CD45$^-$ populations into ice-cold RNase-free PBS with 0.1% BSA. A minimum of 1 million CD45$^+$ and CD45$^-$ cells were harvested when possible, to account for the low capture rate of InDrop system. If fewer cells were available, then two samples were pooled; this occurred for both normal brain samples and for each of the Early samples. Cells were then filtered through a 70 μm filter and brought to the Harvard Medical School Single Cell Core (https://singlecellcore.hms.harvard.edu) where they were run on an InDrop system with the goal of encapsulating 10,000 cells per sample. Following encapsulation, we performed a reverse transcription reaction and prepared libraries[51]. As per core recommendation, four libraries per 10,000 cells were prepared. Following harvest of all samples, libraries were sequenced using the NextSeq and Novaseq systems with 16 and 32 libraries per sequencing run, respectively, so that the depth of sequencing was calculated to be around 25,000 genes per cell.

**Brain tumor flow cytometry.** Mice were perfused with 10 ml PBS, the brain was harvested and cerebellum removed. Brain tissue was minced and resuspended in 1.5 mg ml$^{-1}$ collagenase in Hanks' Balanced Salt Solution (HBSS) with calcium and magnesium. Mixture was incubated rotating at 37 °C for 30 min, with gentle dissociation using a P1000 pipette halfway through to break apart large pieces. Dissociated cell mixture was filtered through a 100 μm filter and diluted with HBSS. All washes were pelleted at 400*g* for 5 min. The mixture was resuspended in 30% Percoll in PBS for myelin removal at 700*g* for 15 min with the brake on low. The myelin layer was removed and the mixture diluted and pelleted at 400*g* for 5 min. Red blood cells were lysed using Biolegend red blood cell (RBC) lysis buffer according to the manufacturer's protocol. Cells were blocked with Fc block for 5 min and stained with antibodies and Biolegend Zombie fixable viability dye. Cell surface antigens were stained 30 min at 4 °C in the dark. Cells were washed twice and fixed using eBioscience FoxP3 intracellular staining kit according to the manufacturer's protocol. Cells were then stained with intracellular antibodies for 30 min at 4 °C in the dark. Cell suspensions were washed twice and resuspended in PBS for analysis. A minimum of 20,000 events were collected on a Beckman Coulter Gallios flow cytometer with acquisition software Kaluza (v.1.1.3) or BD LSR Fortessa with acquisition software FACSDiva (v.8.1) and analyzed using FlowJo (v.10.5.3). We performed compensation using Invitrogen Ultracomp ebeads Compensation Beads, which were stained with appropriate antibody and analyzed on the same voltage and settings. Spleen tissue was harvested by mechanical dissociation and filtered through a 100 μm filter before RBC lysis and stained using the same protocol. BM was flushed from the femur and filtered before staining. A microglia gating strategy was used based on the finding that CD45 high and low are adequate markers in mouse to differentiate between macrophages and microglia, respectively, and P2ry12 was found to be a good secondary marker to confirm microglial identity[52]. MDSCs were identified using Ly6G$^+$Ly6C$^+$ and Ly6G$^-$Ly6C$^+$ for PMN-MDSCs and M-MDSCs, respectively[53]. The antibodies

used in flow cytometry can be found in Supplementary Tables 4 and 6. Antibody combinations against the following target antigens were used to define these cell types: macrophages, CD45$^{hi}$CD11b$^+$Ly6C$^-$Ly6G$^-$P2ry12$^-$; microglia, CD45$^{lo}$CD11b$^+$Ly6C$^-$Ly6G$^-$P2ry12$^+$; PMN-MDSCs CD45$^+$CD11b$^+$Ly6c$^+$Ly6G$^+$; M-MDSCs, CD45$^+$CD11b$^+$Ly6c$^+$Ly6G$^-$; CD8$^+$ T cells, CD45$^+$CD3$^+$CD8$^+$CD4$^-$; CD4$^+$ T cells, CD45$^+$CD3$^+$CD4$^+$CD8$^-$; Treg cells, CD45$^+$CD3$^+$CD4$^+$CD8$^-$Foxp3$^+$; tumor cells, CD45$^-$ hEGFR$^+$.

**BM cells and splenocytes isolation.** Preparation and staining from BM and splenic cells were done as described[24]. Briefly, cells from BM were isolated by flushing the femur of the indicated mice with PBS. ACK lysis was applied for 1 min and cells were washed three times with PBS 1× supplemented with 10% FBS followed by staining with the indicated antibodies and flow cytometry. Single cells were isolated from spleen by mashing on 70 μm filters followed by ACK lysis for 2 min and washing with PBS 1× supplemented with 10% FBS. Single-cell suspensions were plated in 96-well plates. We performed surface staining at 4 °C for 15 min with the flow antibodies listed in Supplementary Table 4. For intracellular staining, Cytofix/Cytoperm and Permwash staining kit (BD Pharmingen) were used according to the manufacturer's instructions. We performed intracellular staining 4 °C for 30 min with flow antibodies listed in Supplementary Table 4.

**Cell cycle and 5-ethynyl-2′-deoxyuridineanalysis by flow cytometry.** To determine the rate of proliferation in tumor microglia, 5-ethynyl-2′-deoxyuridine (EdU) at 100 mg kg$^{-1}$ dose was administered in tumor bearing mice (EGFR WT) intraperitoneally at 16 and 4 h before harvest. After perfusion, tumor-bearing side as well as the contralateral side of brain were harvested, and the cerebellum removed. Brain was minced into small pieces and resuspended in 1.5 mg ml$^{-1}$ collagenase IV in HBSS with calcium and magnesium. The mixture was incubated rotating at 37 °C for 30 min, with gentle dissociation using a P1000 pipette halfway through to break apart large pieces and enhance cell dissociation. The dissociated cell mixture was filtered through a 100 μm cell strainer, diluted with HBSS and pelleted at 400$g$ for 5 min. The pellet was resuspended with 30% Percoll in PBS and centrifuged at 700$g$ for 15 min with low brake to facilitate myelin removal. After myelin removal, the mixture was washed in PBS and pelleted at 400$g$ for 5 min. RBCs were lysed using Biolegend RBC lysis buffer according to the manufacturer protocol. $N=4$ mice were pooled per sample. Age-matched nontumor mice served as controls.

Throughout the experiment, RNase-free reagents were used, and all buffers contained 0.0025% RNasin Plus (Promega). Cells were stained with Zombie Yellow fixable viability dye (Biolegend) in PBS for 20 min in the dark on ice. Cells were washed once with PBS and blocked with Fc block for 5 min. Cell surface antigens (CD45 and P2RY12) were stained with antibodies for 30 min at 4 °C in the dark. Cells were washed twice with FACS buffer (1× RNase-free PBS(Invitrogen), 2% RNase-free BSA (VWR) and 0.0025% RNasin Plus) and fixed for 10 min at 4 °C with 2% paraformaldehyde using fixative provided with the EdU labeling kit (Invitrogen). Cells were permeabilized in saponin based permeabilization buffer (1× permeabilization buffer, 2% RNase-free BSA (VWR) and 0.0025% RNasin Plus). EdU-labeled cells were stained using a Click-IT reaction with Alexa Fluor 488 nm-azide (Click-iT Plus EdU Flow Cytometry Kit, Invitrogen) for 20 min at 4 °C in the dark. Cells were washed twice with 1× permeabilization buffer containing 0.0025% RNasin Plus and resuspended in FACS buffer for FACS.

Cells were placed on ice at all times until sorting. Cells were sorted in 500 μl FACs buffer (containing 1× RNase-free PBS, 2% RNase-free BSA and 0.0025% RNasin Plus). Beckman Coulter MoFlo AstriosEQ was using for cell sorting using a 120 μm nozzle. A minimum of 50,000 events were collected at Beckman Coulter Gallios flow cytometer and analyzed using FlowJo. Compensation was performed using Invitrogen Ultracomp ebeads Compensation Beads, which were stained with specific antibody and analyzed on the same voltage and settings. All samples were acquired under the same setting and analyzed using same gating strategy as shown in Extended Data Fig. 3c. Total RNA was isolated using miRNeasy FFPE Kit (Qiagen), using manufacturer protocol with some modifications. Briefly, cells were pelleted and resuspended in 240 μl of PKD/Proteinase K solution at RT, mixed, and incubated at 56 °C for 45 min. Samples were centrifuged at 16,000$g$ for 20 min. Supernatant was transferred to a new 1.5 ml Eppendorf tube and 10 μl of DNase Booster Buffer and 10 μl of DNase I stock solution were added for DNase treatment for 15 min. RBC buffer (500 μl) RBC buffer was added and mixed by pipetting several times; 1,200 μl ethanol was then added and mixed. Finally, the samples were passed through MinElute spin columns, washed three times with 500 μl RPE Buffer. RNA was eluted into 15 μl of RNase-free water and processed for bulk RNA-seq analysis as described below.

For cell cycle analysis, cells were stained with extracellular markers CD45, CD11b and P2ry12 and then fixed and kept in permeabilization buffer and stained with FxCycle Far Red for 20 min to label DNA content before flow cytometry analysis. DNA content stain was visualized on a linear axis and cells were read at the lowest flow rate for clear distinction in 2$N$ versus 4$N$ cells.

**Single-cell analysis pipeline.** Bioinformatic analyses were performed by the Joslin Diabetes Bioinformatics Core. Gene counts were generated by RapMap aligner[54] of bcbio-nextgen pipeline using the mouse GRCm38 transcriptome (Ensembl v.94).

To distinguish between droplets containing cells and ambient RNA, we used Monte Carlo simulations to compute $P$ values for the multinomial sampling transcripts from the ambient pool[55]. First, we assumed that some barcodes correspond to empty droplets if their total unique molecular identifier (UMI) counts are at or below 150. We then called cells at a false discovery rate (FDR) of 0.1%, meaning that no more than 0.1% of our called barcodes should be empty droplets on average. The number of Monte Carlo iterations determines the lower bound for the $P$ values[56]. There are no nonsignificant barcodes that are bounded by iterations, which indicated there was no need to increase the number of iterations to obtain even lower $P$ values. We computed the maximum contribution of the ambient solution to the expression profile for cell-containing droplets. First, we estimated the composition of the ambient pool of RNA based on the barcodes with total UMI counts less than or equal to 150 for each gene in each sample. Second, we computed the mean ambient contribution for each gene by scaling the ambient pool by some factor. Third, we computed a $P$ value for each gene based on the probability of observing an ambient count equal to or below that in cell-containing droplets based on Poisson distribution. Fourth, we combined $P$ values across all genes using Simes' method. We performed this for a range of scaling factors and identified the largest factor that yields a combined $P$ value above threshold 0.1 so that the ambient proportions are the maximum estimations.

To remove ambient contamination, we transformed and normalized the data for each sample using the R package sctransform[57] and performed principal component analysis (PCA) on the integrated data. To cluster the cells, we construct a K nearest neighbor (KNN) network of the cells based on the Euclidean distance in the space defined by the top principal components (PCs) selected by Horn's parallel analysis[58]. We refined the weights of the connection between pairs of cells based on their shared overlap in their local neighborhoods, that is, Jaccard similarity. To cluster the cells, we applied modularity optimization techniques, that is, the Louvain algorithm. We removed ambient RNA contamination from the cluster-level profiles and propagated the effect of those changes back to the individual cells using the removeAmbience function of the R package DropletUtils[55]. Doublets were identified using the R package scDblFinder and then removed from the downstream analysis. Briefly, thousands of doublets were first simulated by adding together two randomly chosen single-cell profiles. For each original cell, the density of simulated doublets in the surrounding neighborhood was computed. The simulated doublet density was then combined with an iterative classification scheme. For each observed cell, an initial score was computed by combining the fraction of simulated doublets in its neighborhood with another score based on coexpression of mutually exclusive gene pairs[59]. A threshold was chosen that best distinguishes between real and simulated cells, allowing us to obtain putative doublet calls among the real cells. Cell clustering and marker gene identification: we visualized quality control (QC) metrics as violin plots[60] that included the number of genes (nFeature), number of UMI (nCount) and percentage of mitochondrial UMI (percent_mt). We used the QC metrics to filter cells that have less than 500 UMI (low-quality cells), less than 500 features (low-quality cells) and more than 20% mitochondrial UMI (dying cells).

Integrating batch 1 and 2 data: we considered batch 1 and 2 as two independent datasets. Seurat includes a set of methods to match shared cell populations across datasets. These methods first identify cross-dataset pairs of cells that are in a matched biological state ('anchors') that can be used both to correct for technical differences between datasets (that is, batch effect correction) and to perform comparative scRNA-seq analysis of across experimental conditions[61]. We split batch 1 and 2 into two datasets. We transformed and normalized the split datasets separately using the R package sctransform[57]. We identified the 'anchors' and use these anchors to integrate the two datasets together. We performed PCA on the integrated data and only the top 3,000 variable genes were used as input. We performed UMAP analysis using the same top PCs as input to the clustering analysis.

We classified the cell cycle phases using the cyclone function of the R package scran[62]. Searching for DEGs (cluster biomarkers), we found markers for every cluster compared with all remaining cells using the Wilcoxon Rank Sum test, and reported those with a log$_2$FC threshold of 0.25 and expressed in more than 10% of the cells. To identify cell type, we used an automatic tool, SCSA[63], to annotate cell types for all cell clusters. It selects cell cluster markers with a $P$ value cutoff of 0.01. Selected markers were then compared with cell type specific markers in the CellMarker database[64]. If the $z$-score of the first predicted cell type is more than twice as much as the second predicted cell type, or the $z$-score of the second predicted cell type is negative, the first predicted cell type is considered a 'good' prediction by SCSA. If the $z$-score of the first predicted cell type is less than twice as much as the second predicted cell type, we manually selected the one that is likely found in brain tissue or the one that belongs to immune cells.

**PDGFRA and GAL-1 positive GBM populations.** Tumors were dissociated into single-cell suspensions as described above. Single-cell suspensions were stained with antibodies EGFR PerCP Cy5.5 (clone: AY13, Biolegend), with either PDGFR APC (clone: APA5, BD Biosciences) or Galectin-1 PE (R&D systems). Cell surface antigens are stained for 30 min at 4 °C in the dark. Cells were washed with RNase-free PBS with 0.1% BSA and immediately sorted on a FACS Aria Cell Sorter for EGFR$^+$PDGFR$^-$, EGFR$^+$PDGFR$^+$, EGFR$^+$Galectin-1$^+$ and EGFR+Galectin-1$^-$

populations. Sorted cells were resuspended immediately in Qiagen RLT buffer. RNA was isolated using the Qiagen RNeasy Micro kit as per the manufacturer's protocol and eluted using 14 μl RNase-free water.

**Bulk RNA-seq analyses.** RNA isolated from flow cytometry sorted cells was processed for bulk RNAseq as follows: libraries were prepared using SMARTer Stranded Total RNAseq v.3 Pico Input Mammalian sample preparation kits from 1 ng purified total RNA according to the manufacturer's protocol. The finished dsDNA libraries were quantified by Qubit fluorometer and Agilent TapeStation 4200. Uniquely dual indexed libraries were pooled in an equimolar ratio and shallowly sequenced on an Illumina MiSeq to further evaluate library quality and pool balance. The final pool was sequenced on an Illumina NovaSeq 6000 paired-end 150 bp reads (for the EdU-labeled experiment) or a High Output Illumina NextSeq550 (for the PDGFRA and GAL-1 experiment) at the Dana-Farber Cancer Institute Molecular Biology Core Facilities. The reads were trimmed for adapters and poly(A/T) tails, and then filtered by sequencing Phred quality (≥ Q15) using fastp[65]. Mouse genome sequences and gene annotation were downloaded from the UCSC goldenPath, v.mm39. We then used the genomeGenerate module of the STAR aligner[66] to generate the genome indexes. STAR aligner option sjdbOverhang = 74 for 75 bp reads.

We aligned the adapter-trimmed reads to the genome using STAR aligner with the two-pass option. Reads are mapped across the genome to identify novel splice junctions in the first-pass. These new annotations are then incorporated into the reference indexes and reads are realigned with this new reference in the second pass. While more time-intensive, this step can aid in aligning across these junctions, especially in organisms where the transcriptome is not as well annotated. Alignments were assigned to genomic features (that is, the exons for spliced RNAs) by using featureCounts[67]. Multi-mapping reads are counted as fractions. To filter out low expressing genes, we kept genes that have counts per million (CPM) more than 1 in at least three samples; there were 18,180 genes (PDGFRA/GAL1 experiment) and 14,727 genes (for the EdU experiment) after filtering. We then normalized counts by weighted trimmed mean of M-values (TMM)[68]. If no normalization is needed, all the normalization factors will be 1. We used normalization factors between 0.84 and 1.15. To use linear models in the following analysis, we performed Voom transformation[69] to transform counts into logCPM, where logCPM=log_2 ($10^6$×count/(library size×normalization factor)). Voom transformation estimates the mean-variance relationship and uses it to compute appropriate observation-level weights so that more read depth gives more weight. To get an overall view of the similarity and/or difference of the samples, we performed PCA. We then adjusted for the tumor effects and recalculated the PCA analysis using the adjusted data. To discover DEGs, we use limma, an R package that uses linear models to power differential expression analyses[70]. We performed moderated t-tests to detect genes that are deferentially expressed between groups, with the tumors as covariates. We performed GSEA[71] using the microglia sensome gene sets[22]. GSEA for Reactome. We perform GSEA using the Reactome, and the lists of all genes' Z-scores of each comparison.

**Mouse microglia RNAseq comparison analysis.** Data from Bennett et al[72] were acquired from SRA under accession number SRP068621. The downloaded data are paired-end fastq files and each pair should have the same number of matched reads. However, for these data, there are different numbers of reads in each file (probably because some reads fail QC). Therefore, we synchronized the paired-end fastq files and separated unmatched reads using fastq-pair. RNA-seq raw reads were 75-bp unstranded paired-end reads. The reads were trimmed for adapters and filtered by sequencing Phred quality (≥ Q15) using fastp[65]. A count table was generated by aligning reads to the mouse transcriptome (Ensembl v.104) using kallisto[73] and converting transcript counts to gene counts using tximport[74]. These samples had an average alignment rate of ~57%. To filter out low expressing genes, we kept genes that have CPM more than 0.5 in at least 1 sample; there were 19,050 genes after filtering. We then normalized counts by weighted TMM[68]. If no normalization was needed, all the normalization factors will be 1. Here, the normalization factors were between 0.53 and 1.37. To use linear models in the following analysis, we performed Voom transformation as above. We performed Spearman correlation tests between different ages of mouse microglia, central nervous system myeloid, whole-brain gene expression and our microglia gene expression.

**Define M1-like/M2-like phenotypes.** GSEA was used to define the M1-like proinflammatory and M2-like anti-inflammatory phenotypes of clusters. Genes associated with 'classically activated' (M1-like) macrophages and microglia include Ccl5, Ccr7, Cd40, Cd86, Cxcl9, Cxcl10, Cxcl11, Ido1, Il1a, Il1b, Il6, Irf1, Irf5 and Kynu, and genes associated with an 'alternatively activated' (M2-like) anti-inflammatory phenotype are Ccl4, Ccl13, Ccl18, Ccl20, Ccl22, Cd276, Clec7a, Ctsa, Ctsb, Ctsc, Ctsd, Fn1, Il4r, Irf4, Lyve1, Mmp9, Mmp14, Mmp19, Msr1, Tgfb1, Tgfb2, Tgfb3, Tnfsf8, Tnfsf12, Vegfa, Vegfb and Vegfc[31].

**qPCR analysis of cytokine expression.** Cells were sorted on CD45 expression as above and immediately resuspended in Qiagen RLT buffer. RNA was isolated using the Qiagen RNeasy Micro kit as per manufacturer protocol and eluted

using 12 μl of RNase-free water. RNA (35 ng) was used for RT reaction using the SuperScript III cDNA synthesis kit according to the manufacturer's protocol. Primers were found in the literature as cited or from Primerbank[75–77] and can be found in Supplementary Table 7. Primer pairs were optimized for appropriate concentrations by agarose gel to assess for a single product of expected size and minimal dimer formation. Sybr Green was used per manufacturer protocol and run on a Stratagene Mx3000p. Expression level was determined using the ddCt method comparing CD45+ to CD45- expression and normalizing to GAPDH.

**BBB integrity evaluation.** Tumor-bearing mice at the indicated time points and nontumor-bearing mice were anaesthetized using ketamine/xylazine mixture (ketamine 100 mg kg$^{-1}$ and xylazine 10 mg kg$^{-1}$; intraperitoneally (i.p.)) and mice were administered a 2% solution of Evans Blue (EB; Sigma, catalog no. E2129-10G) and 2% solution NaF (F6377-100G) intravenously at a dose of 4 ml kg$^{-1}$ of body weight[30] through the retro-orbital plexus. At 30 min postinjection, mice were perfused transcardially with 0.9% NaCl until the perfusate from right atrium ran clear. Mice were decapitated and dissected into coronal sections to visualize the extravasation of tracer markers. Each region was weighed and then homogenized in 0.75 ml PBS and 0.25 ml 100% trichloroacetic acid (Sigma Aldrich, catalog no. T6399-100G). Fluorescence of the extravasated dye was determined by spectrophotofluorometry (excitation at 620 nm and emission at 680 nm for EB and excitation at 440 nm and emission at 525 nm for NaF) and expressed as micrograms per gram of brain tissue using standard curves for EB and NaF.

Gad-enhanced magnetic resonance imaging (MRI) was performed using a small animal MRI device (Bruker Biospec 94/20, Bruker) equipped with an 84 mm quadrature transmit volume coil and a four-element mouse head array. Animals were anesthetized with isoflurane in oxygen and situated on the MRI bed with respiratory monitoring and thermal support provided by a warm air circulator. For tumor volume quantitation, images of the brain were acquired in axial and coronal planes using a RARE (rapid acquisition with refocused echoes) pulse sequence with T1 and T2 weighting. Axial T1-weighted images used TR/TE=800/23 ms, 0.8 mm slice thickness and 125 μm in-plane resolution. Coronal T1-weighted images had TR/TE=863/23 ms, 0.8 mm slice thickness and 156 μm in-plane resolution. Axial T2-weighted images had TR/TE=1800/33 ms, 0.6 mm slice thickness and 125 μm in-plane resolution, while coronal T2-weighted images were acquired with TR/TE=44/2,246 ms, 0.6 mm slice thickness and 156 μm in-plane resolution. T1-weighted images were acquired with eight signal averages, and T2-weighted images had four signal averages. All images had RARE factor 8. For assessment of tumor permeability, anesthetized animals were given 0.2 mmol kg$^{-1}$ of gadoteridol (ProHance) i.p., dissolved in approximately 100 μl sterile saline. The mice were then placed promptly in the MRI device, and following acquisition of scout images for localization of the relevant anatomy, T1-weighted RARE images were acquired with RARE factor 4, TR/TE=500/16 ms, 0.8 mm slice thickness, 125 μm in-plane resolution and four signal averages. For each animal, the first T1-weighted images were acquired at exactly 10 min following contrast administration. To assess contrast uptake and washout, imaging was repeated every 1–2 min for 30 min. Images were analyzed using Horos DICOM viewer. Three independent measurements from a minimum of two MR slices were used to collect Gd-enhanced data. Volumes of T2 weighted images were calculated by establishing ROI on serial slices.

**EGFR immunohistochemistry.** Deeply anesthetized animals were perfused transcardially with cold PBS. Brains were excised, rinsed in PBS and serial coronal sections cut using a brain mold. Thick sections were postfixed in 4% paraformaldehyde overnight. Formalin-fixed tissues were embedded in paraffin, sectioned at 5–10 μm, and stained with hematoxylin and eosin (Sigma) for histopathological analysis. For immunohistochemistry (IHC), cut sections were deparaffinized and rehydrated through a xylene and graded alcohol series and rinsed for 5 min under tap water. Antigen target retrieval solution (Dako, catalog no. S1699) was used to unmask the antigen (microwave for 10 min at low power then cooled down for 30 min) followed by three washes with PBS for 5 min each. Quenching of endogenous peroxidase activity was performed by incubating the sections for 10 min in 0.3% $H_2O_2$ in methanol followed by PBS washes. Slides were preincubated in blocking solution (5% (v/v) goat serum (Sigma) in PBS 0.3% (v/v) Triton-X100) for 1 h at room temperature, followed by mouse-on-mouse blocking reagent (Vector Labs, Inc., catalog no. MKB-2213) incubation for 1 h. Primary antibody was incubated for 24 h at 4 °C. Secondary antibodies used were biotinylated anti-rabbit or anti-mouse (1:500, Vector Labs, Inc.) for IHC and were incubated for 1 h at room temperature. All antibodies were diluted in blocking solution. All immunobinding of primary antibodies was detected by biotin-conjugated secondary antibodies and Vectastain ABC kit (Vector Labs, Inc.) using DAB (Vector Labs, Inc.) as a substrate for peroxidase and counterstained with hematoxylin. Anti-human EGFR primary antibodies were used (EGFR; catalog no. 28-0005, 1:200, Zymed). EGFR staining (cluster numbers and diameters in millimeters squared) was quantified from photomicrographs and Image J by two independent observers who were blind to the images. At least three fields of view per image, four images per tumor and a minimum of n = 3 tumors were analyzed per timepoint.

**GO analysis.** To identify genes enriched in Biological Process (BP), Molecular Function (MF) and Cellular Component (CC), we utilized the Database for Annotation, Visualization, and Integrated Discovery (DAVID) v.7.0 (refs. [78,79]) with GOTERMs BP, MF and CC. All terms with a *P* value (Benjamini or Benjamini–Hochberg adjusted) less than 0.05 were considered significant and ranked by the number of genes identified in the group.

**Statistical analyses.** scRNA-seq statistical analysis was completed as described above. All other statistical analyses were performed using GraphPad Prism (GraphPad Software v.9.3.0). Values are given as mean ± s.e.m. or s.d. as indicated. Numbers of experimental replicates are given in the figure legends. When two groups were compared, significance was determined using an unpaired two-tailed *t*-test. For comparing more than two groups, one-way analysis of variance (ANOVA) was applied. Significance for survival analyses was determined by the log-rank (Mantel–Cox) test. *P* values < 0.05 are considered as statistically significant. No statistical methods were used to predetermine sample sizes, and our sample sizes were similar to those reported in previous publications[2–4]. Mice were assigned randomly to the various experimental groups described. Data collection and analysis were not performed blind to the conditions of the experiments except for analysis shown in Fig. 2b,c. Data distribution was assumed to be normal, but this was not formally tested. No datapoints were excluded from the analyses.

**Reporting summary.** Further information on research design is available in the Nature Research Reporting Summary linked to this article.

## Data availability
All data and materials used in the analysis are available in some form to any researcher for purposes of reproducing or extending the analysis. In rare instances, a material transfer agreement (MTA) may be required. scRNA-seq and bulk RNAseq data files are publicly accessible in the Gene Expression Omnibus under accession numbers GSE195848, GSE196174, GSE196175 and GSE195813. All analyses and visualizations were performed in R (v.3.6.3). Source data are provided with this paper.

## Code availability
Computer code is available upon reasonable request.

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

## Acknowledgements

The authors would like to thank P. Zhang and H. Jung Jun for critical review of the manuscript and H. Pan and J. Dreyfuss of the Joslin Bioinformatics Core (Harvard Medical School) for their technical contribution. This work was supported by National Institutes of Health grant R01 CA229784 to V.A.B. and A.C., P01 CA069246 to B.S.C. and A.C., U01 CA230697, R01 CA239078, R01 CA237500 to B.S.C.

## Author contributions

B.D., V.A.B. and A.C. conceived and designed the study. B.D., S.R., A.C., A.A., A.T.Y., L.S., L.B., V.A.R., E.J.U., H.V. and B.S.C. were involved in acquisition of data. B.D., S.R., A.C., A.A., A.T.Y., L.S., H.V., V.A.B. and A.C. analyzed and interpreted the data. V.A.B. and A.C. wrote, reviewed and revised the manuscript. A.C. supervised the study. All authors reviewed the manuscript.

## Competing interests
V.A.B. has patents on the PD-1 pathway licensed by Bristol-Myers Squibb, Roche, Merck, EMD-Serono, Boehringer Ingelheim, AstraZeneca, Novartis and Dako. The other authors declare no competing interests.

## Additional information
**Extended data** is available for this paper at https://doi.org/10.1038/s41590-022-01215-0.

**Correspondence and requests for materials** should be addressed to Vassiliki A. Boussiotis or Al Charest.

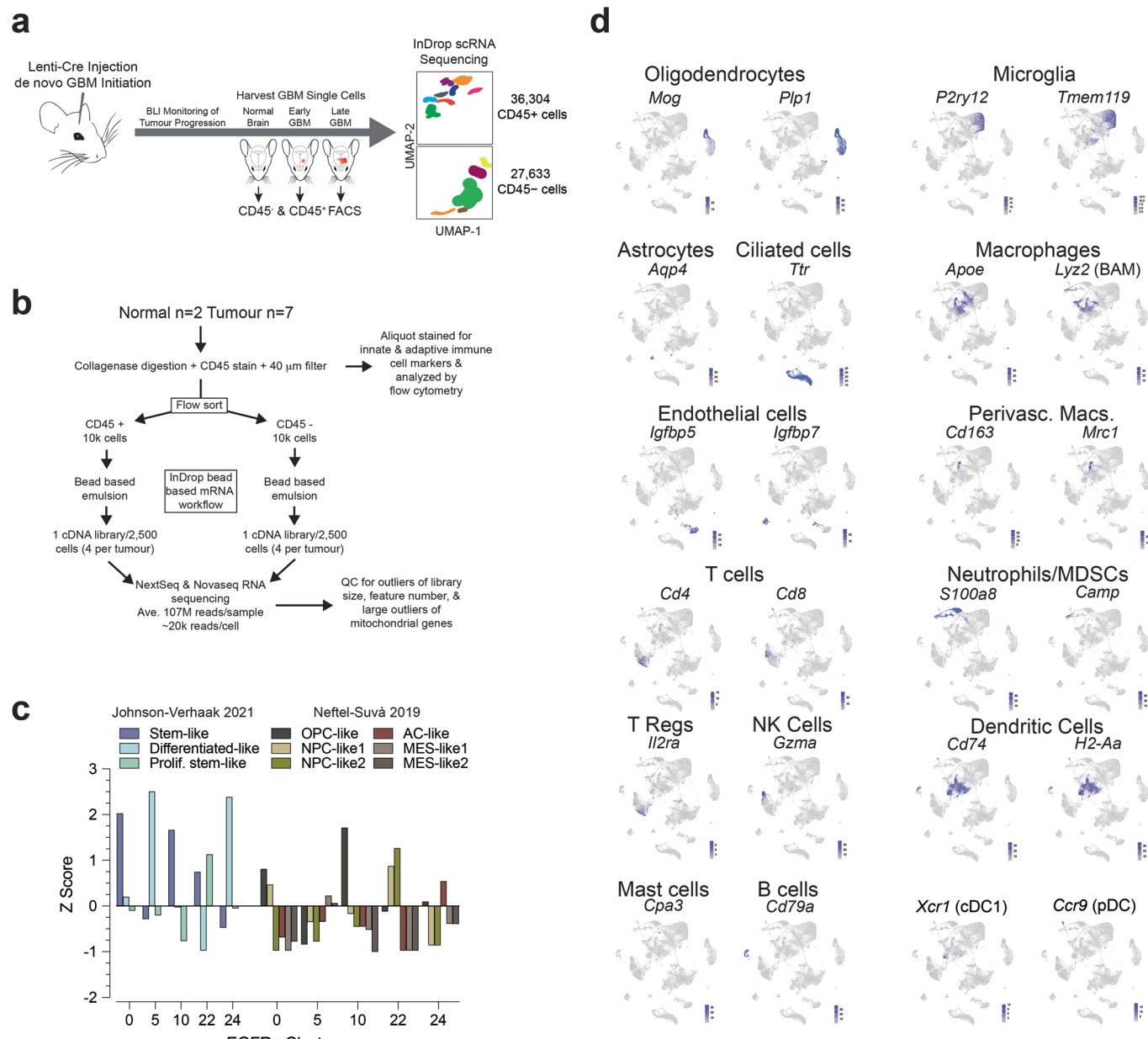

**Extended Data Fig. 1 | Single cell RNAseq. a**, Overview of the experimental procedure. See Methods and main text for details. **b**, Schematic of the single-cell RNA-seq experimental pipeline procedure. **c**, Alignment (Z scored) of mouse GBM cells to human GBM tumor cells subtype markers from scRNA-seq-derived independent analyses. **d**, Expression of characteristic cell-type-specific genes overlaid on the UMAP space.

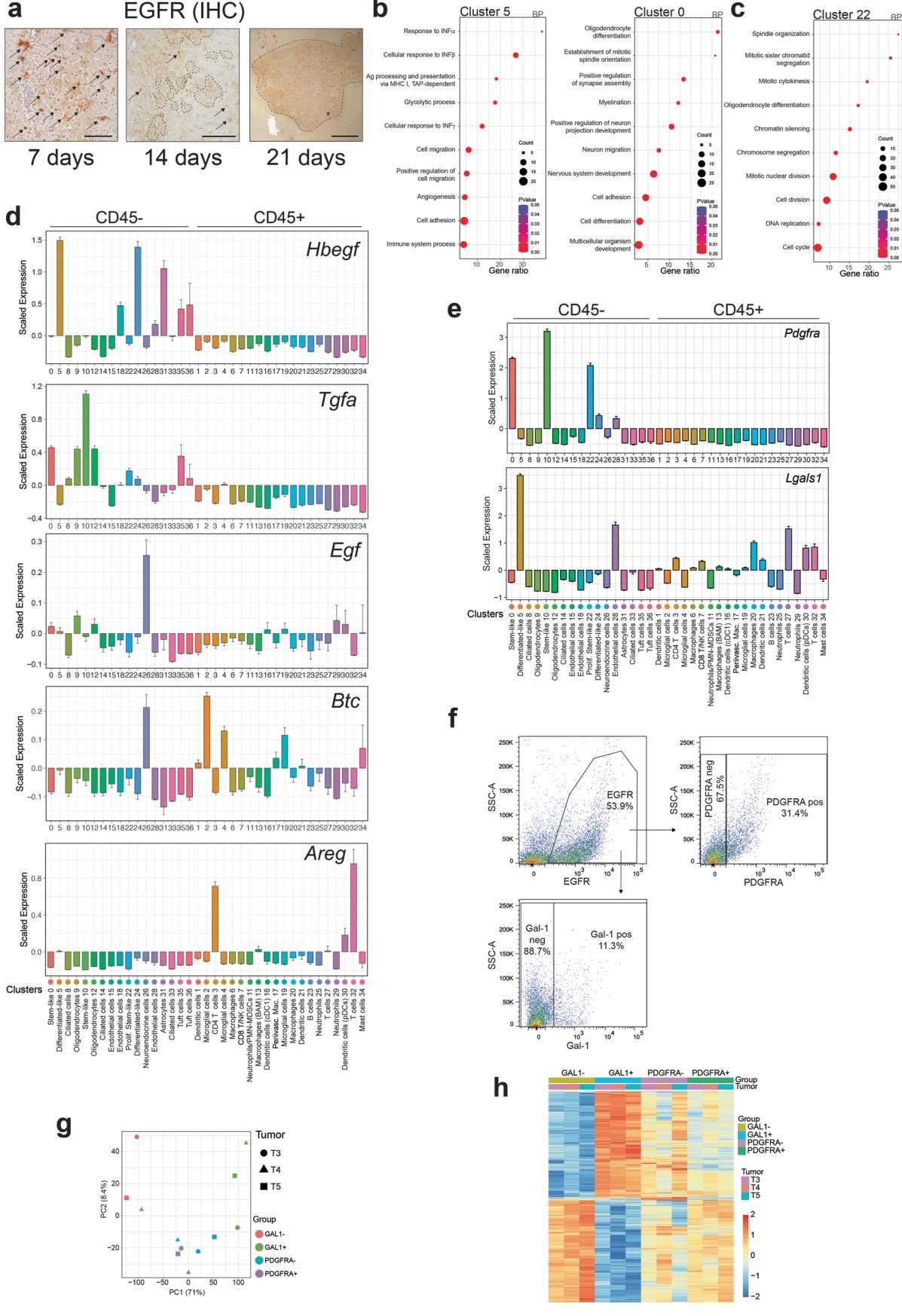

**Extended Data Fig. 2 | See next page for caption.**

**Extended Data Fig. 2 | Characteristics of EGFR positive GBM cells. a**, Representative photomicrographs of EGFR IHC of GBMs at 7, 14 and 21 days post initiation. Scale bars, 7 days 100 µm, 14 days 200 µm and 21 days 500 µm. **b**, GO analysis of DEGs between clusters 0 and 5. BP, biological processes. **c**, GO analysis of cluster 22. **d**, Scaled expression of indicated EGFR ligands in all clusters. **e**, Scaled expression of *Pdgfra* and *Lgals1* (GAL1) in all clusters. **f**, Representative flow cytometry gating scheme used to sort PDGFRA and GAL1 positive and negative populations. **g**, Principal component analysis of the tumor samples' RNAseq datasets. **h**, Unsupervised hierarchical clustering of the bulk RNAseq from PDGFRA⁻, PDGFRA⁺, GAL1⁻ and GAL1⁺ samples.

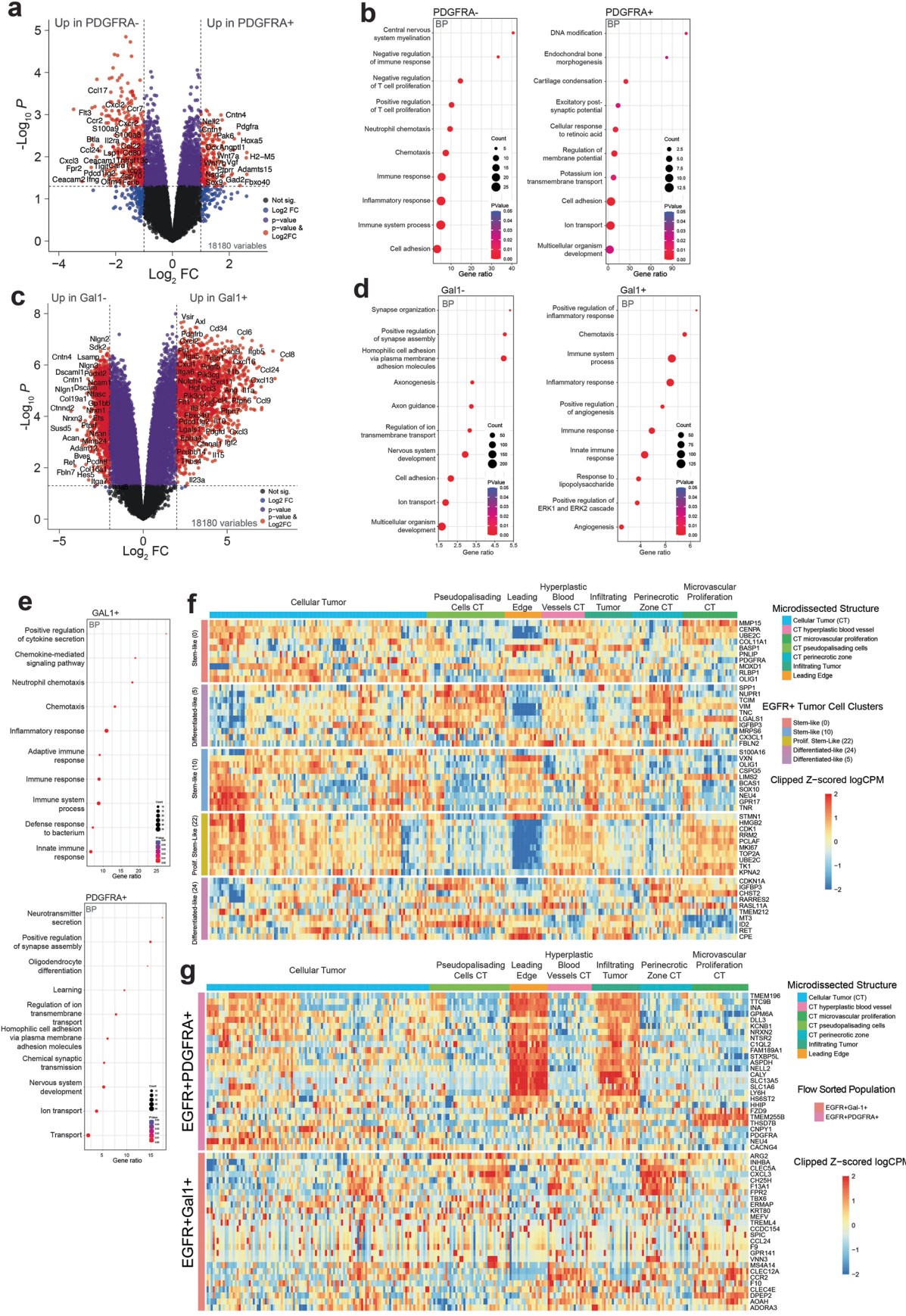

**Extended Data Fig. 3 | See next page for caption.**

**Extended Data Fig. 3 | Transcriptomic characteristics of EGFR positive PDGFRA+ and GAL1+ GBM cells. a, b**, Volcano plot of $Log_2FC$ expression levels of genes (**a**) and GO analysis of DEGs (**b**) from PDGFRA$^+$ vs PDGFRA$^-$ samples. **c, d**, Volcano plot of $Log_2FC$ expression levels of genes (**c**) and GO analysis of DEGs (**d**) from GAL1$^+$ vs GAL1$^-$ samples. **e**, GO analysis of the DEGs DEGs from bulk RNAseq of hsEGFR$^+$PDGFRA$^+$ and hsEGFR$^+$GAL1$^+$ flow-sorted cells.BP; biological processes. **f**, Alignment of the top expressed genes from *EGFR*$^+$ clusters 0, 5, 10, 22 and 24 to the IVY GAP datasets. **g**, Alignment of the top expressed genes from EGFR$^+$ PDGFRA$^+$ and EGFR$^+$ GAL1$^+$ populations to the IVY GAP datasets.

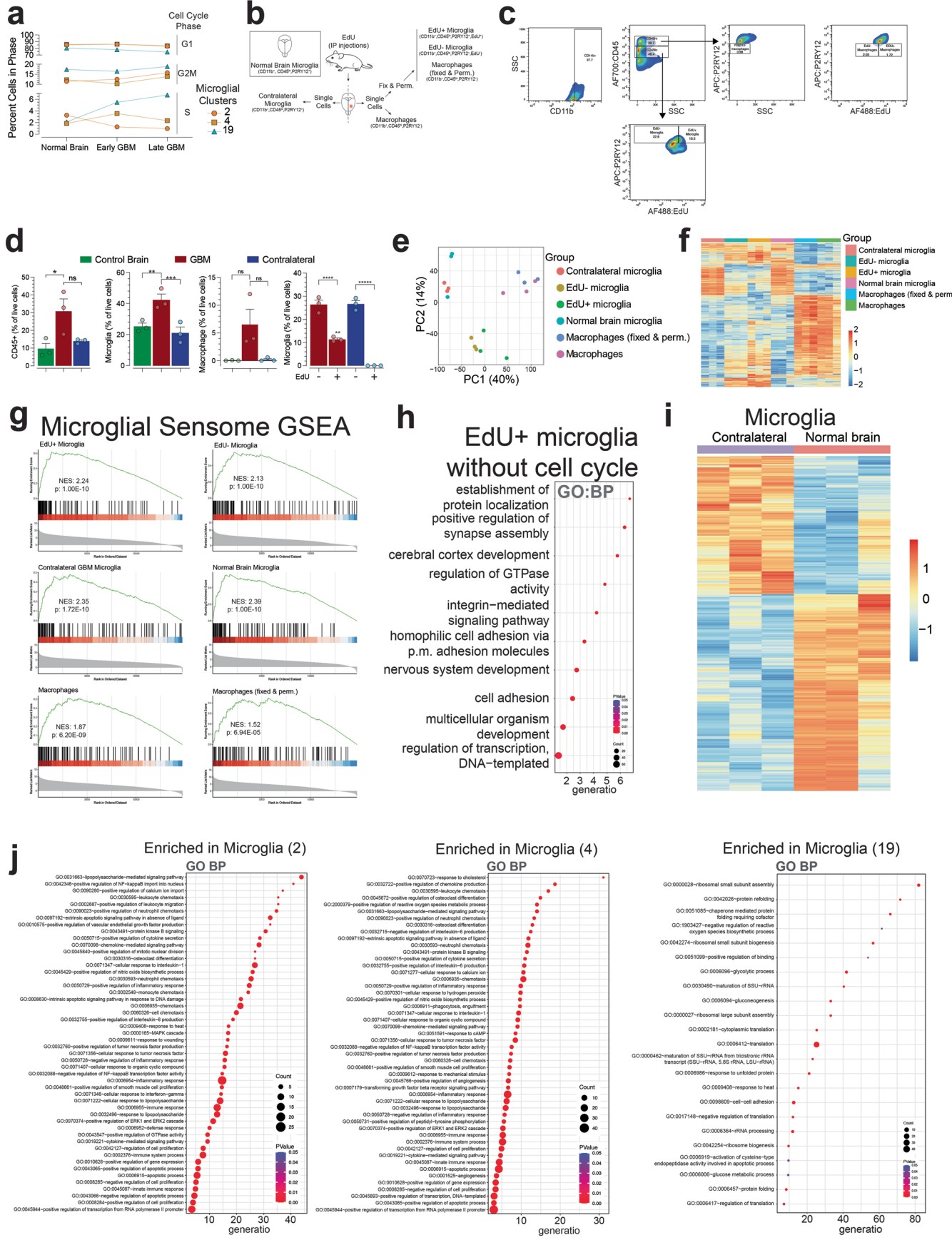

**Extended Data Fig. 4 | See next page for caption.**

**Extended Data Fig. 4 | Isolation and analysis of proliferating microglia cells. a**, Percent microglia expressing cell cycle phase marker genes from clusters 2, 4 and 19 in normal brain control, Early and Late GBM samples. **b**, Scheme to flow sort EdU-labeled microglia. **c**, Flow sorting gating strategy. **d**, Relative numbers of CD45+, microglia and macrophage cells from normal brain controls, GBM and contralateral CNS tissue from GBM brains by flow cytometry. Data is mean ± SEM of biologically independent replicates n = 3 each control brain, GBM and contralateral. One-way analysis of variance (ANOVA) *p = 0.0435, *p = 0.02, panel 4 *p = 0.0018, ** p = 0.0324, ***p = 0.0119, ****p = 0.0008, *****p < 0.0001. ns, not significant. **e**, Principal component analysis of the indicated samples, n = 3. **f**, Heatmap from unsupervised hierarchical clustering of the bulk RNAseq from indicated flow-sorted bulk RNAseq samples. **g**, Gene set enrichment analysis of microglial sensome genes applied to flow-sorted bulk RNAseq samples. **h**, GO analysis of EdU+ microglia DEGs with cell cycle genes removed. BP, biological process. **i**, Heat map of DEGs between GBM contralateral microglia and normal brain microglia. **j**, GO analysis of DEGs enriched in microglia clusters 2, 4, and 19.

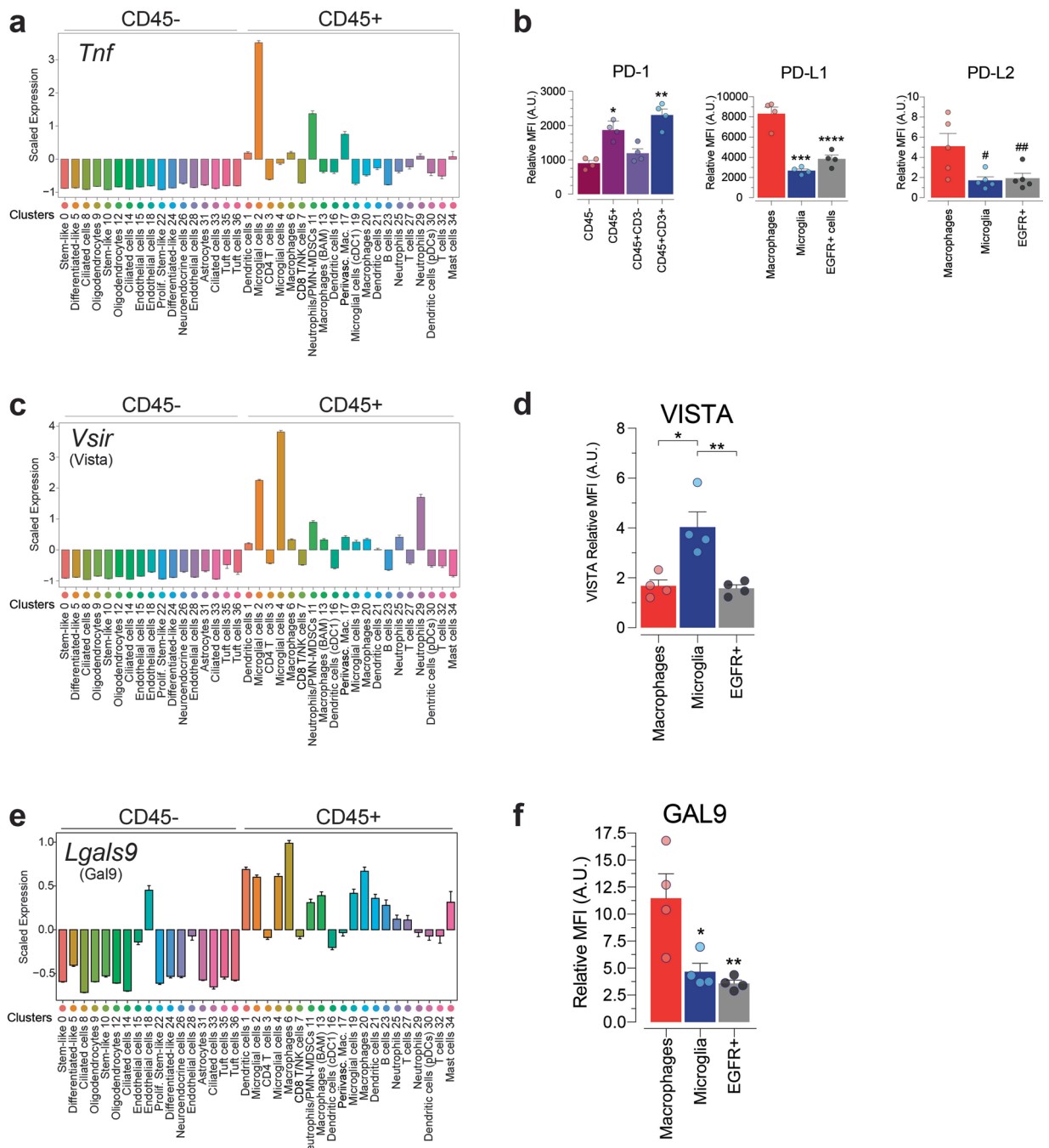

**Extended Data Fig. 5 | Expression of cytokines and checkpoint receptors and ligands is restricted to CD45+ cell populations. a**, Scaled expression levels of *Tnf* transcripts in CD45⁻ and CD45⁺ clusters. **b**, Flow cytometry MFI expression levels of PD-1, PD-L1, and PD-L2 in GBM. Data is mean ± SD of biologically independent tumors n = 4, *p = 0.0008, **p = 0.0024, ***p = 0.0002, ****p = 0.001, #p = 0.0328, ##p = 0.0474 unpaired t test, two-tailed. **c**, Scaled expression levels of *Vsir* (Vista) transcripts in CD45⁻ and CD45⁺ populations. **d**, Flow cytometry MFI expression levels of VISTA in GBM. Data is mean ± SD of biologically independent tumors n = 4, *p = 0.0118, **p = 0.0081, unpaired t test, two-tailed. **e**, Scale expression levels of *Lgals9* transcripts in CD45⁻ and CD45⁺ populations. f, Flow cytometry MFI expression GAL9 in GBM tumors. Data is mean ± SD of biologically independent tumors n = 4, *p = 0.0297, **p = 0.0136, unpaired t test, two-tailed.

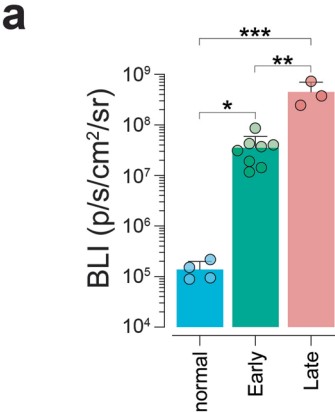

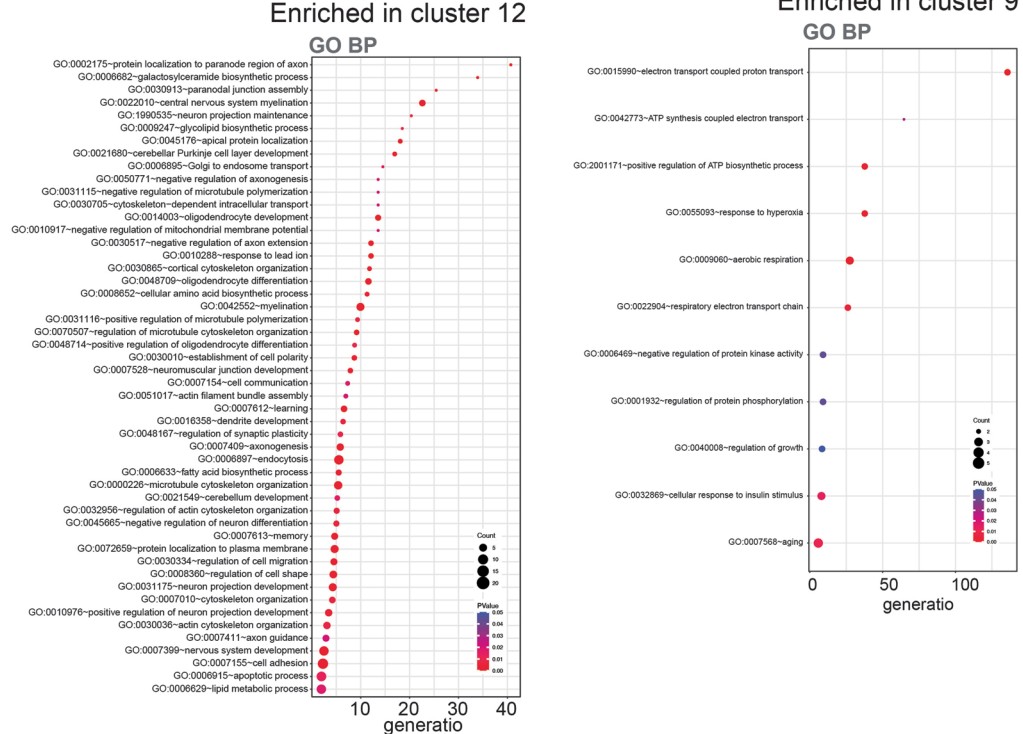

**Extended Data Fig. 6 | Bioluminescence imaging of mice and oligodendrocyte molecular changes during GBM progression. a**, BLI output of normal control mice (showing background levels of bioluminescence), and mice used in Early and Late samples. Data is presented as mean ± SD of biologically independent replicates. Normal brain n = 4, Early GBM n = 8, and Late GBM n = 3, *p = 0.0162, **p = 0.0135, ***p = 0.0006,, unpaired t test two-tail. **b**, GO analysis of oligodendrocyte DEGs enriched between clusters 9 and 12. BP, biological processes.

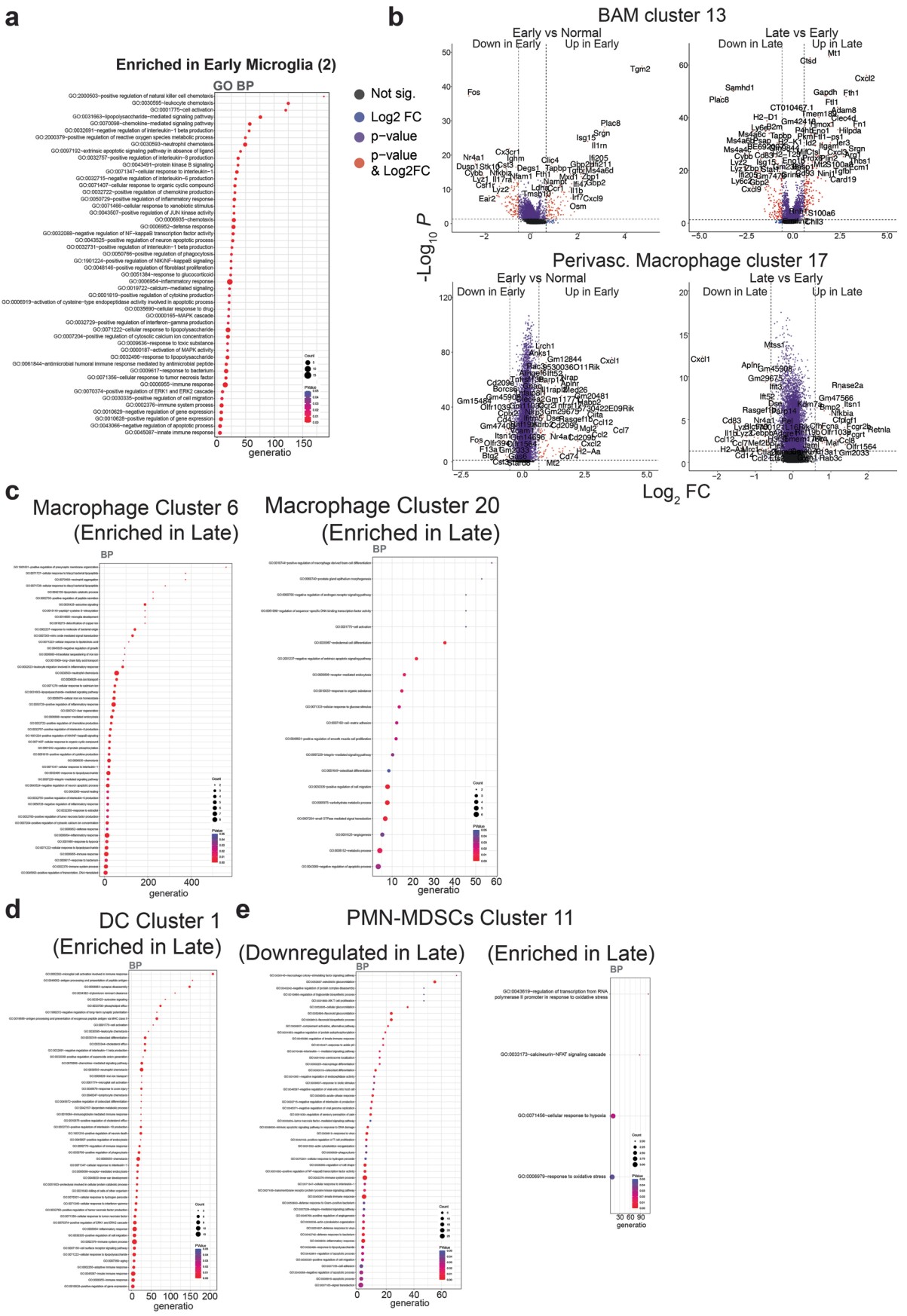

**Extended Data Fig. 7 | See next page for caption.**

**Extended Data Fig. 7 | Changes in GBM infiltrating immune cells during progression. a**, GO analysis of DEGs from microglia cluster 2 enriched in Early GBM. **b**, DEGs of barrier associated macrophage (BAM) cluster 13 and perivascular macrophage cluster 17 enriched in Late and Early GBM samples. **c-e**, GO analysis of DEGs upregulated in Late vs Early GBM samples for macrophage cluster 6 (**c**) and DC cluster 1 (**d**) and enriched and downregulated in Late neutrophil/PMN-MDSCs cluster 11 (**e**).

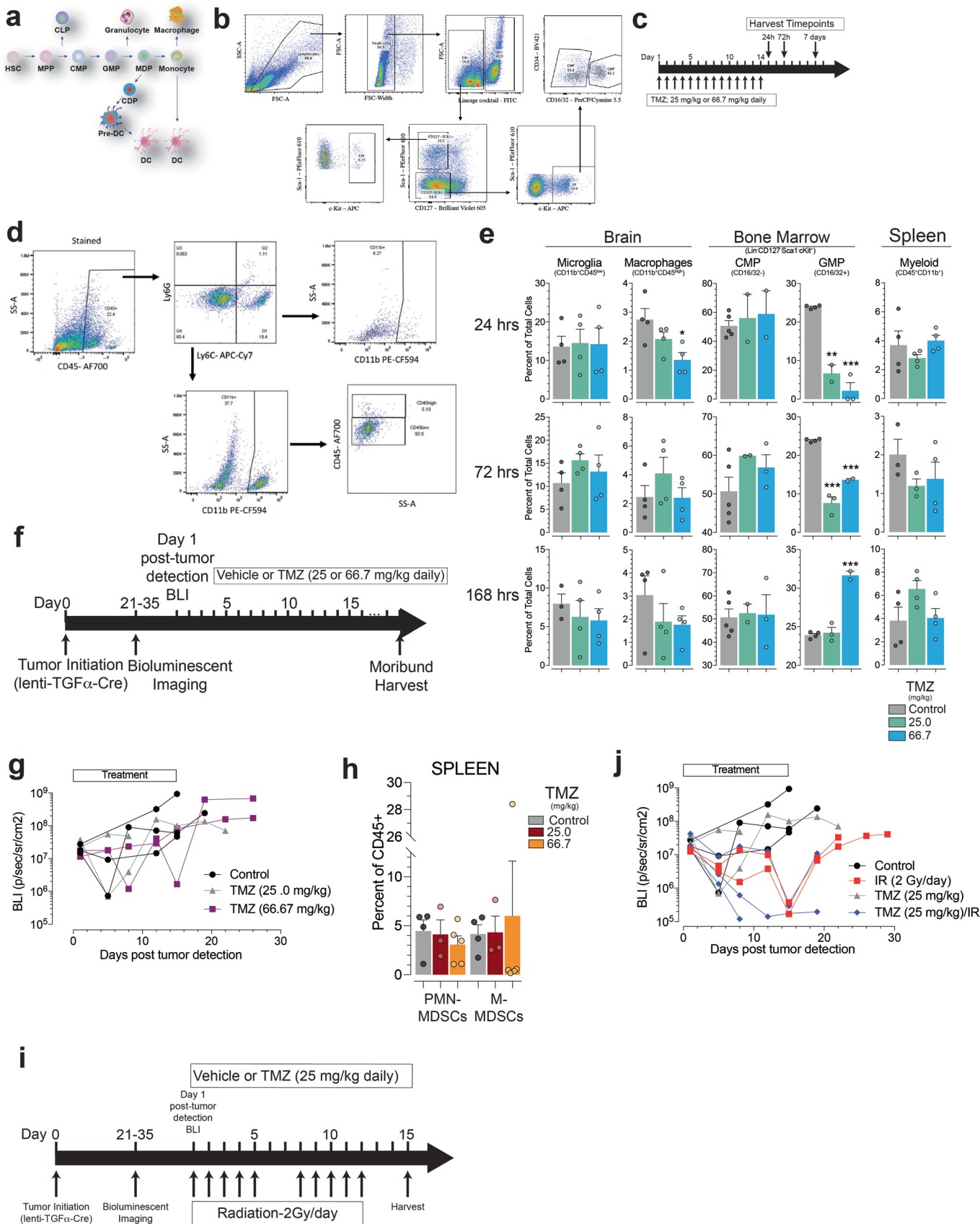

Extended Data Fig. 8 | See next page for caption.

**Extended Data Fig. 8 | Gating strategies and treatment schemes. a**, Hematopoesis schema. HSC: hematopoietic stem cell, MPP: multipotent progenitor, CLP: common lymphoid progenitor, CMP: common myeloid progenitor, GMP: granulocyte-monocyte progenitor, MDP: monocyte-dendritic cell progenitor, CDP: common dendritic progenitor, DC: dendritic cell. **b**, Flow cytometry gating strategy of cells isolated from bone marrow. The schema represents the sequential steps of the gating strategy. The various cell populations within the bone marrow are depicted. LSK ($Lin^{neg}$, $Sca1^{pos}$, $CD127^{neg}$, $c\text{-}kit^{pos}$) and LK ($Lin^{neg}$, $Sca1^{neg}$, $CD127^{neg}$, $c\text{-}kit^{pos}$) hematopoietic precursors and CMP ($Lin^{neg}$, $Sca1^{neg}$, $CD127^{neg}$, $c\text{-}kit^{pos}$, $CD16/CD32^{neg}$) and GMP ($Lin^{neg}$, $Sca1^{neg}$, $CD127^{neg}$, $c\text{-}kit^{pos}$, $CD16/CD32^{pos}$) myeloid precursors. **c**, Schematic of the TMZ treatment strategy. **d**, Flow cytometry gating strategy of cells isolated from normal brain. The schema represents the sequential steps of the gating strategy. The various immune cell populations within the brain are depicted. **e**, Flow cytometry data for the indicated cell types from brain, bone marrow and spleen tissues of control and TMZ (25 mg/kg or 66.7 mg/kg daily) treated mice. Data represent mean ± SEM of biologically independent replicates. n = 4, 4, and 4 for control, 25 and 66.7 mg/kg of TMZ respectively for brain microglia, brain macrophages and spleen myeloid 24, 72 and 168 hrs. n = 5, 2 and 3 for control, 25 and 66.7 mg/kg of TMZ respectively for bone marrow CMP. n = 4, 3 and 2 for control, 25 and 66.7 mg/kg of TMZ respectively for bone marrow GMP. *p = 0.0228, **p = 0.0002, ***p < 0.0001, unpaired t test two-tailed. **f**, Schematic of the TMZ treatment strategy. **g**, Longitudinal BLI from representative GBM mice undergoing TMZ treatments. **h**, Flow cytometry from spleens of GBM mice treated with control vehicle, 25 and 66.7 mg/kg of TMZ. Data is presented as mean ± SD of biologically independent replicates n = 4, 3 and 5 for control, 25 and 66.7 mg/kg TMZ respectively, no statistically significant differences were identified. **i**, Schematic of the TMZ/IR treatment strategy. **j**, Longitudinal BLI from representative GBM mice undergoing TMZ and/or IR treatments.

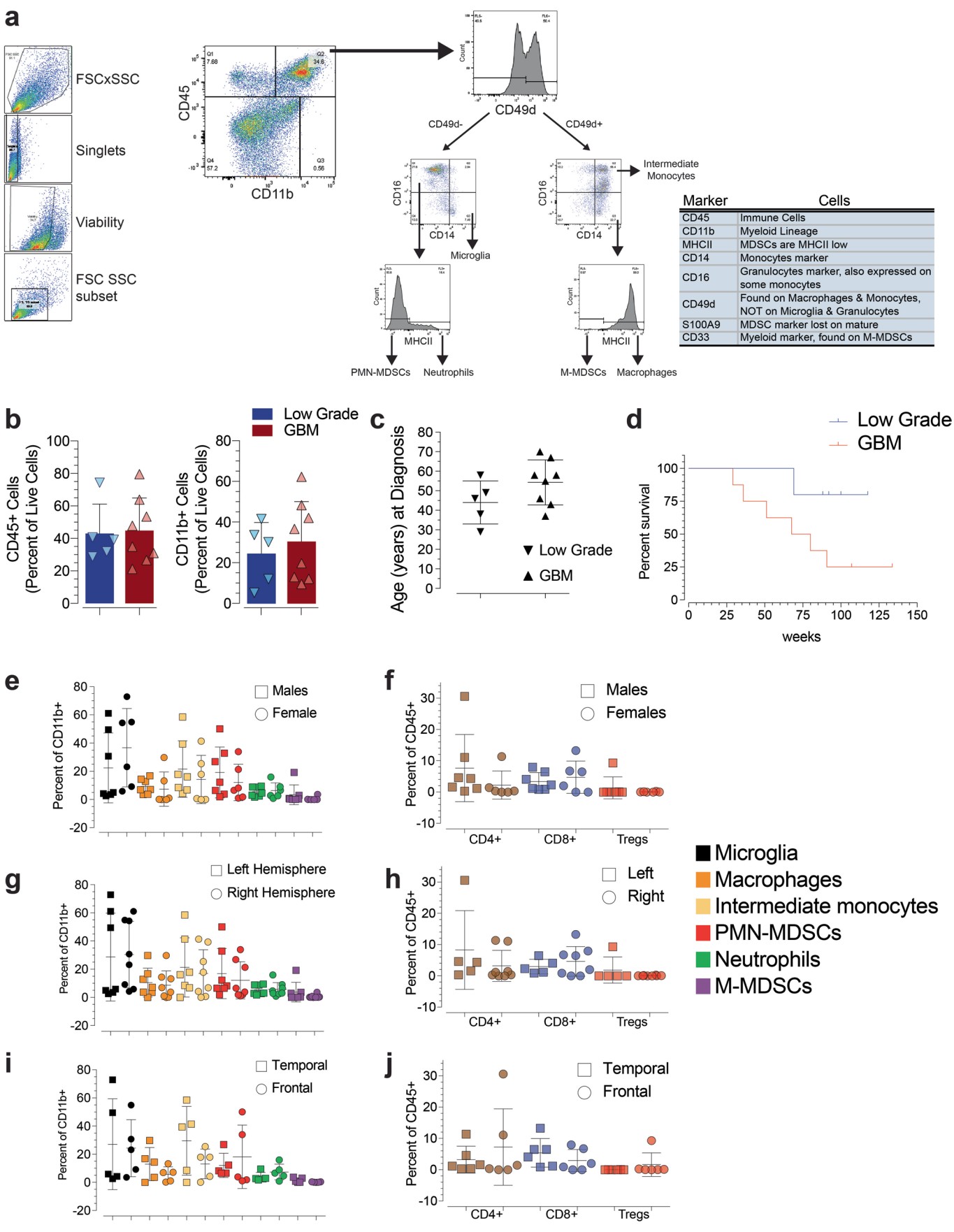

**Extended Data Fig. 9 | See next page for caption.**

**Extended Data Fig. 9 | Characterization of human low-grade glioma and GBM immune composition by flow cytometry. a**, Representative flow cytometry plots depicting the gating strategy used to analyzed and quantitate single cells from patient samples. **b**, Flow cytometry of CD45$^+$ and CD11b$^+$ cells biologically independent low-grade and GBM patient samples n = 5, 8 respectively. Data is presented as mean ± SD. **c**, Median age of low-grade glioma and GBM patients at diagnosis from (**b**), data is mean ± SD. **d**, Kaplan-Meier survival analysis of low-grade glioma and GBM patients in (**b**). **e-j**, Flow cytometry of the indicated cells types as percent of CD45$^+$ or CD11b$^+$ cells based on gender (**e**,**f**) or anatomical location (**g-j**). No statistically significant differences were identified, unpaired t test two-tailed. Data is mean ± SD.

# nature research

Al Charest

# Reporting Summary

Nature Research wishes to improve the reproducibility of the work that we publish. This form provides structure for consistency and transparency in reporting. For further information on Nature Research policies, see our Editorial Policies and the Editorial Policy Checklist.

## Statistics

For all statistical analyses, confirm that the following items are present in the figure legend, table legend, main text, or Methods section.

| n/a | Confirmed | |
|---|---|---|
| ☐ | ☒ | The exact sample size ($n$) for each experimental group/condition, given as a discrete number and unit of measurement |
| ☐ | ☒ | A statement on whether measurements were taken from distinct samples or whether the same sample was measured repeatedly |
| ☐ | ☒ | The statistical test(s) used AND whether they are one- or two-sided<br>*Only common tests should be described solely by name; describe more complex techniques in the Methods section.* |
| ☒ | ☐ | A description of all covariates tested |
| ☒ | ☐ | A description of any assumptions or corrections, such as tests of normality and adjustment for multiple comparisons |
| ☐ | ☒ | A full description of the statistical parameters including central tendency (e.g. means) or other basic estimates (e.g. regression coefficient) AND variation (e.g. standard deviation) or associated estimates of uncertainty (e.g. confidence intervals) |
| ☐ | ☒ | For null hypothesis testing, the test statistic (e.g. $F$, $t$, $r$) with confidence intervals, effect sizes, degrees of freedom and $P$ value noted<br>*Give P values as exact values whenever suitable.* |
| ☒ | ☐ | For Bayesian analysis, information on the choice of priors and Markov chain Monte Carlo settings |
| ☒ | ☐ | For hierarchical and complex designs, identification of the appropriate level for tests and full reporting of outcomes |
| ☒ | ☐ | Estimates of effect sizes (e.g. Cohen's $d$, Pearson's $r$), indicating how they were calculated |

*Our web collection on statistics for biologists contains articles on many of the points above.*

## Software and code

Policy information about availability of computer code

| Data collection | Flow Cytometry: Beckman Coulter Gallios with acquisition software Kaluza (v1.1.3), BD LSR Fortessa with acquisition software FACSDiva (8.1), GraphPad Prism (GraphPad Software v9.3.0)<br>R (v3.6.3) |
|---|---|
| Data analysis | Sequencing Analysis<br><br>Flow Cytometry and<br><br>Patient Data Analysis: Flow cytometry quantification was performing using FLowJo v10.5.3 |

For manuscripts utilizing custom algorithms or software that are central to the research but not yet described in published literature, software must be made available to editors and reviewers. We strongly encourage code deposition in a community repository (e.g. GitHub). See the Nature Research guidelines for submitting code & software for further information.

## Data

Policy information about availability of data

All manuscripts must include a data availability statement. This statement should provide the following information, where applicable:
- Accession codes, unique identifiers, or web links for publicly available datasets
- A list of figures that have associated raw data
- A description of any restrictions on data availability

All data and materials used in the analysis are available in some form to any researcher for purposes of reproducing or extending the analysis. In rare instances, a

# Field-specific reporting

Please select the one below that is the best fit for your research. If you are not sure, read the appropriate sections before making your selection.

☒ Life sciences  ☐ Behavioural & social sciences  ☐ Ecological, evolutionary & environmental sciences

For a reference copy of the document with all sections, see nature.com/documents/nr-reporting-summary-flat.pdf

# Life sciences study design

All studies must disclose on these points even when the disclosure is negative.

| | |
|---|---|
| Sample size | For mouse studies: scRNA seq experiments, no statistical methods were used to predetermine the sample size. For all other mouse studies, a minimum of 3 biological replicates are included in each study. For survival studies, a minimum number of 4 mice are included in each arm. No statistical methods were used to predetermine sample sizes, and our sample sizes were similar to those reported in previous publications. For patient studies: no statistical methods were used to predetermine the sample size. As many patients as possible were included in the study. |
| Data exclusions | no data were excluded |
| Replication | The results from our scRNA-seq experiments were validated by flow cytometry of independent separate cohorts of mice. All of our experiments were performed with biologically independent replicates and were successful at replicating findings. |
| Randomization | Mice were randomly enrolled in treatment cohorts. |
| Blinding | scRNA seq samples were blinded to core facility technicians during sample processing. Individuals analyzing data for Fig 2b, c were blinded to the identity of the images analyzed. For all other experiments blinding was not possible because of the nature of the experimental set up. |

# Reporting for specific materials, systems and methods

We require information from authors about some types of materials, experimental systems and methods used in many studies. Here, indicate whether each material, system or method listed is relevant to your study. If you are not sure if a list item applies to your research, read the appropriate section before selecting a response.

## Materials & experimental systems

| n/a | Involved in the study |
|---|---|
| ☐ | ☒ Antibodies |
| ☐ | ☒ Eukaryotic cell lines |
| ☒ | ☐ Palaeontology and archaeology |
| ☐ | ☒ Animals and other organisms |
| ☐ | ☒ Human research participants |
| ☒ | ☐ Clinical data |
| ☒ | ☐ Dual use research of concern |

## Methods

| n/a | Involved in the study |
|---|---|
| ☒ | ☐ ChIP-seq |
| ☐ | ☒ Flow cytometry |
| ☒ | ☐ MRI-based neuroimaging |

## Antibodies

| | |
|---|---|
| Antibodies used | Flow cytometry antibodies used:<br>Anti Mouse<br><br>Purified CD16/32  BioLegend 101302<br>Lineage Cocktail FITC  BioLegend  133301<br>Ly-6A/E (Sca-1) PE-Fluor610 ThermoFisher Scientific  61-5981-82<br>CD127 (IL-7R?) Brilliant Violet 605  BioLegend  135041<br>CD117 (c-Kit) APC  BioLegend  105812<br>CD16/32 PerCP-Cyanine5.5  BioLegend  101323<br>CD34 Brilliant Violet 421  BioLegend  119321<br>CD45 APC/Cyanine7  BioLegend  103116<br>CD11b PE/Cyanine7  BioLegend  101216<br>F4/80 Brilliant Violet 605  BioLegend  123133<br>Ly-6G Brilliant Violet 421  BioLegend  127628<br>Ly-6C FITC  BioLegend  128006 |

I-A/I-E Alexa Fluor700  BioLegend  107622
CD11c Brilliant Violet 510  BioLegend  117353
CD3 FITC  BioLegend  100204
CD279 (PD-1) Brilliant Violet 421  BioLegend  109121
CD152 PE  BioLegend  106305
CD278 (ICOS) Brilliant Violet 650  BioLegend  313549
CD274 (BH-H1, PD-L1) PE/Cyanine7  BioLegend  124314
CD366 (Tim-3) PE/Cyanine7  BioLegend  134010
LIVE/DEAD Fixable Aqua Dead Cell Stain Kit  ThermoFisher Scientific   L34957
Arg1 APC BioLegend  17-3697-82
CCR7 PE BioLegend  120106
CD11b PE-CF594 BD Biosciences 562287
CD11b PerCP/Cy5.5 BD Biosciences 101228
CD206 PE-cy7 BioLegend  141720
CD3 BV510 BioLegend  100234
CD3 PerCPcy5.5 BioLegend  100218
CD335 PE Cy7 BioLegend  137618
CD4 PE CF594 BD Biosciences  562285
CD44 APC BioLegend  103012
CD45 AF700 BioLegend  103128
CD45 APC R700 BD Biosciences  565478
CD45 APC Cy7 BD Biosciences  557659
CD49d af488 BioLegend  103611
CD62L PE BioLegend  104408
CD8 BB515 BD Biosciences  564527
CD8 APC Cy7 BD Biosciences  557760
CD80 PE Cy7 BioLegend  104734
CD86 af488 BioLegend  105018
CTLA4 PE BioLegend  106306
EdU AF488 Invitrogen C10632
EGFR (AY13) PerCPcy5.5 BioLegend  352914
EGFR af488 BioLegend  352907
FOXP3 BV421 BioLegend  126419
Galectin 1 PE R&D systems IC1245P
Galectin 9 APC BioLegend  137912
Granzyme B PE Cy7 ThermoFisher Scientific  25-8898-82
Ki67 BV421 BioLegend  652411
Lag3 PE Cy7 BioLegend  125226
LY6C APC Cy7 BioLegend  128026
Ly6C APC BioLegend  128016
LY6G PE BioLegend  127608
Ly6G PEcy7  BioLegend  127618
MHCI PerCPcy5.5 BioLegend  107626
MHCII BV421 BioLegend  107632
P2ry12 APC BioLegend  848006
PD-1 PerCP Cy5.5 BioLegend  109120
PDGFRA (APA5) APC BD Biosciences 562777
PDL1 PE BioLegend  155404
PDL2 BV421 BioLegend  564245
Tim3  APC BioLegend  134008
TNF APC Cy7 BioLegend  506344
Viability Zombie Yellow BioLegend  423104
Viability Zombie NIR BioLegend  423106
Vista af488 BioLegend  143720

Anti Human

MRP-14 FITC  BioLegend  350703
CD49d  APC  BioLegend  304308
CD33 PE-CF594  BD Biosciences  562492
CD16 Alexa Fluor700  BioLegend  302025
HLA-DR-PE/Cy7  BioLegend  307615
CD15 PerCP/Cy5.5  BioLegend  323019
CD45 APC-H7  BD Biosciences  560274
CD14-VioBlue  Miltenyi Biotec  130-098-058
CD4 PE-CF594  BD Biosciences  562316
CTLA-4 PE  eBioscience  12-1529-42
CD45 Alexa Fluor700  Biolegend  368513
TIM3 (CD366) APC  Biolegend 345012
FOXP3 PE-Cy7  eBioscience  25-4777-42
Human TruStain FcX  Biolegend  422301
CD45 Alexa Fluor700  Biolegend  304023
HLA-DR FITC  BD Biosciences  562008
CD8 BB515  BD Biosciences  564526
CD3 BV510  BD Biosciences 563109
CD11b/Mac-1 PE  BD Biosciences  561001

PD-1 (CD279) PerCP-Cy5.5  BD Biosciences  561273
LAG-3 (CD223) PE/Cy7  Biolegend  369310
CD14 PE-Vio770  Milentyi Biotech  130-098-074

| Validation | All  anti mouse antibodies were validated and optimized on mouse tissues and all anti human antibodies were validated and optimized on human tissues by the source companies as they are all commercially available. All antibodies used were evaluated by the manufacturers as provided on their websites. |

# Eukaryotic cell lines

Policy information about cell lines

| Cell line source(s) | for HEK293T, source ATCC  catalog CRL3216 |
| Authentication | No authentication was applied as these cells are commercially available. |
| Mycoplasma contamination | Cells tested negative for mycoplasm using a sensitive commercial PCR assay (Lookout kit, MP0035) |
| Commonly misidentified lines (See ICLAC register) | n/a |

# Animals and other organisms

Policy information about studies involving animals; ARRIVE guidelines recommended for reporting animal research

| Laboratory animals | mus musculus, conditional transgenic for EGFR wild type, constitutive loss of Cdkn2a, conditional PTEN lox/lox, and conditional Lox-STOP-lox Luciferase on a mixed background. Both males and females 1:1 ratio of >6 weeks of age were used in these studies. |
| Wild animals | No wild animals were used in this research |
| Field-collected samples | No field-collected samples were used in this research |
| Ethics oversight | All mouse procedures were carried out in accordance with Beth Israel Deaconess Medical Center recommendations for care and use of animals and were maintained and handled under protocols approved by Institutional Animal Care and Use Committee (IACUC). |

Note that full information on the approval of the study protocol must also be provided in the manuscript.

# Human research participants

Policy information about studies involving human research participants

| Population characteristics | Human samples were obtained from glioma patients undergoing debulking resection surgery. 7 males and 6 females, ages ranging from 29 to 70 years old. |
| Recruitment | The human samples used in this study are human subject and protected under IRBs from BIDMC and MGH. Obtained informed consent were obtained under those IRBs. Human samples were de identified. Patients were recruited on basis of their survival surgical needs. There are no self selection bias or any other bias from recruitment as these surgical procedures are life saving and do not impact research. |
| Ethics oversight | BIDMC IRB and MGH IRB. |

Note that full information on the approval of the study protocol must also be provided in the manuscript.

# Flow Cytometry

## Plots

Confirm that:

☒ The axis labels state the marker and fluorochrome used (e.g. CD4-FITC).

☒ The axis scales are clearly visible. Include numbers along axes only for bottom left plot of group (a 'group' is an analysis of identical markers).

☒ All plots are contour plots with outliers or pseudocolor plots.

☒ A numerical value for number of cells or percentage (with statistics) is provided.

## Methodology

| Sample preparation | Cells from tumor tissues were isolated as described in the Methods section. After isolation, cells were analyzed for cell surface or intracellular markers using fluorophore-conjugated antibodies. Antibody staining was performed in 1X PBS and |

intracellular staining was performed using the eBioscience FoxP3/TF kit. Cells were stained for 30 min in dark then analyzed by flow cytometry as described in Methods. Compensation was performed using fluorescence minus one.

**Instrument**

Beckman Coulter Gallios
BD LSR Fortessa

**Software**

Acquisition software Kaluza (Beckman Coulter Gallios)
Acquisition software FACSDiva (BD LSR Fortessa)
Quantification software FlowJo v10.5.3

**Cell population abundance**

For scRNAseq CD45+ and CD45- populations were flow sorted and average viability was 66% and 87% respectively. Percent of CD45+ cells ranged between 6.8% to 44% with percent of CD45- 56% to 93.2%.

**Gating strategy**

Cells were first gated based on size using Forward and Side scatter, followed by identification of singlets using FSC-H and FSC-A.
To isolate cells for the single cell experiments, freshly dissociated GBM tumor cells were gated as follows for the sorted populations: Non-immune cells (CD45-), Immune cells (CD45+).

For validation and other flow cytometry experiments, freshly dissociated GBM tumor cells were gated as follows for these populations of cells:
EGFR positive cancer cells (CD45-EGFR+),
Macrophages CD45highCD11b+Ly6C-Ly6G-P2ry12-
Microglia CD45lowCD11b+Ly6C-Ly6G-P2ry12+
PMN-MDSCs CD45+CD11b+Ly6c+Ly6G+
M-MDSCs CD45+CD11b+Ly6c+Ly6G-
CD8+ T cells CD45+CD3+CD8+CD4-
CD4+ T cells CD45+CD3+CD4+CD8-
Regulatory T cells CD45+CD3+CD4+CD8-Foxp3+

Mouse bone marrow cells were gated for the following populations:
GMP: Lin-, Sca1-, CD127-, c-kit+, CD16/32+
CMP: Lin-, Sca1-, CD127-, c-kit+, CD16/32-
LK: Lin-, Sca1-, CD127-, c-kit+
LSK: Lin-, Sca1+, CD127-, c-kit+

Mouse spleen cells were gated as follows:
myeloid cells CD45+CD11b+,
T cells CD45+CD3+

For human glioma freshly dissociated tumor cells:
PMN-MDSC, neutrophil, granulocyte CD45+; CD11b+; Ly6G+; Ly6C+
M-MDSC, monocyte CD45+; CD11b+; Ly6G-; Ly6C+
Tumor associated Macrophages CD45+; CD11b+; Ly6G-; Ly6C-
CD4 T-cell CD45+; CD3+; CD4+
CD8 T-cell CD45+; CD3+; CD8+
CD8 Effector memory T-cell (TEM) CD45+; CD3+; CD8+; CD44+; CD62L-
CD8 Central memory T-cell (TCM) CD45+; CD3+; CD8+; CD44+; CD62L+
CD8 Naïve T-cell CD45+; CD3+; CD8+; CD44-; CD62L+
CD4 Effector memory T-cell (TEM) CD45+; CD3+; CD4+; CD44+; CD62L-
CD4 Central memory T-cell (TCM) CD45+; CD3+; CD4+; CD44+; CD62L+
CD4 Naïve T-cell CD45+; CD3+; CD4+; CD44-; CD62L+
Regulatory T-cell CD45+; CD3+; CD4+; Foxp3+

☒ Tick this box to confirm that a figure exemplifying the gating strategy is provided in the Supplementary Information.

