## [Peer Review File · Nature Immunology]

Peer Review Information

Journal: Nature Immunology

Manuscript Title: Single-Cell RNA sequencing reveals evolution of immune landscape during Glioblastoma progression

Corresponding author name(s): Al Charest

Editorial Notes:

**Redactions –
unpublished data**

Reviewer Comments & Decisions:

Decision Letter, initial version:

Subject: Decision on Nature Immunology submission NI-RS32188

Message: 13th Jul 2021

Dear Dr Charest,

Many thanks for your letter detailing how you would respond to the comments of the Referees- this was helpful. We would be willing to consider the manuscript further if the issues of the Referees are addressed as outlined in your letter. We would then send the manuscript back to the Referees for re-review. In order to help guide your revision I've marked up your response letter (see my comments in red- attached). I suspect however that carefully articulating the key advances of this study will be critical to secure an endorsement from these Refs.

We hope you will find the referees' comments useful as you decide how to proceed. If you wish to submit a substantially revised manuscript, please bear in mind that we will be reluctant to approach the referees again in the absence of major revisions.

If you choose to revise your manuscript taking into account all reviewer and editor comments, please highlight all changes in the manuscript text file.

* If you have not done so already please begin to revise your manuscript so that it conforms to our Resource format instructions at <http://www.nature.com/ni/authors/index.html>. Refer also to any guidelines provided in this letter.

The Reporting Summary can be found here:
<https://www.nature.com/documents/nr-reporting-summary.pdf>

When submitting the revised version of your manuscript, please pay close attention to our [href="https://www.nature.com/nature-research/editorial-policies/image-integrity">Digital Image Integrity Guidelines. and to the following points below:](https://www.nature.com/nature-research/editorial-policies/image-integrity)

You may use the link below to submit your revised manuscript and related files:
[REDACTED]

If you wish to submit a suitably revised manuscript we would hope to receive it within 6 months. If you cannot send it within this time, please let us know. We will be happy to consider your revision so long as nothing similar has been accepted for publication at Nature Immunology or published elsewhere.

Nature Immunology is committed to improving transparency in authorship. As part of our efforts in this direction, we are now requesting that all authors identified as 'corresponding

author' on published papers create and link their Open Researcher and Contributor Identifier (ORCID) with their account on the Manuscript Tracking System (MTS), prior to acceptance. ORCID helps the scientific community achieve unambiguous attribution of all scholarly contributions. You can create and link your ORCID from the home page of the MTS by clicking on 'Modify my Springer Nature account'. For more information please visit please visit www.springernature.com/orcid.

Thank you for the opportunity to review your work.

Sincerely,

Zoltan Fehervari, Ph.D.
Senior Editor
Nature Immunology

The Macmillan Building
4 Crinan Street
Tel: 212-726-9207
Fax: 212-696-9752
z.fehervari@nature.com

Referee expertise:

Referee #1: GBM, cancer immunology

Referee #2: -omics, cancer immunology

Reviewers' Comments:

Reviewer #1:

Remarks to the Author:

Deleuze et al utilized genetically engineered GBM mouse models and scRNA seq to identify the cell composition in immune and non-immune (CD45+ and CD45-) populations. The authors then evaluated the effect of TMZ/IR treatment on myelopoiesis and the immune microenvironment in non-tumor and GBM bearing mice. Finally, the immune composition in low-grade and GBM patients was determined using flow cytometry. Most concerns surround the novelty of the work. Major findings from this manuscript largely confirm the previously published work. Examples include PMC7083893, PMID:33782623, PMID: 32669424, PMC7411074, and others are described below. Further, mouse data does not necessarily match with the patient data presented in Figure 8. This work would have been more appealing/exciting if the authors had compared the glioblastoma immune microenvironment across commonly found genetic alterations such as PTEN, PDGF, TP53, IDH, and NF1 or molecular subtypes. However, some major and minor suggestions for improving the impact of the manuscript are described below:

Figure 1

What is the necessity of this Figure? Figure 5A shows the t-SNE plots and cell composition present at distinct stages of GBMs. The data in this Figure highlights the same scRNA-seq information obtained by pooling all the cells from three distinct stages of GBM.

- As using only BLI imaging (Jost et al PMID: PMC3831362) is not always a reliable method for determining the size of the brain tumor. Was any other method used to measure tumor volume? This is especially important as all the control mice are dead in less than 25 days post tumor detection.
- Statistical description in the figure legends and materials and methods is lacking. It is clear for S1 and S2, the samples were pooled from independent tumors. However, what is not clear is whether scRNA-seq was performed on one sample (n=1) or multiple samples from each stage. Please add number of samples used for each experiment in the text.
- Figure 1B, what is the identity of the cells present in the cluster 4 and 13? What cell types are present in these clusters? At least label the clusters with the top marker genes present.
- In figure 1B, what is the reason for identifying cluster 11 as T cells/NK cells? These two cell types express distinct marker genes and can be easily separated? Was flow cytometry performed to confirm the existence of T and NK cells in CD45+ sorted cells?
- Are there any Myeloid-Derived Suppressor Cells (MDSCs) present?
- Top differentially expressed genes (1C and 1E) that were used to identify the cell types in each of the clusters shown in Figure 1B (both CD45+ & CD45-) are not readable.

Figure 2

- Can authors point out in the IHC images (Supplementary Figure 2) how they define a cluster? As written and shown, it looks like a single mass at day 14 & day 21.
- It is interesting to find two clusters of tumor cells in mouse models and GBM patients. How do these two clusters compare to CD45+ cluster 14 cells from Figure 1?
- Is it possible to stain for the top markers from clusters 7 and 14 in their mouse/human tumor slides and see if the hypothesis about regional differences is true, and can be spatially resolved?
- Can the authors comment on the reason/consequence of activating two different EGFR ligands in tumor cells? Can it be because of their location (tumor core vs margin)? Further, can they test if different ligand activation also leads to different downstream signaling pathway activation in GBM cells? For example, look for enriched pathways by running GSEA, DAVID, or Metascape on differentially expressed genes in clusters 2 and 7. Alternatively, treat mouse GBM cells with different ligands and look at the expression of downstream targets such as AKT, PLC, STAT3 etc. If there is difference, then validate the findings in mouse and human GBM slides.

Figure 3

- Number of studies have indicated that late stage GBM tumors are populated with higher number of blood derived macrophages (BDMs) compared to microglia (PMCID: PMC4872227, PMID:25580734). How do the authors reconcile the presence of highly proliferating microglia at the late-stage GBMs in their study? Is this unique to their GBM mouse model? Are these three different microglia subsets present in human GBM patients? Was any other GBM mouse model used to validate these findings?
- Can pseudo time analysis (monocle or slingshot) be performed to determine if these proliferating microglia cells give rise to either cluster 2 or 5? Also, see if cluster 2 can give rise to cluster 5 or vice versa.
- Do these inflammatory microglia have upregulation of antigen presentation genes? Also, is there any evidence of microglial phagocytosis (upregulation of known phagocytosis genes)?
- Recently Khalsa et al reported the presence of 3 different microglia population in mouse

GBMs using scRNA-seq. How does the microglia identified in this manuscript compare to them and other published studies (PMID:30606613, PMID: 31772305)?

- Can authors describe the reason for the existence of less inflamed microglia subset? Do these microglia cells expand in late-stage GBMs? Are these involved in resolving the inflammation caused by cells in cluster 2?
- What are the molecular features of the microglia present in cluster 4 of CD45⁻ cells? Are they proliferative, pro or anti-inflammatory?

Figure 4

- One of the major ligands produced by tumor cells in EGFR amplified GBM (PMID: 30401716) to recruit macrophages is CCL2. Further, CCL2 is also produced by immune cells in glioma (PMID: 27530322). It's surprising that authors don't find the expression of CCL2 in either CD45⁺ or CD45⁻ cells. Can they explain the reason for this?
- TNF feature plots from figure 3h are repeated here. Remove this and refer to figure 3h.
- Log₂CPM values are very low for all the immune checkpoint molecules shown. Usually, genes with these low counts are filtered out in RNA-seq data sets. Were the findings from scRNA-seq validated using qPCR or flow cytometry?
- Overall, based on the previously published studies, it's not surprising to find that cytokines, immune checkpoint molecules are expressed on CD45⁺ immune cells (PMID:33853689 PMID: 28115578, PMID:30824583, PMID:30498094)

Figure 5

- This should have been Figure 1 in the manuscript.
- For extended figure 5A, run statistics and include error bars.
- Please include the mouse survival data. Also, include a figure that shows the days post tumor implantation at which the early, mid and late-stage tumors were harvested.
- In addition to the dye, did the authors perform an DCE-MRI to visualize the blood brain barrier disruption?
- Overall, the findings that increase in macrophages, lymphocytes, monocytes, NK cells and decrease in microglia confirm some of the previously published studies as mentioned above.

Figure 6

- What is the reason for comparing only the early and late-stage macrophages and microglia in this figure? Also, are there any differences when comparing middle vs late or middle vs early-stage microglia and macrophages?
- Was comparison of microglia cluster 6 early vs late performed? Also, stage specific comparison of CD45⁻ microglia is missing.
- Cell count matrix in the form of table would be helpful in determining if there is an expansion/decrease in specific microglial cluster over time.

Figure 7

- The axis and the percentage of cells in each quadrant are not readable for the flow plots provided in extended data 6B and 6D
- How do the authors know if TMZ reached the brain in non-tumor bearing mice?
- Labeling and representation of Figure 7B could be better for easy understanding. What is the reason for decrease of macrophages?
- What is the effect of TMZ on T-cells in extended data Figure 6E? Further, statistics is missing for this Figure.
- For extended data Figure 6H, What is the effect of TMZ treatment on MDSCs in peripheral blood?
- If ~10% of MDSCs are detected by flow cytometry in Figure 7D, why is it not detected in

the scRNA-seq data?

- Recently, Akkari et al, using mouse glioma models showed that macrophage and microglia ratios don't change following radiation compared to untreated mice. However, Figure 7F shows that there is a significant decrease in microglia and macrophages following IR. Can authors explain the reason for this?
- Why does the TMZ/IR treatment decrease the survival of mice compared to IR alone in Figure 7E? Is that significant or labeling is incorrect? Perform statistics comparing different groups.

Figure 8

- What is the EGFR status of the patients analyzed? What is the reason to include IDH WT and mutant patients? This doesn't really fit with the premise of this manuscript.
- From Fig 7C, median age for low-grade was 45 days where as for GBM it is 55? Will aging part from tumor-grade contribute to the observed differences in immune cell infiltrates?
- Data from the mouse model suggests that accumulation of Tregs (Fig. 6E) and MDSCs (Fig. 7E) contributes to the immune suppression observed in late stage GBMs (Fig. 5C). However, there are no significant differences in MDSCs, neutrophils and Tregs while comparing low-grade and GBM patients. Can authors explain the discrepancy for this?

Minor comments

- In the methods section for single cell pathway analysis describe the databases that were used
- Look at the order of the figures. Rearrange to better present the logical flow. For example, supplemental Figure 5A should have been in the supplemental Figure 1A.
- Mention the age of the control mice that received TMZ treatments in extended Figure 6C.
- Make the flow plots more readable. The axis and percentages are not clearly visible.
- Statistics is missing for most of the figures included in supplementary.

Reviewer #2:

Remarks to the Author:

The manuscript entitled "Single-cell RNA sequencing reveals evolution of immune landscape during glioblastoma progression" described the major cellular components during the glioblastoma progression by using single-cell sequencing in a mouse model. They also studied the cellular changes after temozolomide and radiation therapies, providing potential mechanistic insights into glioblastoma therapy. Finally, they examined the immune cell populations in human samples. The manuscript may be a good resource for the research field. However, the work is a descriptive in nature, lacking mechanistic insights into glioblastoma etiology, immunity, and therapy. Furthermore, several GBM scRNA-seq datasets, including immune cellular landscape, have been published. Nonetheless, the cellular changes observed in the course of tumor progression and under different therapies may be useful and informative in designing novel preclinical and clinical studies on glioblastoma.

Major comments:

1. The single-cell sequencing was conducted in a single tumor in the late stage (S3), two tumors in early stage (S1) and mid-stage (S2). This design has its inherited limitation.
2. The paper described the major cellular components, including CD45+ immune cells and CD45- non-immune cells, and their dynamic changes in the tumor microenvironment (TME). However, the biological importance of different populations and the paired ligand-

receptor interaction is unknown.

2.1 Tumor cells: Two clusters of tumor cells were identified. Different functional signatures were observed. a. Regionalization of these two populations was hypothesized, but no analysis has been done in the mouse model, although some data from patient samples support this. It is feasible and should be validated at least in the mouse model. Besides, it was hypothesized that the distinct gene expression profiles observed are influenced by their localization or regionalization, the authors provided no analysis and evidence, supporting this possibility. b. Different EGFR ligands were expressed by the two different tumor clusters. Based on the scRNA-seq data, are we able to discriminate whether these ligands are functional in autocrine or paracrine ways? Since different ligands can exert different effects on downstream signaling, does the downstream signaling of EGFR show difference between these two clusters of tumor cells? If yes, does the difference of EGFR signaling drive the distinct expression profile since EGFR is a tumor driver mutation. c. Are these two clusters developmentally different? Are they two independent clusters in tumor initiation? Or one cluster is originated/developed/reprogrammed from the other one? For scRNA-seq data, pseudotime and RNA-velocity trajectory analysis may provide insights into this.

2.2 Microglia

a. Cluster-4 was also defined as microglia, but it was not discussed at all.

Does CD45⁻ microglia have different function comparing to CD45⁺ microglia?

b. Three CD45⁺ microglia clusters (C-2/5/6) were observed. C-6 is a proliferating cluster and seems to be stable regarding its proportion from early to late stage. However, C-2 and C-5 were decreased. How did this happen?

2.3 Macrophage

a. Increased macrophage population was observed at mid-S2 and late-S3 stage. Are they monocyte-derived macrophages (e.g. CCR2 expression)?

b. Macrophage population shows high heterogeneity (Figure 5A). This heterogeneity and the dynamic changes were not discussed.

c. The study proposed that "macrophage infiltration coincides with BBB disruption". However, the disruption of BBB occurred at a later stage (Figure 5F,G), rather than early/mid stage (Figure 5A-E).

3. The current analysis looked at cytokines and cytokine receptors, as well as checkpoint molecules and checkpoint receptors.

a. Was the analysis able to provide information on which pair(s) of molecular interaction and cellular interaction, play critical role(s) in the TME and during tumor progression. Interactome analysis (e.g. PMID: 31675496, 32302573) may provide insights regarding this part.

b. Figure 4A may be a biased analysis, how were the 37 cytokines selected?"

c. For checkpoint related analysis, myeloid cells (DCs and macrophages) express checkpoints. The present analysis demonstrated high levels of checkpoint molecules in myeloid cells. However, the analysis reflects gene expression per cell, the quantity of different cell subsets in the TME should be at least discussed.

4. The study described the changes in immune cell subsets in the TME, following different therapeutic treatments. However, whether and how immune cell contribute to the outcome is unknown.

5. The analysis has several potential therapeutic applications. But none of them were

tested experimentally.

Other specific comments:

6. At which time points Early-S1, Mid-S2, Late-S3 samples were collected?
7. Figure 1C, E, the resolution is not enough to read the individual genes.
8. Figure 5D is not consistent with BLI data in Fig 5A. BLI data shows strong signal in mid-S2 stage, while not much change in mid-S2 to late-S3 stage. Please provide explanation.
9. Figure 5E, Y-axis is different. Is it mislabeled?
10. Figure 6D, M2 markers are not consistent with the gene expression analysis which show upregulated M2 signature at a late stage.
11. Figure 7D, please check the sample size since it's inconsistent.
12. Please call the figures out in order. E.g. Figure 6A,C, B
13. Figure 7A, B, T cells are increased, are they conventional T cells or Treg cells (CTLA4+)? Treg cells are increased as shown in Figure 6E.

Author Rebuttal to Initial comments

Reviewer #1

Most concerns surround the novelty of the work. Major findings from this manuscript largely confirm the previously published work. Examples include PMC7083893, PMID:33782623, PMID: 32669424, PMC7411074, and others are described below.

We would like to point out that although these studies are of high merit, they all used artificial models that do not recapitulate the properties of human GBM. Most are based on the implantable GL261 tumor model that 1) not only has no molecular similarities with human GBM since it carries a RAS mutation, a molecular feature never found in human GBM (<1%) and 2) are syngeneic allografts, which by nature completely prevents studies on the evolution of GBM from inception to growth progression in an ever-evolving tumor immune microenvironment. While we also report similar immune-related features as the GL261 model in our end-stage samples, we also report on the evolution of how these features developed over time. To our knowledge, our study is the first to employ a longitudinal approach to show how disease evolution correlates with transcriptome changes of immune populations at a single level in GBM, because such studies cannot be performed using syngeneic models.

Note that PMC7083893 by von Roemeling et al., isn't relevant to the concern of the reviewer since it doesn't contain scRNAseq data. PMID:33782623 by Antunes et al., had just been published at the time of submission of the present manuscript. We have now leveraged some of their observations to better define cell populations within our scRNAseq datasets. PMID: 32669424 by Akkari et al., did not perform scRNAseq. Furthermore, we address the divergence

between our results and theirs in terms of radiation- and temozolomide-induced changes in the tumor immune microenvironment. PMC7411074 from Khalsa et al., performed CyTOF on syngeneic models which does not allow for in depth analysis of transcriptomes of the various immune cells. Also, longitudinal studies aimed at deciphering evolutionary changes in transcriptomes during GBM growth are not feasible in these syngeneic models.

Further, mouse data does not necessarily match with the patient data presented in Figure 8.

The main focus of our study is the molecular profile of the TME related to GBM evolution and acquisition of a gradually aggressive and progressive nature. We believe that this process is most accurately recapitulated by comparing human low-grade glioma vs. GBM, two forms of the same type of malignancy characterized by different level of cancer grade. This is highlighted in our conclusion from Fig. 8 demonstrating that low-grade gliomas are mostly composed of microglia whereas GBM tumors are predominantly composed of macrophages, features that we also observe in our mouse model.

This work would have been more appealing/exciting if the authors had compared the glioblastoma immune microenvironment across commonly found genetic alterations such as PTEN, PDGF, TP53, IDH, and NF1 or molecular subtypes.

What the reviewer is considering as more “appealing/exciting” cannot be performed without a detailed characterization of the TME during disease evolution, which is the focus of our present study. Any conclusions made by assessing the immune microenvironment across different genetic alterations might simply be a consequence of different level of cancer growth and progression and not a difference related to the distinct molecular features of GBM. This might be an appropriate future research goal but it is unrealistic to be part of the present study, which is already very dense and extensive.

However, some major and minor suggestions for improving the impact of the manuscript are described below:

Figure 1

- What is the necessity of this Figure? Figure 5A shows the t-SNE plots and cell composition present at distinct stages of GBMs. The data in this Figure highlights the same scRNA-seq information obtained by pooling all the cells from three distinct stages of GBM.

We have modified Figure 1 based on the new tumor samples we added to the analyses. The purpose of this figure is to provide basic information about the experimental model used in the

study and sets the stage for the subsequent assessments. It defines the various clusters we identified using cell specific markers.

- As using only BLI imaging (Jost et al PMID: PMC3831362) is not always a reliable method for determining the size of the brain tumor. Was any other method used to measure tumor volume? This is especially important as all the control mice are dead in less than 25 days post tumor detection.

We have previously demonstrated tumor volume increases as measured by BLI correlates to those measured using MRI (Zhu et al., PNAS, 2009). In the revised manuscript we added new data (Fig. 1b) demonstrating side-by-side the relationship between BLI and MRI-derived GBM volumes. We demonstrate that BLI is more sensitive than MRI in detecting early phases of intracranial growth and therefore is an appropriate method to use to determine GBM growth characteristics. Here, all our studies and comparisons among our groups were performed using the exact same method (BLI) for assessment of tumor growth. Thus, our conclusions are based on the same approach for assessment of tumor size. We chose BLI as our method because this provides the most accurate, reliable, fast and reproducible for longitudinal evaluation of tumor growth in live animals.

- Statistical description in the figure legends and materials and methods is lacking. It is clear for S1 and S2, the samples were pooled from independent tumors. However, what is not clear is whether scRNA-seq was performed on one sample (n=1) or multiple samples from each stage. Please add number of samples used for each experiment in the text.

We apologize for the lack of clarity. We have added detailed information regarding the number of mice/tumors analyzed in the manuscript as Extended Data Table 1.

- Figure 1B, what is the identity of the cells present in the cluster 4 and 13? What cell types are present in these clusters? At least label the clusters with the top marker genes present.

This information is no longer necessary in our upgraded analysis. All clusters are clearly identified.

- In figure 1B, what is the reason for identifying cluster 11 as T cells/NK cells? These two cell types express distinct marker genes and can be easily separated? Was flow cytometry performed to confirm the existence of T and NK cells in CD45+ sorted cells?

CD8 T cells and NK cells share many common markers. In our updated larger dataset, CD8 T cells and NK cells cluster together. This is not uncommon as this phenomenon is observed in several scRNAseq studies. Not related to this project, [REDATED].

- Are there any Myeloid-Derived Suppressor Cells (MDSCs) present?

MDSC are present and are shown in the UMAP (cluster 11) and in flow cytometry.

- Top differentially expressed genes (1C and 1E) that were used to identify the cell types in each of the clusters shown in Figure 1B (both CD45+ & CD45-) are not readable.

We apologize for providing low resolution of these figures and we have provided high-resolution images in this resubmission.

Figure 2

- Can authors point out in the IHC images (Supplementary Figure 2) how they define a cluster? As written and shown, it looks like a single mass at day 14 & day 21.

We have indicated these clusters with arrows and circles in revised Extended Data Fig. 2a.

- It is interesting to find two clusters of tumor cells in mouse models and GBM patients. How do these two clusters compare to CD45+ cluster 14 cells from Figure 1?

Our new version now shows EGFR+ clusters only in the CD45 negative population.

- Is it possible to stain for the top markers from clusters 7 and 14 in their mouse/human tumor slides and see if the hypothesis about regional differences is true, and can be spatially resolved?

Our new analysis demonstrates that there are now 5 clusters of EGFR+ cells (in revised Fig. 1c,e,f), which can be grouped into 3 groups based on similarities with human scRNAseq subtyping. Spatially resolving these using IHC with specific markers is a daunting task and is beyond the scope of the manuscript. However, to address the reviewer's comment, we have identified two markers from the scRNAseq dataset (Pdgfra and Gal-1) that are enriched in clusters (0 and 10) and 5 respectively and performed bulk RNAseq on flow sorted PDGFRA+ and

GAL-1+ cells to determine transcriptomes and pathway differences between these two populations (in revised Fig. 2g-m).

- Can the authors comment on the reason/consequence of activating two different EGFR ligands in tumor cells? Can it be because of their location (tumor core vs margin)? Further, can they test if different ligand activation also leads to different downstream signaling pathway activation in GBM cells? For example, look for enriched pathways by running GSEA, DAVID, or Metascape on differentially expressed genes in clusters 2 and 7. Alternatively, treat mouse GBM cells with different ligands and look at the expression of downstream targets such as AKT, PLC, STAT3 etc. If there is difference, then validate the findings in mouse and human GBM slides.

We have done a few things to address this comment. First, we have performed GO analysis on these clusters and report the data from our extended dataset. We have furthered these analyses by using additional bioinformatics tools as suggested by the reviewer and reported the results in Fig. 2. To get more depth in sequencing, we flow sorted EGFR positive cells using differentially expressed markers (PDGFRA and GAL-1) and performed RNAseq on these and additional bioinformatics analyses to determine changes in gene expression as a result of differences in signaling networks downstream of EGFR (Fig. 2k).

We feel that performing signaling studies (phosphoAKT, phosphoERK1/2, phosphoSTAT3 etc) in situ, as the reviewer is proposing are not only beyond the scope of this paper but also technically very challenging. We suspect that the changes in signaling between an HB-EGF stimulated EGFR and a TGFa-stimulated EGFR will not be detected by conventional methods (IF or IHC, which are not quantitative) and would require quantitative phosphoproteomic approaches from freshly excised and flow sorted cells, which may lose their signaling details during isolation procedures. Thus, such data cannot be generated for the needs of the present study.

Figure 3

- Number of studies have indicated that late stage GBM tumors are populated with higher number of blood derived macrophages (BDMs) compared to microglia (PMCID: PMC4872227, PMID:25580734). How do to the authors reconcile the presence of highly proliferating microglia at the late-stage GBMs in their study? Is this unique to their GBM mouse model? Are these three different microglia subsets present in human GBM patients? Was any other GBM mouse model used to validate these findings?

We have characterized the nature of the proliferating microglia. We would like to point out that in our updated analysis, cycling microglia cells are now present in all 3 microglia clusters instead

of one discrete cluster as was the case in our initial analysis included in the original manuscript. To better describe these cycling microglia, we flow sorted EdU labeled GBM microglia and have performed bulk RNA sequencing. We added these new results and analyses in the revised manuscript (Fig. 3). We revised our longitudinal flow analysis of GBM tumors and show a relative increase in the number of microglia during GBM progression (revised Fig. 5f), which may result from the proliferative microglia population. We can't comment on the uniqueness of this proliferative population in our model because no studies (either mouse or human) have reported on cycling parameters of GBM microglia.

- Can pseudo time analysis (monocle or slingshot) be performed to determine if these proliferating microglia cells gives rise to either cluster 2 or 5? Also, see if cluster 2 can give rise to cluster 5 or vice versa.

We have used monocle3 to determined pseudotime between different populations of cells. We observed that microglia cluster 4 appears to give rise to cluster 2. The results are shown in Fig. 2d.

- Do these inflammatory microglia have upregulation of antigen presentation genes? Also, is there any evidence of microglial phagocytosis (upregulation of known phagocytosis genes)?

We have determined that indeed one of the microglial clusters has increased phagocytosis. This data was added as part of Fig. 3i and Extended data Fig. 3j and discussed in the text.

- Recently Khalsa et al reported the presence of 3 different microglia population in mouse GBMs using scRNA-seq. How does the microglia identified in this manuscript compare to them and other published studies (PMID:30606613, PMID: 31772305)?

The data generated in Khalsa et al., is from CyTOF analysis and is not amenable to compare transcriptomes with our datasets. We utilized the datasets from Li et al., (PMID:30606613) to compare to our flow sorted EdUpos and EdUneg microglia and normal adult microglia and contralateral microglia. Since Li et al., utilized only normal brains and not GBM brains, we could only perform a comparison of their dataset to our normal brain control scRNAseq. We believe that this would not be informative and relevant to the scope of the present study.

- Can authors describe the reason for the existence of less inflamed microglia subset? Do these

microglia cells expand in late-stage GBMs? Are these involved in resolving the inflammation caused by cells in cluster 2?

When we expanded the number of GBM in our new analysis, we did not observe the existence of less inflamed microglia subset.

- What are the molecular features of the microglia present in cluster 4 of CD45- cells? Are they proliferative, pro or anti-inflammatory?

There is no longer a microglia cluster in the CD45- population in our updated analysis.

Figure 4

- One of the major ligands produced by tumor cells in EGFR amplified GBM (PMID: 30401716) to recruit macrophages is CCL2. Further, CCL2 is also produced by immune cells in glioma (PMID: 27530322). It's surprising that authors don't find the expression of CCL2 in either CD45+ or CD45- cells. Can they explain the reason for this?

We would like to point out that CCL2 expression is detected in both qPCR and scRNAseq data (Fig. 4a,b, Fig. 6c).

- TNF feature plots from figure 3h are repeated here. Remove this and refer to figure 3h.

We have addressed this issue in this revision.

- Log2CPM values are very low for all the immune checkpoint molecules shown. Usually, genes with these low counts are filtered out in RNA-seq data sets. Were the findings from scRNA-seq validated using qPCR or flow cytometry?

*Key findings were validated by qPCR and flow cytometry. The data is in the manuscript (Fig. 4h and **Extended Data Fig. 4**).*

- Overall, based on the previously published studies, it's not surprising to find that cytokines, immune checkpoint molecules are expressed on CD45+ immune cells (PMID:33853689 PMID: 28115578, PMID:30824583, PMID:30498094)

We agree that it is becoming increasingly understood that checkpoint ligands are mainly expressed in immune cells. However, the general concept persists that checkpoint immunotherapy blocks the interaction between checkpoint receptors expressed on T cells and their ligands expressed in cancer cells. As our model is the first to most closely recapitulate human GBM, it is important to provide this information clearly and explicitly.

Figure 5

- This should have been Figure 1 in the manuscript.
- For extended figure 5A, run statistics and include error bars.
- Please include the mouse survival data. Also, include a figure that shows the days post tumor implantation at which the early, mid and late-stage tumors were harvested.

-It is important that we define the cell types in UMAP clusters early in the manuscript in Fig. 1. Pooling of all samples is necessary to achieve this. Fig. 5 pertains to the evolution of tumorigenesis from normal brain to Early to Late stage GBMs and required a prior establishment of the UMAP clusters (from Fig. 1) to ascertain flux of cells during progression.

-Extended Fig. 5a has been updated with additional data.

- Details pertaining to times of tumor harvest post initiation (incl. survival of the mice from which the Late samples were harvested) are included in Extended Data Table 1. We would like to point out that our model doesn't involve tumor implantation but rather tumor initiation through de novo ablation of tumor suppressor genes and activation of EGFR expression (see introduction for details).

- In addition to the dye, did the authors perform an DCE-MRI to visualize the blood brain barrier disruption?

We did not perform Dynamic Contrast-Enhanced MRI. DCE MRI is an active research area with no standardized sets of analysis parameters. Very few US laboratories are expert in this area and we were not able to establish a collaboration within the time frame of resubmission. Moreover, tail vein catheterization of animals is required for DCE MRI and therefore limits acquisition of

time point evaluation to only two readings (mice only have two tail veins and catheterized tail veins collapse post removal of catheter, making them unusable for subsequent use). Instead, we opted for a constant timed contrast enhanced MR imaging, which allowed us to image the same animals more than twice and to normalize MRI time acquisition. These results are included in Fig. 5j and detailed in Supplementary methods section. More importantly, in doing so, we were able to validate our previous observations of the timing of BBB disruption.

- Overall, the findings that increase in macrophages, lymphocytes, monocytes, NK cells and decrease in microglia confirm some of the previously published studies as mentioned above.

We respectfully disagree with the reviewer. There is no study demonstrating dynamic changes in TME composition during GBM growth and evolution. Syngeneic models, by nature, are not models that can be studied dynamically. Human patient samples are from one time point only. The GLASS consortium has recently published data on profiling of primary and recurrent GBM pairs but only from bulk RNAseq material. There remain still no extensive studies of GBM TME evolution under native conditions of GBM initiation and progression. Our model is the first of its kind and every aspect of its biology is studied and reported for the first time.

Figure 6

- What is the reason for comparing only the early and late-stage macrophages and microglia in this figure? Also, are there any differences when comparing middle vs late or middle vs early-stage microglia and macrophages?
- Was comparison of microglia cluster 6 early vs late performed? Also, stage specific comparison of CD45- microglia is missing.
- Cell count matrix in the form of table would be helpful in determining if there is an expansion/decrease in specific microglial cluster over time.

With our updated analysis, we only compared Early vs Late since we did not process mid stage tumors. There is a cell count matrix already available in Extended data Table 2. At the discretion [REDACTED].

Figure 7

- The axis and the percentage of cells in each quadrant are not readable for the flow plots provided in extended data 6B and 6D
- How do the authors know if TMZ reached the brain in non-tumor bearing mice?

- Labeling and representation of Figure 7B could be better for easy understanding. What is the reason for decrease of macrophages?
- What is the effect of TMZ on T-cells in extended data Figure 6E? Further, statistics is missing for this Figure.
- For extended data Figure 6H, What is the effect of TMZ treatment on MDSCs in peripheral blood?
- If ~10% of MDSCs are detected by flow cytometry in Figure 7D, why is it not detected in the scRNA-seq data?

We apologize for the poor quality of Extended Data Fig. 6b,d. (now Extended Data Fig. 7). We have replaced with clearer figures.

Published PK/PD studies of TMZ showed that TMZ penetrates intact brain.

Labelling and representation of Fig. 7b has been improved. The decrease in macrophage could be a reflection of the increases in other myeloid populations which relatively speaking, decrease the percent of macrophages.

We discuss in the manuscript potential reason(s) for the decrease in macrophages observed in spleen of GBM bearing mice.

*The effects of TMZ on T cells is represented in Fig. 7f. We didn't look at T cell specifically in Extended Data Fig. 6E (which is now 7e). Statistics are present in Extended Figure 6E (now 7e). The figure legend reads "Unless otherwise noted, no significant statistical differences exist in pairwise comparisons of control and TMZ treated. Data is mean \pm S.D., $n \geq 3$, * $p < 0.0002$, ** $p < 0.0001$, *** $p < 0.03$."*

We have not determined the effect(s) of TMZ on MDSCs from peripheral blood.

We would like to point out that PMN-MDSC are detected in scRNA seq.

- Recently, Akkari et al, using mouse glioma models showed that macrophage and microglia ratios don't change following radiation compared to untreated mice. However, Figure 7F shows that there is a significant decrease in microglia and macrophages following IR. Can authors explain the reason for this?

There are several reasons why Akkari's data differs from ours. Radiation treatment dosage were different. Also, their model is genetically distinct from ours. It is entirely possible and likely that the genotype of the GBM tumor affects the immune response to IR.

- Why does the TMZ/IR treatment decrease the survival of mice compared to IR alone in Figure 7E? Is that significant or labeling is incorrect? Perform statistics comparing different groups.

Although it appears that TMZ/IR decrease survival of mice, this is not statistically significant. Statistics comparing control to TMZ/IR and control to IR are already present in the figure legend, we added stats on TMZ/IR to IR.

Figure 8

- What is the EGFR status of the patients analyzed? What is the reason to include IDH WT and mutant patients? This doesn't really fit with the premise of this manuscript.

We omitted the EGFR status because it is known for only some of the non-IDH1 mut patients. We felt that a partial data list would not be very informative. We can amend our patient table if necessary. 80% of low grade gliomas are IDH mutated and almost completely mutually exclusive of EGFR amplification. It is difficult to obtain many IDH1 WT low grade gliomas. The premise of the manuscript is to compare low grade to high grade GBMs and the majority of low-grade gliomas are IDH mutated.

- From Fig 7C, median age for low-grade was 45 days where as for GBM it is 55? Will aging part from tumor-grade contribute to the observed differences in immune cell infiltrates?

Changes in immune subsets occur only in aged mice. Our mice do not belong to this group at any time of the study and our experimental design is not purposed to address this question.

- Data from the mouse model suggests that accumulation of Tregs (Fig. 6E) and MDSCs (Fig. 7E) contributes to the immune suppression observed in late stage GBMs (Fig. 5C). However, there are no significant differences in MDSCs, neutrophils and Tregs while comparing low-grade and GBM patients. Can authors explain the discrepancy for this?

Although the mouse data show changes in other cell types during progression, our patient sample data suggests that macrophages are the relevant cell type establishing an immunosuppressed GBM microenvironment.

Minor comments

- In the methods section for single cell pathway analysis describe the databases that were used
- Look at the order of the figures. Rearrange to better present the logical flow. For example, supplemental Figure 5A should have been in the supplemental Figure 1A.
- Mention the age of the control mice that received TMZ treatments in extended Figure 6C.
- Make the flow plots more readable. The axis and percentages are not clearly visible.
- Statistics is missing for most of the figures included in supplementary.

We have included these changes throughout the manuscript. Some of these requests are now deprecated by our updated analysis. Some of the figures appear to have missing statistics, however, in the figure legends, we mention “Unless otherwise noted, no significant statistical differences exist in pairwise comparisons”. We avoided cluttering the graphs with unnecessary “n.s.” not significant labels.

Reviewer #2

(Remarks to the Author)

The manuscript may be a good resource for the research field. However, the work is a descriptive in nature, lacking mechanistic insights into glioblastoma etiology, immunity, and therapy.

The work is a characterization of the immune landscape, not meant to be an identification of mechanisms that shape the immune landscape in TME. By nature, scRNA seq studies are descriptive.

Furthermore, several GBM scRNA-seq datasets, including immune cellular landscape, have been published.

Although we agree that mouse GBM immune TME has been characterized using scRNAseq and CyTOF, none of the published datasets were generated from a clinically relevant GBM model but rather from implantable GL261 and other syngeneic tumor models that carry irrelevant mutations (such as Ras in GL261), that are not relevant to human GBM. Furthermore,

implantable models do not allow for studies of immune TME changes over time due to their initiation nature (implantation of 50,000-100,000 cells). More importantly, there has been no studies evaluating the changes in TME during GBM growth and evolution over time, from early lesions to full-blown symptomatic GBM. This is a novelty of our study. Please also see response to a relevant comment from reviewer #1.

Nonetheless, the cellular changes observed in the course of tumor progression and under different therapies may be useful and informative in designing novel preclinical and clinical studies on glioblastoma.

Major comments:

1. The single-cell sequencing was conducted in a single tumor in the late stage (S3), two tumors in early stage (S1) and mid-stage (S2). This design has its inherited limitation.

We agree with the reviewer's comment and we have addressed this very important limitation by adding 2 samples (derived from 2 mice pooled each) of normal brain controls, 2 additional early stage GBMs (each from 2 mice pooled) and 2 additional Late stage GBMs. Combined with our original datasets we now have a total of 2 normal control brains, 4 Early GBMs and 3 Late stage GBMs.

2. The paper described the major cellular components, including CD45+ immune cells and CD45- non-immune cells, and their dynamic changes in the tumor microenvironment (TME). However, the biological importance of different populations and the paired ligand-receptor interaction is unknown.

2.1 Tumor cells: Two clusters of tumor cells were identified. Different functional signatures were observed. a. Regionalization of these two populations was hypothesized, but no analysis has been done in the mouse model, although some data from patient samples support this. It is feasible and should be validated at least in the mouse model. Besides, it was hypothesized that the distinct gene expression profiles observed are influenced by their localization or regionalization, the authors provided no analysis and evidence, supporting this possibility.

This has been addressed above with one of reviewer 1 comments. It is correct that few statements in the manuscript are hypothetical. We included such hypothesis in the discussion of

our findings as intriguing possibilities that could form the basis of future studies. These can be [REDACTED].

b. Different EGFR ligands were expressed by the two different tumor clusters. Based on the scRNA-seq data, are we able to discriminate whether these ligands are functional in autocrine or paracrine ways?

Unfortunately, the scRNAseq data does not allow us to address this interesting question.

Since different ligands can exert different effects on downstream signaling, does the downstream signaling of EGFR show difference between these two clusters of tumor cells? If yes, does the difference of EGFR signaling drive the distinct expression profile since EGFR is a tumor driver mutation.

A similar question was addressed above with one of reviewer 1 comments. We addressed this by identifying markers that label two populations of EGFR+ cells (PDGFRA+ and GAL-1+) and isolating these populations by FACS and performing bulk RNAseq on these cells. These new data are shown in Fig. 2 of the revised manuscript. We discovered substantial differences in their transcriptome with respect to cytokine expression/production. We surmise that these are the result of EGFR signaling pathways being different in these two population of cells, perhaps due to a difference in ligand utilization and the differential presence of other signaling members such as PDGFRA for example.

c. Are these two clusters developmentally different? Are they two independent clusters in tumor initiation? Or one cluster is originated/developed/reprogrammed from the other one? For scRNA-seq data, pseudotime and RNA-velocity trajectory analysis may provide insights into this.

We have performed pseudotime analysis and demonstrated that cluster 10 give rise to clusters 0, 22 and 24 (Fig. 4d of the revised manuscript).

2.2 Microglia

a. Cluster-4 was also defined as microglia, but it was not discussed at all.

Does CD45- microglia have different function comparing to CD45+ microglia?

This is not relevant anymore in the new UMAPs.

b. Three CD45+ microglia clusters (C-2/5/6) were observed. C-6 is a proliferating cluster and seems to be stable regarding its proportion from early to late stage. However, C-2 and C-5 were decreased. How did this happen?

In our new analysis with more samples, we still observed several clusters of microglia, and clusters 2 and 4 slightly decrease in numbers between Early and Late GBMs. We do not know the causative mechanisms. It is possible that this could simply be the result of other populations of cells (EGFR+, PMN-MDSCs, macrophages, DCs) infiltrating the GBMs and making it look like some microglia are decreasing in number.

2.3 Macrophage

- a. Increased macrophage population was observed at mid-S2 and late-S3 stage. Are they monocyte-derived macrophages (e.g. CCR2 expression)?
- b. Macrophage population shows high heterogeneity (Figure 5A). This heterogeneity and the dynamic changes were not discussed.

We now have different clusters of macrophages, some are brain resident and others arise from monocytes. We further refined our analysis of the heterogeneity and dynamics of our macrophage populations and discuss in the manuscript.

c. The study proposed that “macrophage infiltration coincides with BBB disruption”. However, the disruption of BBB occurred at a later stage (Figure 5F,G), rather than early/mid stage (Figure 5A-E).

Of the CD45+ cells, the macrophages increase between 25 and 30+ days, which coincide with a disruption of the BBB.

3. The current analysis looked at cytokines and cytokine receptors, as well as checkpoint molecules and checkpoint receptors.

- a. Was the analysis able to provide information on which pair(s) of molecular interaction and cellular interaction, play critical role(s) in the TME and during tumor progression. Interactome analysis (e.g. PMID: 31675496, 32302573) may provide insights regarding this part.

Most of revised Fig. 4 covers exactly what the reviewer is referring to. Our goal was to demonstrate that the major source of chemokines/cytokines were from CD45+ cells. We did identify a pair of receptor-ligand pair (Cxcr4-Cxcl12) where endothelial cells express Cxcl12 and several immune cells (CD4, CD8 T cells, NK cells, neutrophils, macrophages, DCs) that express Cxcr4 receptors. We added data to demonstrate that during GBM progression, the levels of Cxcr4 do not significantly fluctuate except perhaps in neutrophil cluster 25 (Fig. 6c). We discuss the potential consequences of this in the text although the interpretation of these kinds of analyses is somewhat difficult. For instance, ligands are secreted proteins that can physiologically travel/migrate considerable distances, thus weakening the concept of cell-cell interactions based on cytokine receptor-ligand pairs. Instead, we provide additional analyses on cytokine receptor-ligand pairs expression over time, which we feel are informative in terms of changes in TME during GBM progression.

b. Figure 4A may be a biased analysis, how were the 37 cytokines selected?"

These cytokines were selected from the literature for their well-validated qPCR probes. We understand the perception of a biased analysis, however the point we are making is that most cytokines within our sampling are expressed and produced by CD45+ cells.

c. For checkpoint related analysis, myeloid cells (DCs and macrophages) express checkpoints. The present analysis demonstrated high levels of checkpoint molecules in myeloid cells. However, the analysis reflects gene expression per cell, the quantity of different cell subsets in the TME should be at least discussed.

We have discussed the quantity of different cell subsets in the text.

4. The study described the changes in immune cell subsets in the TME, following different therapeutic treatments. However, whether and how immune cell contribute to the outcome is unknown.

The focus of the manuscript is on establishing the immune TME first with independent follow up studies on how specific cell type contribute to the outcome.

5. The analysis has several potential therapeutic applications. But none of them were tested

experimentally.

We agree that functional experimentations are important however they are beyond the scope of the present study, which provides an atlas of the immune populations during GBM progression, and are best suited for follow up studies. We have added text to the discussion that pertains to this effect.

Other specific comments:

6. At which time points Early-S1, Mid-S2, Late-S3 samples were collected?

We now have a table (Extended Data Table 1) with details on time of collection for each samples.

7. Figure1C, E, the resolution is not enough to read the individual genes.

These panels have been replaced in the resubmission. We apologize for this inconvenience.

8. Figure 5D is not consistent with BLI data in Fig5A. BLI data shows strong signal in mid-S2 stage, while not much change in mid-S2 to late-S3 stage. Please provide explanation.

The addition of more samples shows that BLI data does track with Early vs Late samples.

9. Figure 5E, Y-axis is different. Is it mislabeled?

This is no longer an issue in the resubmission.

10. Figure 6D, M2 markers are not consistent with the gene expression analysis which show upregulated M2 signature at a late stage.

This is no longer an issue in the resubmission.

11. Figure7D, please check the sample size since it's inconsistent.

We have fixed the discrepancy. We apologize for the inconvenience.

12. Please call the figures out in order. E.g. Figure 6A,C, B

All figures are now called in order in the text.

13. Figure 7A, B, T cells are increased, are they conventional T cells or Treg cells (CTLA4+)? Treg cells are increased as shown in Figure 6E.

In our new analysis, we discuss the fluctuation of the various cell types during GBM progression and better define the T cell populations, including T regs.

Decision Letter, first revision:

Subject: Your manuscript, NI-RS32188A

Message: Our ref: NI-RS32188A

14th Mar 2022

Dear Dr. Charest,

Thank you for your patience as we've prepared the guidelines for final submission of your Nature Immunology manuscript, "Single-Cell RNA sequencing reveals evolution of immune landscape during Glioblastoma progression" (NI-RS32188A). Please carefully follow the step-by-step instructions provided in the attached file, and add a response in each row of the table to indicate the changes that you have made. Please also check and comment on any additional marked-up edits we have proposed within the text. Ensuring that each point is addressed will help to ensure that your revised manuscript can be swiftly handed over to our production team.

When you upload your final materials, please include a point-by-point response to any remaining reviewer comments and please make sure to upload your checklist.

In recognition of the time and expertise our reviewers provide to Nature Immunology's editorial process, we would like to formally acknowledge their contribution to the external peer review of your manuscript entitled "Single-Cell RNA sequencing reveals evolution of immune landscape during Glioblastoma progression". For those reviewers who give their assent, we will be publishing their names alongside the published article.

Nature Immunology offers a Transparent Peer Review option for new original research manuscripts submitted after December 1st, 2019. As part of this initiative, we encourage our authors to support increased transparency into the peer review process by agreeing to have the reviewer comments, author rebuttal letters, and editorial decision letters published as a Supplementary item. When you submit your final files please clearly state in your cover letter whether or not you would like to participate in this initiative. Please note that failure to state your preference will result in delays in accepting your manuscript for publication.

Cover suggestions

As you prepare your final files we encourage you to consider whether you have any images or illustrations that may be appropriate for use on the cover of Nature Immunology.

Nature Immunology has now transitioned to a unified Rights Collection system which will allow our Author Services team to quickly and easily collect the rights and permissions required to publish your work. Approximately 10 days after your paper is formally accepted, you will receive an email in providing you with a link to complete the grant of rights. If your paper is eligible for Open Access, our Author Services team will also be in touch regarding any additional information that may be required to arrange payment for your article.

Please note that *Nature Immunology* is a Transformative Journal (TJ). Authors may publish their research with us through the traditional subscription access route or make their paper immediately open access through payment of an article-processing charge (APC). Authors will not be required to make a final decision about access to their article until it has been accepted. [Find out more about Transformative Journals](https://www.springernature.com/gp/open-research/transformative-journals).

If you have any questions about costs, Open Access requirements, or our legal forms, please contact ASJournals@springernature.com.

Please use the following link for uploading these materials: [REDACTED]

Best regards,

Elle Morris
Senior Editorial Assistant
Nature Immunology
Phone: 212 726 9207
Fax: 212 696 9752
E-mail: immunology@us.nature.com

On behalf of

Laurie A. Dempsey, Ph.D.
Senior Editor
Nature Immunology
l.dempsey@us.nature.com
ORCID: 0000-0002-3304-796X

Reviewer #1:

Remarks to the Author:

I am satisfied with the revised version of this manuscript and it should be considered for publication.

Reviewer #2:

Remarks to the Author:

The paper has been improved. However, the work remains descriptive in nature, lacking mechanistic insights into glioblastoma etiology, immunity, and therapy.

Final Decision Letter:**Subject:** Decision on Nature Immunology submission NI-RS32188B**Message:** In reply please quote: NI-RS32188B

Dear Dr. Charest,

I am delighted to accept your manuscript entitled "Single-Cell RNA sequencing reveals evolution of immune landscape during Glioblastoma progression" for publication in an upcoming issue of Nature Immunology.

Over the next few weeks, your paper will be copyedited to ensure that it conforms to Nature Immunology style. Once your paper is typeset, you will receive an email with a link to choose the appropriate publishing options for your paper and our Author Services team will be in touch regarding any additional information that may be required.

Please note that *Nature Immunology* is a Transformative Journal (TJ). Authors may publish their research with us through the traditional subscription access route or make their paper immediately open access through payment of an article-processing charge (APC). Authors will not be required to make a final decision about access to their article until it has been accepted. [Find out more about Transformative Journals](https://www.springernature.com/gp/open-research/transformative-journals).

Authors may need to take specific actions to achieve [compliance](https://www.springernature.com/gp/open-research/funding/policy-compliance-faqs) with funder and institutional open access mandates. If your research is supported by a funder that requires immediate open access (e.g. according to [Plan S principles](https://www.springernature.com/gp/open-research/plan-s-compliance)) then you should select the gold OA route, and we will direct you to the compliant route where possible. For authors selecting the subscription publication route, the journal's standard licensing terms will need to be accepted, including [journal-](https://www.springernature.com/gp/open-research/policies/journal-)

policies">self-archiving policies. Those licensing terms will supersede any other terms that the author or any third party may assert apply to any version of the manuscript.

Your paper will be published online soon after we receive your corrections and will appear in print in the next available issue. Content is published online weekly on Mondays and Thursdays, and the embargo is set at 16:00 London time (GMT)/11:00 am US Eastern time (EST) on the day of publication. Now is the time to inform your Public Relations or Press Office about your paper, as they might be interested in promoting its publication. This will allow them time to prepare an accurate and satisfactory press release. Include your manuscript tracking number (NI-RS32188B) and the name of the journal, which they will need when they contact our office.

About one week before your paper is published online, we shall be distributing a press release to news organizations worldwide, which may very well include details of your work. We are happy for your institution or funding agency to prepare its own press release, but it must mention the embargo date and Nature Immunology. Our Press Office will contact you closer to the time of publication, but if you or your Press Office have any enquiries in the meantime, please contact press@nature.com.

Also, if you have any spectacular or outstanding figures or graphics associated with your manuscript - though not necessarily included with your submission - we'd be delighted to consider them as candidates for our cover. Simply send an electronic version (accompanied by a hard copy) to us with a possible cover caption enclosed.

If you have not already done so, we strongly recommend that you upload the step-by-step protocols used in this manuscript to the Protocol Exchange. Protocol Exchange is an open online resource that allows researchers to share their detailed experimental know-how. All uploaded protocols are made freely available, assigned DOIs for ease of citation and fully searchable through nature.com. Protocols can be linked to any publications in which they are used and will be linked to from your article. You can also establish a dedicated page to collect all your lab Protocols. By uploading your Protocols to Protocol Exchange, you are enabling researchers to more readily reproduce or adapt the methodology you use, as well as increasing the visibility of your protocols and papers. Upload your Protocols at www.nature.com/protocolexchange/. Further information can be found at

www.nature.com/protocolexchange/about .

Please note that we encourage the authors to self-archive their manuscript (the accepted version before copy editing) in their institutional repository, and in their funders' archives, six months after publication. Nature Research recognizes the efforts of funding bodies to increase access of the research they fund, and strongly encourages authors to participate in such efforts. For information about our editorial policy, including license agreement and author copyright, please visit www.nature.com/ni/about/ed_policies/index.html

Kind regards,

Laurie

Laurie A. Dempsey, Ph.D.
Senior Editor
Nature Immunology
l.dempsey@us.nature.com
ORCID: 0000-0002-3304-796X